# A provable control of sensitivity of neural networks through a direct parameterization of the overall bi-Lipschitzness

**Yuri Kinoshita, Taro Toyoizumi**
Department of Mathematical Informatics,
Graduate School of Information Science and Technology,
The University of Tokyo, Tokyo, Japan.
Laboratory for Neural Computation and Adaptation,
RIKEN Center for Brain Science, Wako, Japan.
`yuri-kinoshita111@g.ecc.u-tokyo.ac.jp`

## Abstract

While neural networks can enjoy an outstanding flexibility and exhibit unprecedented performance, the mechanism behind their behavior is still not well-understood. To tackle this fundamental challenge, researchers have tried to restrict and manipulate some of their properties in order to gain new insights and better control on them. Especially, throughout the past few years, the concept of *bi-Lipschitzness* has been proved as a beneficial inductive bias in many areas. However, due to its complexity, the design and control of bi-Lipschitz architectures are falling behind, and a model that is precisely designed for bi-Lipschitzness realizing a direct and simple control of the constants along with solid theoretical analysis is lacking. In this work, we investigate and propose a novel framework for bi-Lipschitzness that can achieve such a clear and tight control based on convex neural networks and the Legendre-Fenchel duality. Its desirable properties are illustrated with concrete experiments to illustrate its broad range of applications.

## 1 Introduction

### 1.1 Background

Nowadays, neural networks have become an indispensable tool in the field of machine learning and artificial intelligence. While they can enjoy an outstanding flexibility and exhibit unprecedented performance, the mechanism behind their behavior is still not well-understood. To tackle this fundamental challenge, researchers have tried to restrict and manipulate some of their properties in order to gain new insights and better control on them. Especially, throughout the past few years, the concept of *sensitivity* has been proved as a beneficial inductive bias in many areas.

Sensitivity can be translated into the concept of *bi-Lipschitzness*, which combines two different properties, namely, *Lipschitzness* and *inverse Lipschitzness*. The former describes the maximal, and the latter the minimal sensitivity of a function. Bi-Lipschitzness is attracting interests in various fields not only as it proposes a promising solution to avoid unexpected and irregular results caused by a too sensitive or too insensitive behavior of the trained function, but also as it achieves an approximate isometry that preserves geometries of the input dimension (Li et al., 2020). It plays an essential role in generative models such as normalizing flows to guarantee invertibility (Behrmann et al., 2019), in uncertainty estimation to prevent the recurrent problem of *feature collapse* (Liu et al., 2020a; Van Amersfoort et al., 2020) and in inverse problems (Kruse et al., 2021) to assure the stability of both the forward and inverse function (Behrmann et al., 2021).

Unfortunately, the appropriate design and effective control of bi-Lipschitz neural networks are far from simple, which hinders their application to prospective areas. First of all, the *estimation* of bi-Lipschitz constants is an NP-hard problem (Scaman and Virmaux, 2018). Second, the *design* of bi-Lipschitz models is even harder as we cannot straightforwardly extend existing Lipschitz architectures that exploit some unique properties of the concept to inverse Lipschitzness and keep their advantages, and *vice-versa*. Finally, the *control* of bi-Lipschitz constants requires particular attention as it is a question of manipulating two distinct concepts in a preferably independent and simple manner.

Currently existing bi-Lipschitz models still present some issues in terms of design or control. On the one hand, some lack theoretical guarantees because they impose soft constraints by adding regularization terms to the loss function (Van Amersfoort et al., 2020), or because their expressive power is not well-understood and may be more limited than expected. On the other hand, others restrict bi-Lipschitzness on a layer-wise basis (Behrmann et al., 2019; Liu et al., 2020a). Particularly, this means these approaches can only build in the essence a simple bi-Lipschitz function employed as a layer of a more complex neural network. In practice, this kind of parameterization impacts the generalization ability of the model or leads to loose control (Fazlyab et al., 2019). They can also contain so many parameters affecting sensitivity to the same extent that controlling all of them is unrealistic and fixing some may affect the expressive power. See Section 2 for further details.

Therefore, taking into account both the potential and complexity of this inductive bias, we first and foremost need a model that is precisely designed for bi-Lipschitzness realizing a direct and simple control of the constants of the overall function more complex than a single layer neural network equipped with solid theoretical analysis. In this work, we investigate an architecture that can achieve such a clear and tight control and apply it to several problem settings to illustrate its effectiveness.

## 1.2 Contributions

Our contributions can be summarized as follows. First, we construct a model bi-Lipschitz by design based on *convex neural networks* and the *Legendre-Fenchel duality*, as well as a *partially* bi-Lipschitz variant. This architecture provides a simple, direct and tight control of the Lipschitz and inverse Lipschitz constants through only two parameters, the ideal minimum, equipped with theoretical guarantees. These characteristic features are illustrated and supported by several experiments including comparison with prior models. Finally, we show the utility of our model in concrete machine learning applications, namely, uncertainty estimation and monotone problem settings and show that it can improve previous methods.

**Organization**  In Section 2, we will first explain in more detail existing architectures around bi-Lipschitzness. In Section 3, we will develop our model followed by theoretical analyses. The next Section 4 will be devoted to experiments and applications of our proposed method.

**Notation**  Throughout this paper, the Euclidean norm is denoted as $\|\cdot\|$ for vectors unless stated otherwise. Similarly, for matrices, $\|\cdot\|$ corresponds to the matrix norm induced by the Euclidean norm. For a real-valued function $F : \mathbb{R}^m \to \mathbb{R}$, $\nabla F$ is defined as $(\partial f(x)/\partial x_1, \ldots, \partial f(x)/\partial x_n)^\top$, and $\nabla^\top F$ as its transpose. When the function is a vector $F : \mathbb{R}^m \to \mathbb{R}^n$, then its Jacobian is defined as $\nabla^\top F = (\partial f_i/\partial x_j)_{i,j}$. The subderivative is denoted as $\partial_{\mathrm{sub}}$.

## 2 Preliminaries

In this section, we explain mathematical backgrounds and existing bi-Lipschitz models to clarify the motivation of this work.

### 2.1 Definition

Let us first start by the definition of bi-Lipschitzness.

**Definition 2.1** (bi-Lipschitzness). *Let $0 < L_1 \leq L_2$. $f : \mathbb{R}^l \to \mathbb{R}^t$ is $(L_1, L_2)$-bi-Lipschitz if $L_1\|x - y\| \leq \|f(x) - f(y)\| \leq L_2\|x - y\|$ holds for all $x, y \in \mathbb{R}^l$. The right (left) inequality is the (inverse) Lipschitzness with constant $L_2$ ($L_1$). $L_1$ and $L_2$ are called together bi-Lispchitz constants.*

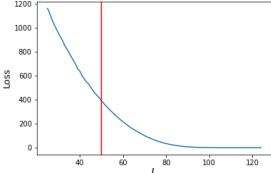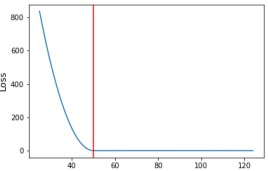

Figure 1: Results of fitting $y = 50x$ with a Lipschitz model (SN (left) or our model (right)), where the Lipschitz constant is constrained by an upper bound $L$. $L = 50$ (red line) is where an $L$-Lipschitz model with perfect tightness and expressive power should achieve a 0 loss for the first time. SN achieves this only from around $L = 100$ while ours at $L = 50$. See Appendix G.4 for further details.

Several interpretations can be attributed to this definition. Bi-Lipschitzness appears when we want to guarantee a high sensitivity and a high smoothness for better generalization such as in uncertainty estimation. This can be also regarded as a quasi-isometry with a distortion of $L_2/L_1$, which means that the structure of the input is relatively inherited in the output as well. Moreover, since a bi-Lipschitz function is by definition invertible, the inverse Lipschitz can be interpreted as the Lipschitz constant of the inverse function. Therefore, controlling the bi-Lipschitz constants is beneficial for the stability of both the forward and inverse.

## 2.2 Desired Features for a Controllable Inductive Bias

In order to gain new insights of a problem and a predictable behavior of the bi-Lipschitz model, it is primordial that bi-Lipschitzness works as an inductive bias with high controllability and theoretical foundation. For this goal, the following points are of particular interest: (a) bi-Lipschitzness guaranteed by design, (b) theoretical bounds on the bi-Lipschitz constants, (c) tight bounds, (d) independent control of the Lipschitz and inverse Lipschitz constants, (e) a minimal number of hyper-parameters to tune, (f) theoretical guarantee on the expressive power and (g) direct parameterization of the optimized variables (in the sense of Definition 2.3 of Wang and Manchester (2023)).

## 2.3 Related Works and Bi-Lipschitz Models

There are currently three methods that are mainly employed to achieve and control bi-Lipschitz neural networks. See Appendix A for a brief overview on Lipschitz and inverse Lipschitz architectures.

**Regularization**    The first one is based on a regularization method (Gulrajani et al., 2017; Van Amers-foort et al., 2020). It adds to the loss a term that incites bi-Lipschitzness. While the method is simple, the resulting function may not be bi-Lipschitz at the end of the training, we have no theoretical bounds on the bi-Lipschitz constants and cannot achieve a real control of them.

**i-ResNet**    The invertible residual network (i-ResNet) of Behrmann et al. (2019) is based on the composition of several $(1 - L_{g_i}, 1 + L_{g_i})$-bi-Lipschitz layers of the type $f_i(x) = x + g_i(x)$, where $g_i$ is Lipschitz with constant $L_{g_i} < 1$. The Lipschitz $g_i$ can be constructed by layer-wise methods such as spectral normalization of weights (SN) (Miyato et al., 2018). However, this kind of layer-wise control of (bi-)Lipschitzness is known to be loose in general (Fazlyab et al., 2019). See Figure 1 for an illustration. The i-ResNet has limited expressive power as it cannot represent the function $y = -x$ (Zhang et al., 2020), and the construction of $g_i$ risks to introduce more restriction on the expressive power in practice (Anil et al., 2019). A similar model, the Lipschitz monotone network (LMN) of Nolte et al. (2023) combines the GroupSort activation function with SN to create $g_i$. These approaches requires to adjust the Lipschitzness of each weight during the training process, which may be sub-optimal (i.e., no direct parameterization).

**BiLipNet**    The concurrent work of Wang et al. (2024) also provides a bi-Lipschitz neural network (BiLipNet), which was mainly used for creating a Polyak-Łojasiewicz function useful in surrogate loss learning. It extends the Lipschitz architecture of Wang and Manchester (2023). The BiLipNet is constructed as the composition of monotone Lipschitz layers and orthogonal layers. The former layer is realized based on a direct parameterization of the IQC theory of Megretski and Rantzer

(1997). While they can achieve a certified control of the bi-Lipschitz constant, theoretical guarantee on its expressive power is lacking. Moreover, BiLipNet composes many layers which may harm the tightness of the bounds.

Therefore, current models fail to satisfy the desired features of Subsection 2.2. Notably, all these points cannot be satisfied as long as we rely on the simple layer-wise control which offers too conservative bounds and a number of tunable hyper-parameters proportional to that of the layers in general. In this work, we will establish a bi-Lipschitz architecture that satisfies almost all these expected attributes mentioned above. In terms of mathematical foundations, it takes inspiration from the inverse Lipschitz architecture of Kinoshita et al. (2023).

## 3 Bi-Lipschitz Neural Network

We introduce our novel bi-Lipschitz model, followed by some theoretical analyses and discussion.

### 3.1 Additional Definitions

We clarify notions of smoothness, convexity and the Lengendre-Fenchel transformation (LFT) which will be a core process in our construction.

**Definition 3.1.** *Let $\gamma > 0$. $F : \mathbb{R}^m \to \mathbb{R}$ is $\gamma$-smooth if $F$ is differentiable and $\nabla F$ is $\gamma$-Lipschitz.*

**Definition 3.2.** *Let $\mu > 0$. $F : \mathbb{R}^m \to \mathbb{R}$ is $\mu$-strongly convex if $F(x) - \frac{\mu}{2}\|x\|^2$ is convex.*

**Definition 3.3.** *Let $F : I \to \mathbb{R}$ a convex function over $I \subset \mathbb{R}^m$. Its Legendre-Fenchel transformation $F^*$ is defined as $F^*(x) := \sup_{y \in I} \{\langle y, x \rangle - F(y)\}$.*

### 3.2 Construction of Bi-Lipschitz Functions

We extend the approach of Kinoshita et al. (2023) and propose a method that creates bi-Lipschitz functions, where it is sufficient to control two parameters to manipulate the overall bi-Lipschitzness without needing to dissect the neural network and control each layer, a feature that existing methods for (bi-)Lipschitzness cannot deliver. The first step is to notice that the gradient of a real-valued $\mu$-strongly convex function becomes $\mu$-inverse Lipschitz, and that of a $\gamma$-smooth function $\gamma$-Lipschitz by definition. Therefore, we aim to compose a function which is both strongly convex and smooth. Interestingly, the LFT of a $1/\beta$-strongly convex function is $\beta$-smooth. This leads to our main theorem.

**Theorem 3.4.** *Let $F$ be a closed $1/\beta$-strongly convex function and $\alpha \geq 0$. Then the following function is $\alpha$-strongly convex and $\alpha + \beta$-smooth: $\sup_{y \in I} \{\langle y, x \rangle - F(y)\} + \frac{\alpha}{2}\|x\|^2$. Thus, its derivative is $(\alpha, \alpha + \beta)$-bi-Lipschitz which equals $f^*(x) := \text{argmax}_y \{\langle y, x \rangle - F(y)\} + \alpha x$.*

See Appendix B.2 for the proof. The term $\alpha/2\|x\|^2$ has the effect of turning a convex function into an $\alpha$-strongly convex function by definition of strong convexity. This $f^*(x)$ constructed as described in the above theorem is precisely the bi-Lipschitz model we propose in this paper.

### 3.3 Implementation of the Forward Pass

Based on Theorem 3.4, we can create a bi-Lipschitz function parameterized by neural networks, and thus applicable to machine learning problems. Here, we clarify the implementation of a strongly convex function and the LFT. The overall explicit formulation of our bi-Lispchitz neural network (BLNN) is already shown in Algorithm 1 for convenience.

**Strongly Convex Neural Networks**   A $\mu$-strongly convex neural network can be constructed by adding a regularization term to the output of the Input Convex Neural Network (ICNN) from Amos et al. (2017). The resulting structure can be written as $F_\theta(y) = z_k + \frac{\mu}{2}\|y\|^2$, where $z_{i+1} = g_i(W_i^{(z)} z_i + W_i^{(y)} y + b_i)$ $(i = 0, \ldots, k-1)$. $\{W_i^{(z)}\}_i$ are non-negative, $W_0^{(z)} = 0$, and all functions $g_i$ are convex and non-decreasing. This architecture is a universal approximator of $\mu$-strongly convex functions defined on a compact domain endowed with the sup norm (Chen et al., 2019). Note that any other choice for the convex architecture is possible. This is only an example, also employed in Huang et al. (2021) and Kinoshita et al. (2023). Indeed, we could opt for a convolutional version by using the convolutional ICNN proposed by Amos et al. (2017).

---
**Algorithm 1:** Forward pass of $(\alpha, \beta)$-BLNN
---
**Input**: input data $x$, an ICNN $G_\theta$ with parameters $\theta$ and constants $\alpha, \beta \geq 0$
**Output** : image of $x$ by an $(\alpha, \alpha + \beta)$-bi-Lipschitz function
Step 0: Construct $\frac{1}{\beta}$-strongly convex function $F_\theta(y) = G_\theta(y) + \frac{1}{2\beta}\|y\|^2$
Legendre-Fenchel transformation Step: Find $y_\theta^*(x) = \text{argmax}_y \{\langle y, x \rangle - F_\theta(y)\}$
Gradient Step: Compute $f_\theta^*(x) = \nabla_x \left(F_\theta^*(x) + \frac{\alpha}{2}\|x\|^2\right) = y_\theta^*(x) + \alpha x$
**return** $f_\theta^*(x)$
---

**Algorithms for LFT** By Theorem 3.4, we only need to compute the optimal point of the LFT, i.e., $y_\theta^*(x) := \text{argmin}_y \{F_\theta(y) - \langle y, x \rangle\}$, which is a strongly convex optimization. This computation is thus rather fast, and we can use various algorithms with well-known convergence guarantees such as the steepest gradient descent (Shamir and Zhang, 2013; Bansal and Gupta, 2017). Such convex solvers will generate for a fixed $x$ a sequence of points $\{y_t(x)\}_{t=0,...,T}$, and its last point will be an estimate of $y_\theta^*(x)$. However, this kind of discrete finite time approximation could compromise the bi-Lipschitzness of the whole algorithm. The bi-Lipschitz behavior of $y_t(x)$ can be explicitly described for the steepest gradient descent as follows. For other algorithms, see Appendix C.2.1.

**Theorem 3.5.** *Let the symbols defined as in Algorithm 1. Consider the steepest gradient descent of $\sup_y \{\langle y, x \rangle - F_\theta(y)\}$ generating points $\{y_t(x)\}_t$ at the $t$-th iterations and $y_\theta^*(x)$ is the global maximum. If $F_\theta$ is $\mu$-strongly convex and $\gamma$-smooth then the point $\lim_{t \to \infty} y_t(x)$ is $(1/\gamma, 1/\mu)$-bi-Lipschitz without any bias. Moreover, with $\eta_t = 1/(\mu(t+1))$ as a step size and $y_0(x_i) = y_0$ as initial point, then for all $x_i$, $x_j$, $\|y_{t+1}(x_i) - y_{t+1}(x_j)\| \leq h(t)\|x_i - x_j\|$ where $\lim_{t \to \infty} h(t) = 1/\mu$.*

See Appendix C.2.2 for the concrete formulation of $h(t)$ and the proof. The above theorem guarantees that the optimization scheme will ultimately provide a $y_t(x)$ that is bi-Lipschitz with the right constants. This was not trivial as a bias may have persisted due to the non-zero step size. More importantly, this statement offers a non-asymptotic bound for Lipschitzness. This is greatly useful when we theoretically need to precisely assure a certain degree of Lipschitzness such as in robustness certification against adversarial attacks (Szegedy et al., 2013). The step size $\eta_t = 1/(\mu(t+1))$ can be precisely calculated as $\mu = 1/\beta$ for an $(\alpha, \beta)$-BLNN. Experimental results suggest a similar behavior for inverse-Lipschitzness. Note the strong convexity of $F$ is always satisfied in our setting and smoothness can be fulfilled if we use an ICNN with softplus activation functions $\log(1 + e^x)$. Since this simple gradient descent has been thoroughly analysed and assured to properly behave in practice as well, we will employ it as the algorithm of LFT in the remainder of this paper.

## 3.4 Expressive Power

As we mentioned earlier, layer-wise approaches realize a bi-Lipschitz neural network by restricting the sensitivity of each layer. Nevertheless, this kind of construction may be sub-optimal as it does not take into account the interaction between layers, limiting more than expected the expressive power, which was often not considered in the original papers. Layer-wise bi-Lipschitz approaches cannot inherit the proofs and properties of the original network, which makes the understanding of their behavior more difficult. As for our model, the $(\alpha, \beta)$-BLNN, we can guarantee the following universality theorem.

**Theorem 3.6.** *For any proper closed $\alpha$-strongly convex and $\alpha + \beta$-smooth function on a compact domain, there is a BLNN without taking the gradient at the end, i.e., $\sup_y\{\langle y, x \rangle - (G_\theta(y) + \|y\|^2/(2\beta))\} + \alpha\|x\|^2/2$ where $G_\theta$ is an ICNN with ReLU or softplus-type activation function, that approximates it within $\epsilon$ in terms of the sup norm.*

Thus, after taking the gradient, we can create a sequence of functions that converges point-wise to any function which is $\alpha + \beta$-Lipschitz, $\alpha$-strongly monotone (Definition B.12) and the derivative of a real-valued function. See Appendix B.3 for the proofs and further discussion.

## 3.5 Backward Pass of BLNN

The most straightforward way to compute the gradient of a BLNN is to track the whole computation of the forward pass including the optimization of the LFT and back-propagate through it. However,

this engenders a crucial bottleneck since back-propagating over the for-loop of the optimization involves many computations of the Hessian and the memorization of the whole computational graph. Nevertheless, if the activation function of the ICNN is chosen as softplus, the convergence of the LFT is quite fast making this strategy scalable to moderately large data.

Interestingly, when $F_\theta := F(\cdot; \theta)$ is $C^2$, the backward pass can be computed only with the information of the core ICNN, which means we do not need to track the whole forward pass involving many recurrent computations:

**Theorem 3.7.** *Suppose a loss function $L : z \mapsto L(z)$, and the output of the BLNN is $f^*(x; \theta) := \nabla F_\theta^*(x) + \alpha x$ as defined in Algorithm 1. If $F$ is $C^2$ and $F^*$ is differentiable, then the gradient $\nabla_\theta^\top L(f^*(x; \theta))$ can be expressed as $-\nabla_z^\top L(z) \left\{ \nabla_y^2 F(y^*(x; \theta); \theta) \right\}^{-1} \partial_\theta^\top \nabla_y F(y^*(x; \theta); \theta)$.*

See Appendix B.4 for the proof and formulation with more complex architectures. In practice, coding is simple since we can directly employ the backward algorithm provided in libraries such as Pytorch (Corollary B.19). The advantage of this second method is threefold. First, it can reduce the computational and memory cost of the backward process as it does not depend on the iteration number and involves fewer matrix manipulations. Second, this leaves the freedom to choose the optimizer to calculate the LFT, while the brute force method requires the gradient to be tracked during the optimization process, which is a feature that many current solvers do not provide. Third, this method approximates the gradient by taking the derivative along the true curve at a point $y_\theta^{(t)}(x)$ close to $y_\theta^*(x)$, which is a fair approximation. In contrast, the brute force method approximates the gradient based on the recurrent computation of $y_\theta^{(t)}(x)$ but we do not know whether this is really a good estimate.

### 3.6 Comparison with Deep Equilibrium Models

Interestingly, our BLNN can be regarded as a bi-Lipschitz version of a broader family of models called deep equilibrium models (DEQs) from Bai et al. (2019). A DEQ is an implicit model whose output $z$ is defined as the fixed point of a neural network $h_\theta$, i.e., $z = h_\theta(z, x)$, where $x$ is the input. Similarly, the output of our algorithm can be re-formulated as finding the solution $z$ of $z = x - \nabla F_\theta(z) + z$ which corresponds to a DEQ with $h_\theta(z, x) = x - \nabla F_\theta(z) + z$. The general properties of a DEQ concerning the computational complexity (both time and space) are thus also inherited in our model. One major difference is that the iteration to find this fixed point $z$ is in our case unique and guaranteed to converge, which addresses one of the main concerns of general DEQs. We believe this interpretation is promising for future work to increase the generality of our model as it enables to converge to the larger flow of work around DEQs and apply the various improvements for DEQs researched so far. However, we will not pursue this direction further since our primary goal is to develop a bi-Lipschitz model with direct and simple control.

### 3.7 Increased Scalability and Expressive Power: Partially BLNN

Despite all the theoretical guarantees mentioned above and the correspondence of the proposed method to DEQs, its main drawbacks persist in computational efficiency and expressive power. However, these weaknesses can be alleviated by imposing the bi-Lipschitz requirement on a limited number of variables. This can be realized by using the partially input convex neural network (PICNN) instead of the ICNN in our architecture (Amos et al., 2017). A PICNN is convex with respect to a pre-defined limited set of variables. Based on this architecture, we can proceed similarly to Algorithm 1 and obtain a *partially* bi-Lipschitz neural network (PBLNN). See Appendix F.2 for further details. As a result, all heavy operations such as the LFT and the gradient are applied on this smaller set of variables, which makes the architecture much more scalable to higher dimensions. Moreover, a PICNN with $k$ layers can represent any ICNN with $k$ layers and any purely feedforward network with $k$ layers, as shown in Proposition 2 of Amos et al. (2017). This also enhances the expressive power of our model with larger liberty on the precise construction of the architecture.

### 3.8 Computational Complexity

Regarding the time and space complexity of our model, it is largely equivalent to that of a DEQ (Bai et al., 2019) except that the core function is the derivative of a neural network, which adds some computational burden. In Figure 2, we present a comparison of the computational complexity in

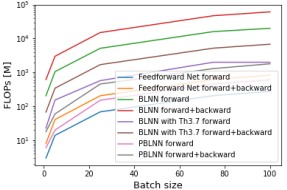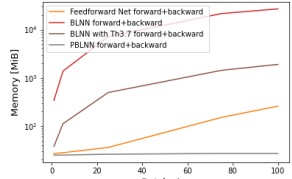

Figure 2: Comparison of the time (left) and space (right) complexity for a single iteration between a traditional feedforward network and various BLNN variants.

floating point number operations (FLOPs) and memory for a single iteration between a traditional feedforward network and various BLNN variants: BLNN (with brute force backpropagation), BLNN with Theorem 3.7, PBLNN with only one variable constrained to be bi-Lipschitz. Comparing our model with Theorem 3.7 to a traditional feed-forward neural network, we can conclude that their difference is only a factor of order 10. Theorem 3.7 greatly contributes to reducing both computational and space requirements (improvement of order of 10 and $10^2$, respectively). This explains the scalability of our model to large datasets. Finally, the PBLNN clearly decreases the complexity of the model by limiting the number of variables we impose bi-Lipschitzness. See Appendix H.1 for details on the experimental setup.

### 3.9 Discussion

Our bi-Lipschitz model BLNN (and PBLNN) possesses interesting properties. First, it provides by design a theoretical bound of the bi-Lipschitz constants. Some methods cannot necessarily afford this. Furthermore, it enables a direct parameterization of the overall bi-Lipschitzness through the addition of two regularization terms with respective coefficients $\alpha$ and $\beta$ at different outputs. This translation of bi-Lipschitzness into strong convexity plays an indispensable role as we do not disturb the formulation of the core function, do not track the transformation of the input down to the output and only have two hyper-parameters to monitor, the strict minimum. As a result, we can create bi-Lipschitz functions with known expressive power and more complex than a single layer which others had to compose many times to create a deep bi-Lipschitz neural network. We also avoid the risk of losing tightness in the bounds of the bi-Lipschitz constants caused by this kind of layer-wise parameterization. Therefore, our model satisfies the desired points of Subsection 2.2.

While the PBLNN solves several issues of the BLNN, optimization inside the forward pass is still an expensive procedure. Nevertheless, there are plenty of approaches to accelerate the LFT and backpropagation through approximations as well. Since the objective function of the LFT is always strongly concave, its convergence speed does not depend on the dimension, and the only bottleneck is the computation of the gradient information of the objective function (i.e., the ICNN). Approximations by zeroth-order methods can improve the procedure, for example. Moreover, amortizing the LFT as Amos (2023) did may also be a promising simplification of the forward pass. On the other hand, backpropagation through the whole forward pass can also be simplified based on Theorem 3.7. For instance, the Hessian matrix can be replaced by its diagonal or upper triangular elements, making the inversion operation easier. Other improvements, including those for the backward pass, can be adapted from research on DEQs (Bai et al., 2019).

Another limitation, which is overcome by the PBLNN, is that the BLNN cannot represent a function that is not the gradient of a convex function. However, such a type of function is the core of machine learning problems related to optimal transport (Santambrogio, 2015) and some physical systems (Huang et al., 2021). In the following chapter, we will show that our model can be applied to various problems and its performance is not overshadowed by these limitations. In short, it is a necessary price to pay to gain a bi-Lipschitz model with features such as known expressive power and high controllability so that it can outperform other methods with looser constraint but lower controllability and fewer guarantees. Indeed, if we define the *unit* of a bi-Lipschitz model as the basic architecture that requires the minimal number of hyperparameters, i.e., one for Lipschitzness and one for inverse-Lipschitzness, most existing models are limited to constructing simple (bi-)Lipschitz units with low expressive power (e.g., only linear) and they have to compose those units to achieve higher expressive power, which leads to looser bounds. In that sense, our model still has a higher expressive

Table 1: Tightness of Lipschitz bound when fitting $f(x) = x\ (x < 0),\ x + 1\ (x \geq 0)$. Mean over five trials. See Table 5 for further results.

| Models | $L = 5$ | $L = 10$ | $L = 50$ |
|---|---|---|---|
| SN | 80.1% | 68.9 % | 32.5% |
| Orthogonal | 75.1% | 48.9% | 14.6% |
| AOL | 65.9% | 46.5% | 15.4% |
| SLL | 77.6% | 50.0% | 15.7% |
| Sandwich | 84.3% | 60.2% | 16.4% |
| LMN | **100.0%** | 98.5% | 26.0% |
| BiLipNet | 98.0% | 58.9% | 6.8% |
| Ours | **100.0%** | **99.4%** | **99.9%** |

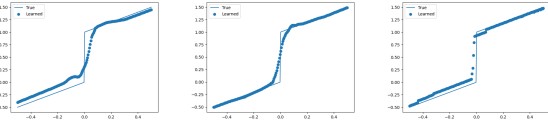

Figure 3: Results of fitting $f(x) = x\ (x < 0),\ x + 1\ (x \geq 0)$ with SLL (left), Sandwich (middle) and our method (right) with a specified Lipschitzness of 50. See Figure 16 for further details.

power and tighter bounds than other (bi-)Lipschitz units as ours can *with only one unit* produce complex functions and its parameterization is not layer wise. Now, if we can afford to sacrifice the tightness of the bounds to further increase expressive power, we can proceed like other methods by stacking multiple BLNNs or combining them with other architectures to suit the characteristics of the problem at hand (see Appendix F).

Further substantial extensions of our method can be found in Appendix F, such as generalization to different input and output dimensions, to non-homogeneous functions and to other norms.

## 4 Experiments

Experimental setups can be found in Appendix H and codes in `https://github.com/yuri-k111/Bi-Lipschitz-Neural-Network`. Further detailed results are summarized in Appendix G.

### 4.1 Bi-Lipschitz Control

In this subsection, the goal is to empirically verify that (i) our model achieves a tight control of bi-Lipschitz constants, and (ii) it provides new beneficial behaviors different from other methods. We focus on simple problems as they effectively convey key ideas.

**Tightness of the Bounds** Here, we verify the quality of the bi-Lipschitz bounds when the model undergoes training. Especially, we focus on the Lipschitz bound since it is the most affected by the approximation of LFT, and there exist many works to compare with it. Inspired by the experiment of Wang and Manchester (2023), we aim to learn the function $f(x) = x\ (x < 0),\ x + 1\ (x \geq 0)$ that has a discontinuity at $x = 0$. We take as comparison the LMN (Nolte et al., 2023), the BiLipNet (Wang et al., 2024) and the i-ResNet network represented by its substructures: spectral normalization (SN) (Miyato et al., 2018), AOL (Prach and Lampert, 2022), Orthogonal (Trockman and Kolter, 2021), SLL (Araujo et al., 2023), Sandwich (Wang and Manchester, 2023). The Lipschitzness of each model is constrained by a constant $L$. A model with a tight Lipschitz bound should achieve that upper bound around $x = 0$ in order to reproduce the behavior of $f$. The percentage between the empirical Lipschitz constant and the imposed upper bound $L$ can be found in Table 1. Interestingly, our method is the only one that achieves an almost perfect score for all settings, while for others the tightness is decreasing. This can be understood as the result of the direct control of bi-Lipschitzness without relying on the information of the individual layers and the construction which uses only direct parameterizations. See Figure 3 for a visualization and Appendix G.2 for more results.

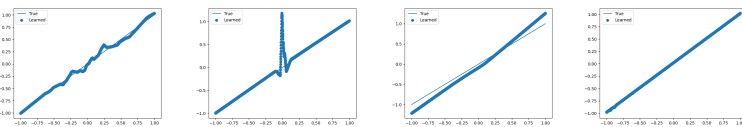

Figure 4: Results of fitting the linear function $y = x$ with (from left) AOL, Sandwich, BiLipNet and our method with a specified Lipschitzness of 1000. See Figures 17 and 18 for further results.

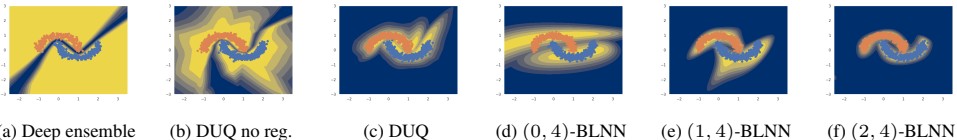

| (a) Deep ensemble | (b) DUQ no reg. | (c) DUQ | (d) $(0, 4)$-BLNN | (e) $(1, 4)$-BLNN | (f) $(2, 4)$-BLNN |

Figure 5: Uncertainty estimation with the two moons data set with several models. Blue indicates high uncertainty, and yellow low uncertainty. (d)-(f) are with DUQ+BLNN, where $(\alpha, \beta)$ are clarified.

**Flexibility of the Model** If we can underestimate the Lipschitz constant of the target function as the previous experiment, we may also overestimate it. In this case, we observe that previous methods are influenced by the imposed Lipschitz bound presenting high fluctuations in the learned representation or slower learning speed. This is illustrated when learning the identity function $y = x$ with a Lipschitz constraint of $L = 1000$ in Figure 4. Our model can learn without any problem and the learning speed is hardly affected by $L$. This may be due to the way we control the Lipschitz constant of the model. In layer-wise models, the Lipschitz constant is adjusted by scaling the input, while in ours we add a regularization term at the end, resulting in different loss landscapes with seemingly better regularization performance for the latter strategy. See Appendix G.3 for more results.

## 4.2 Uncertainty Estimation

We now illustrate that our model can be efficiently used in areas where bi-Lipschitzness already plays an essential role as an inductive bias. In the estimation of uncertainty based on a single neural network, out-of-distributions and in-distributions points may overlap in the feature space making them indistinguishable. Imposing inverse Lipschitzness on the underlying neural network of the architecture is thus important as it can avoid this phenomenon called *feature collapse*. Lipschitzness is also beneficial to improve the generalization performance. In one of the state-of-the-art approaches called deterministic uncertainty quantification (DUQ) from (Van Amersfoort et al., 2020), bi-Lipschitzness is constrained through a regularization method. Therefore, we can replace the gradient penalty with a hard restriction by changing the neural network to a BLNN, resulting in a method we call here DUQ+BLNN. See Appendix E for a precise formulation of this method and theme.

**Two moons** The first experiment we lead is with the two moons dataset. Here, we compare DUQ+BLNN with the deep ensembles method (Lakshminarayanan et al., 2017), DUQ and a DUQ with no bi-Lipschitz regularization (DUQ with no reg.). Results are plotted in Figure 5. The ideal scenario is for the boundary of the yellow area to be as close as possible to the training data. As we can observe, by adding a bi-Lipschitz constraint, the area of certainty is decreased around the training data. DUQ+BLNN with $(\alpha, \beta) = (2, 4)$ achieves a tighter area than DUQ. We chose the value of $\alpha$ and $\beta$ based on a grid search. We took the hyper-parameters with the highest accuracy and $\alpha$ since a higher $\alpha$ is expected to create a tighter yellow area as shown in Figure 5. Note this strategy only relies on the training data, and such strategy is possible thanks to our direct and simple parameterization of the bi-Lipschitz constants. This clearly shows the advantage of the unique features of our model. See Appendix G.6.1 for further results and discussion on this experiment.

**Fashion-MNIST** Next, we use real world data of FashionMNIST (Xiao et al., 2017), MNIST (LeCun and Cortes, 2010) and NotMNIST (Bulatov, 2011). The task is to learn to classify FashionMNIST, but at the end of training we verify whether the uncertainty of the model significantly increases when other types of data such as MNIST or NotMNIST is given to the model. This task to distinguish FashionMNIST from MNIST datasets is known to be a complicated task (Van Amersfoort et al., 2020). We compute the AUROC for the detection performance. The result is shown in Table 2. Our

Table 2: Out-of-distribution detection task of FashionMNIST vs MNIST and FashionMNIST vs NotMNIST with DUQ and DUQ+$(0,3)$-BLNN. Means over five trials.

| Models | Accuracy | BCE | AUROC MNIST | AUROC NotMNIST |
|---|---|---|---|---|
| DUQ | 0.889 | 0.064 | 0.862 | 0.900 |
| DUQ+BLNN | **0.899** | **0.060** | **0.896** | **0.964** |

Table 3: Comparison of our model with state-of-the-art monotone models in benchmark datasets. Means over three trials. Results of LMN and SMNN are from the original papers. BF = BlogFeedBack, LD = LoanDefaulter, HD = HeartDisease, Acc. = accuracy. See Table 9 for complete results.

| Models | COMPAS (Acc.) | BF (RMSE) | LD (Acc.) | HD (Acc.) | AutoMPG (MSE) |
|---|---|---|---|---|---|
| LMN | 69.3 % | 0.160 | 65.4 % | 89.6 % | 7.58 |
| SMNN | 69.3 % | **0.150** | 65.0 % | 88.0 % | 7.44 |
| Ours | **69.4 %** | 0.157 | **65.5 %** | **90.2 %** | **7.13** |

model achieves not only higher performance for FashionMNIST but also better detection of MNIST and NotMNIST dataset. See Appendix G.6.2 for further results and discussion on this experiment.[1]

### 4.3 Partially Monotone Settings

Sometimes, it happens that we have preliminary knowledge on some type of monotone behaviors of the dataset (Nolte et al., 2023). For example, COMPAS (Angwin et al., 2016), BlogFeedBack LoanDefaulter (Nolte et al., 2023), HeartDisease (Janosi et al., 1988) and AutoMPG (Quinlan, 1993) are benchmark datasets that possess a monotone inductive bias on some variables. See Nolte et al. (2023) for further details on the dataset. As a result, it is more efficient to tune the architecture of the trained model so that it successfully reflects this inductive bias, and various models have been proposed to address this challenge. This is another field where we can apply our architecture that can create monotone (or inverse Lipschitz) functions. We can also control the Lipschitzness to improve the generalization performance. We compare our PBLNN with two state-of-the-art methods: LMN (Nolte et al., 2023) and SMNN (Kim and Lee, 2024) in Table 3, and with other models in Table 9 as well. As we can observe, our model is competitive with the others.

**Generalization and Scalability Test**   Furthermore, we used the dataset provided by Nolte et al. (2023), CIFAR101, which is a slight augmentation of the original CIFAR100 dataset and designed to exhibit a monotone behavior with respect to one variable. We adopted their training scheme, utilizing the entire dataset for training and intentionally overfitting the data in order to assess both the scalability and expressive power of the model. Successfully, we achieved a 0 loss and 100% accuracy for this experiment, and the convergence was faster than that of Nolte et al. (2023). This illustrates the high scalability and expressive power of the PBLNN.

## 5   Conclusion

We built a model called BLNN based on convex neural networks and the LFT so that bi-Lipschitz functions more complex than a single layer can be constructed and its bi-Lipschitz constants manipulated through the coefficient of two regularization terms added at different outputs. That way, BLNN not only achieves such a tight, direct and simple control but also provides rigorous straightforward analysis on the expressive power and on approximations involved in practice. We illustrated with experiments its distinctive advantageous features compared to prior models. While the primary focus of this paper was to establish a framework suited for solid analyses and for the practical control of bi-Lipschitzness, it is, of course, essential to complement this effort with more varied machine learning applications in future work. We still believe this work on its own contributes to the further exploitation of bi-Lipschitzness in prospective fields and to the deeper understanding of neural networks by delivering a model with unique features for the control of this central inductive bias.

---

[1]We also tried to combine DUQ with BiLipNet but could not find competitive parameter settings.

## Acknowledgments and Disclosure of Funding

Y.K. was partially supported by Grant-in-Aid for JSPS Fellows Grant Number JP24KJ0862 and JST BOOST Japan Grant Number JPMJBS2418. T.T. was supported by RIKEN Center for Brain Science, JST CREST program JPMJCR23N2 and RIKEN TRIP initiative (RIKEN Quantum). We also thank anonymous reviewers for their valuable feedback.

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

# A   Lipschitz and Inverse Lipschitz models

In this appendix, we review existing architectures that aim to control the Lipschitzness and inverse Lipschitzness separately as existing bi-Lipschitz models borrow a lot of ideas from prior works on Lipschitzness and inverse Lipschitzness. Moreover, this appendix may constitute a reference for future bi-Lipschitz architectures.

Let us start by the Lipschitz property which has been extensively researched over the past few years.

## A.1   Lipschitzness

### A.1.1   Definition

**Definition A.1** (Lipschitzness)**.** *Let $L > 0$. $f : \mathbb{R}^l \to \mathbb{R}^t$ is $L$-Lipschitz (continuous) if $\|f(x) - f(y)\| \leq L\|x - y\|$ holds for all $x, y \in \mathbb{R}^l$. $L$ is called the Lipschitz constant and denoted as $Lip(f)$.*

As we can observe, this definition represents the concept of maximal sensitivity with respect to the input as changes in the input upper-bounds changes in the output. This is also referred to as smoothness since when the function is differentiable the Lipschitz constant upper bounds the $L_2$-norm of its Jacobian.

A convenient property which is often used in Lipschitz constrained neural networks is its closure under composition and addition. Thanks to these properties, the problem of Lipschitz regulation can be decomposed into smaller ones for neural networks constructed by the composition and addition of many simple layers.

### A.1.2   Models

Due to its omnipresence in the field of machine learning, especially in the certification of robustness against adversarial attacks, Lipschitz constrained neural networks have been the focus of many prior works. Interestingly, their realization greatly varies, delivering different solutions for the same ultimate goal.

### A.1.3   Gradient Clipping

One of the earliest works trying to control Lipschitzness in the context of machine learning proposes to clip the weights of the function to lie within a range of $[-c, c]$ after back-propagation (Arjovsky et al., 2017). Consequently, the set of possible parameters would be limited to a compact set, meaning that the overall function is guaranteed to be Lipschitz with a constant dependent on $c$. However, we cannot precisely know this value, and Miyato et al. (2018) pointed out this kind of brutal restriction favors low-rank weights, leading to a serious decrease of the expressive power of the neural networks.

**Regularization**   Another simple approach but quite effective is to add a regularization term to the loss function so that the behavior of the overall function is encouraged to become Lipschitz in a certain way. On the one hand, we can introduce a regularization term which penalizes the whole gradient of the network as Gulrajani et al. (2017) did in the context of Wasserstein GAN (Arjovsky et al., 2017) with the following term:

$$\mathbb{E}_x[(\|\nabla F(x)\|_2 - 1)^2]. \tag{1}$$

On the other hand, we can just focus on the weights. For example, Cisse et al. (2017) proposed a Lipschitz architecture called the Parseval network which uses the following regularization term:

$$\|W^\top W - I\|^2.$$

As a result, all the weight matrices are incited to become orthogonal, which means 1-Lipschitz.

The main downside of these regularization methods is that they can only impose soft constraints and cannot provide any theoretical guarantees on the Lipschitzness of the neural network. This is crucial in some cases such as adversarial attacks where sensitivity has to be exactly controlled. While the implementation is not so complicated as we only need to add a term to the loss function, the computational cost heavily depends on the formulation of the regularization term. Moreover, penalty on the whole gradient like that of equation (1) may not be truly efficient since the expectation can only be computed for a limited number of points, which means that it is unclear whether or not the regularization effect will propagate throughout the whole function.

**Spectral Normalization**   The Lipschitz constant of a linear layer is equivalent to its largest singular value since

$$\|Wx - Wy\| \leq \|W(x - y)\| \leq \|W\|\|x - y\|.$$

As a result, the problem amounts to normalizing the spectral norm of the weights. This is one of the most famous approaches in this sector. The estimation of the largest singular value is thus crucial for this technique, and power iteration is often used in practice.

Once we evaluated the spectral norm of each layer, we can for example normalize it (Miyato et al., 2018):

$$W = V/\|V\|,$$

or rescale only weights whose norm is above a certain threshold $\lambda$ (Gouk et al., 2021):

$$W = \frac{1}{\max(1, \|V\|_2/\lambda)} V.$$

This can be applied to most types of layers used so far in practice including convolutional layers which are only in essence linear operators (Farnia et al., 2019).

During training, since the weights are constantly changing, it is not realistic to exactly compute the spectral norm at each iteration. Nevertheless, it has been observed that weights evolve more slowly than the convergence of the power-iteration. As a consequence, we can only execute a few steps of the power-iteration at each step and at the end of training arrive at linear layers normalized as desired. This drastically reduces the computational cost.

Spectral normalization enables us to simplify the problem of Lipschitz control of the overall function into that of the singular value of the linear units, as the overall Lipschitz constant becomes the product of the spectral norm of each layer. However, it has been repeatedly shown that this approach considerably overestimates the true Lipschitz constant of the whole function (Fazlyab et al., 2019). For example, consider the composition of the following two matrices:

$$A = \begin{pmatrix} 100 & 0 \\ 0 & 1/100 \end{pmatrix}, \quad B = \begin{pmatrix} 1/100 & 0 \\ 0 & 100 \end{pmatrix}.$$

The spectral norm is determined by the direction of the maximal expansion of the layer but this direction may not be aligned throughout all the layers, especially when there is a nonlinear activation function between each of them. As a result, it engenders a gap between the overall maximal sensitivity and the product of that of each layer, and an intentional control becomes really difficult.

Similarly, since we are manipulating layers without taking into account the complex interaction between each other, the magnitude of the gradient may vanish (Anil et al., 2019). Indeed, when the weights are restricted to be 1-Lipschitz, the norm of the gradient can only decrease through the layers during the backward pass, leading to potential vanishing gradients (Anil et al., 2019). Notably, it was proved that this kind of 1-Lipschitz neural networks with spectral normalization and with ReLU activation functions cannot represent the absolute value $|x|$ (Anil et al., 2019) which is 1-Lipschitz.

In short, layer-wise spectral normalization risks overestimating the general Lipschitz constant, which is problematic if we want to control it, may result in the problem of vanishing gradient and introduce issues into the expressive power of the neural network.

**Orthogonalization**   The problem of vanishing gradient can be solved by restricting the matrices to be orthogonal since the eigenvalues of an orthogonal matrix always equal 1. The fact that all matrices are isotropic help to stabilize the training (Prach and Lampert, 2022). This leads to a large body of work that investigates orthogonal linear layers.

There exist several direct realizations of orthogonality. For instance, Xu et al. (2022); Huang et al. (2020) uses the form of

$$W = (VV^\top)^{-1/2}V.$$

Singla and Feizi (2021) provides a parameterization with a skew matrix $V$ as follows:

$$W = \exp\left(V - V^\top\right).$$

The latter method needs to approximate the exponential operator with a finite sum. Anil et al. (2019) also construct an orthogonal weight from matrix power series:

$$A_{k+1} = A_k \left( I + \frac{1}{2}Q_k + \ldots + (-1)^p \begin{pmatrix} -1/2 \\ p \end{pmatrix} Q_k^p \right),$$

where $Q_k = I - A_k^\top A_k$. Orthogonal convolutional layers are also studied (Li et al., 2019; Wang et al., 2020), and other realizations were considered based on Cayley transform including convolutions (Trockman and Kolter, 2021; Yu et al., 2022).

The Almost-orthogonal layer (AOL) (Prach and Lampert, 2022) reduces the computational cost of creating orthogonal layers by approximating them by the following formulation:

$$W = V \operatorname{diag} \left( \sum_j |V^\top V|_{ij} \right)^{-1/2}.$$

Empirically, these weights were nearly orthogonal at the end of the training.

In order to create $L$-Lipschitz functions, we can simply multiply the output or input by $L$. Nevertheless, while orthogonal layers are able to stabilize the training, this regularization destroys the information about the spectrum by setting all the singular values to one. Moreover, the problem of looseness of the Lipschitz bound persists since nonlinear activations are considered apart, and we do not know how they interact with the linear layers.

**Nonlinear Lipschitz Layer** So far, we have reviewed methods that deal with linear layers. Recently, some that incorporate nonlinear functions have started to appear.

Meunier et al. (2022) suggest to use residual type layers of the form

$$f(x) = x - 2/\|W\|^2 W^\top \sigma(Wx + b),$$

where $W^\top \sigma(Wx + b)$ is the derivative of $s(Wx + b)$ and $s$ is a 1-smooth convex function so that $s' = \sigma$. This type of layer is guaranteed to be 1-Lipschitz. This has been generalized by Araujo et al. (2023) who built a nonlinear layer of the form

$$f(x) = Hx + G\sigma(Wx + b).$$

If there exists a diagonal matrix $\Lambda$ with non-negative scalars that satisfies

$$\begin{pmatrix} \gamma I - H^\top H + 2\alpha\beta W^\top \Lambda W & -H^\top G - (\alpha + \beta)W^\top \Lambda \\ -G^\top H - (\alpha + \beta)\Lambda W & 2\Lambda - G^\top G \end{pmatrix} \succeq 0,$$

where $\sigma$ is slope restricted to $[\alpha, \beta]$, then $f(x)$ is $\sqrt{\gamma}$-Lipschitz. Especially, when $\gamma = 1$, $H = 1$ and $G = -2WT^{-1}$, where $T$ is a diagonal with non-negative entries so that $T \succeq W^\top W$, the corresponding layer is called SDP-based Lipschitz Layer (SLL). A choice of $T$ is

$$T_{ii} = \sum_{j=1}^{n} |W^\top W|_{ij} q_j / q_i,$$

where $q_i > 0$. See Theorem 3 of their work for further detail.

Fazlyab et al. (2019) proposes LipSDP which estimates the Lipschitz constant of a multi-layer neural network through a semi-definite programming (SDP). It is based on a SDP derived from an integral quadratic constraint. Suppose we reformulate the simple multi-layer neural network as

$$BX = \sigma(AX + b), \quad f(x) = CX + b_L,$$

where $\sigma$ is $[\alpha, \beta]$ slope-restricted, $X = (x_0^\top, x_1^\top, \cdots, x_L^\top)^\top$,

$$A = \begin{pmatrix} W_0 & 0 & \cdots & 0 & 0 \\ 0 & W_1 & \cdots & 0 & 0 \\ \vdots & \vdots & \ddots & \vdots & \vdots \\ 0 & 0 & \cdots & W_{L-1} & 0 \end{pmatrix},$$

$$B = \begin{pmatrix} 0 & I_{n_1} & 0 & \cdots & 0 \\ 0 & 0 & I_{n_2} & \cdots & 0 \\ \vdots & \vdots & \vdots & \ddots & \vdots \\ 0 & 0 & 0 & \cdots & I_{n_L} \end{pmatrix},$$

and

$$C = (0, \ldots, 0, W_L), \quad b = (b_0^\top, \ldots, b_{L-1}^\top)^\top.$$

Now, if there is a diagonal matrix $\Lambda$ with non-negative entries such that

$$\begin{pmatrix} A \\ B \end{pmatrix}^\top \begin{pmatrix} -2\alpha\beta\Lambda & (\alpha+\beta)\Lambda \\ (\alpha+\beta)\Lambda & -2\Lambda \end{pmatrix} \begin{pmatrix} A \\ B \end{pmatrix} + \begin{pmatrix} -\gamma I_{n_0} & 0 & \cdots & 0 \\ 0 & 0 & \cdots & 0 \\ \vdots & \vdots & \ddots & \vdots \\ 0 & 0 & \cdots & W_L^\top W_L \end{pmatrix} \preceq 0$$

is satisfied, then $f$ is $\sqrt{\gamma}$-Lipschitz.

Wang and Manchester (2023) provides a direct parameterization for this SDP. Its 1-layer version is called the Sandwich layer and formulated as follows:

$$f(x) = \sqrt{2}A^\top \Psi \sigma(\sqrt{2}\Psi^{-1}Bx + b),$$

where A and B are produced from the Cayley transformation of an arbitrary matrix with correct dimensions, and $\Psi$ is a diagonal matrix with positive entries.

While these methods provide new interesting possibilities to parameterize Lipschitz functions, they are mainly layer-wise, still leading to overestimation of the overall Lipschitz constant, or use the typical structure of the Lipschitzness, meaning that we cannot extend them to bi-Lispchitzness for our purpose. Notably, the SDP-based approach that comes from Fazlyab et al. (2019) cannot handle more general structures such as skip connections.

## A.2 Inverse Lipschitzness

### A.2.1 Definition

**Definition A.2** (inverse Lipschitzness). *Let $L' > 0$. $f : \mathbb{R}^l \to \mathbb{R}^t$ is $L'$-inverse Lipschitz if*

$$\|f(x) - f(y)\| \geq L'\|x - y\|$$

*holds for all $x, y \in \mathbb{R}^l$. $L'$ is called the inverse Lipschitz constant and denoted as invLip($f$).*

The above property implies that an inverse Lipschitz function is always injective, which means that it has an inverse which is $1/L'$-Lipschitz.

The inverse Lipschitzness is a mathematical description of the minimal sensitivity of the function. By increasing the inverse Lipschitz constant, we can dilute the function and make it more sensitive to small changes of the input. Unfortunately, the inverse Lipschitzness is not closed under addition as we can understand with the simple example of $0 = x - x$.

### A.2.2 Models

To the best of our knowledge, the only model that was built specially for the control of inverse Lipschitzness is that of Kinoshita et al. (2023). The same model was used earlier by Huang et al. (2021) but for the construction of normalizing flows compatible with Brenier's theorem in optimal transport.

Kinoshita et al. (2023) observed that the derivative of an $\alpha$-strongly convex function is $\alpha$-inverse Lipschitz. As a result, they propose a model of the type

$$\nabla\left(F(x) + \frac{\alpha}{2}\|x\|^2\right) = \nabla F(x) + \alpha x, \tag{2}$$

where $F$ is any convex function parameterized by a neural network. This approach has the interesting property that we do not need to know what happens between the layers as we only add a term to the output, which is possible thanks to the convexity of $F$. That is, a layer-wise control is not required, resulting in a simple and tight control of the inverse Lipschitz constant thanks to the fact that before taking the derivative $F$ can also freely reproduce any convex function that is not necessarily strongly convex. In our work, we start from this model and extend it to bi-Lipschitzness.

### A.3 Bi-Lipschitzness

### A.3.1 Definition

Finally, a function which is both Lipschitz and inverse-Lipschitz is called bi-Lipschitz. Note that by definition the inverse Lipschitz constant cannot exceed the Lipschitz constant.

**Definition A.3** (bi-Lipschitzness). *Let* $0 < L_1 \leq L_2$. *$f : \mathbb{R}^l \to \mathbb{R}^t$ is $(L_1, L_2)$-bi-Lipschitz if*

$$L_1 \|x - y\| \leq \|f(x) - f(y)\| \leq L_2 \|x - y\|$$

*holds for all $x, y \in \mathbb{R}^l$. $L_1$ and $L_2$ are called together* bi-Lispchitz *constants.*

### A.3.2 Applications

Several interpretations can be attributed to this definition. Here, we provide some examples with clearer explanations of the importance of bi-Lipschitzness which were omitted in the main paper. In these applications, both Lipschitzness and inverse Lipschitzness are useful, or sometimes indispensable, and Lipschitzness alone becomes insufficient.

1. Injectivity and Out-of-Distribution Detection: Bi-Lipschitz functions are injective thanks to inverse Lipschitzness. This helps distinguish out-of-distribution and in-distribution data in the feature space, making uncertainty estimation possible (Van Amersfoort et al., 2020). Without inverse Lipschitzness, the problem of feature collapse can occur and compromise the detection of outliers. The inverse Lipschitz constant can control the sensitivity to out-of-distribution points. Please see Appendix E for further details on this topic.

2. Quasi-Isometry and Dimensionality Reduction: Bi-Lipschitz functions can be regarded as a quasi-isometry, meaning that the structure of the input is inherited in the output as well. This property is used for adequate dimensionality reduction or embeddings (Li et al., 2020).

3. Invertibility and Solving Inverse Problems: Bi-Lipschitz functions are invertible, and the inverse Lipschitz constant serves as the Lipschitz constant of the inverse function. In that sense, imposing inverse Lipschitzness helps guarantee good properties of the inverse function, just as Lipschitzness assures them for the forward function (Behrmann et al., 2021). This aspect is used to create normalizing flows (Behrmann et al., 2019) and solve inverse problems (Kruse et al., 2021).

4. Balancing Sensitivity and Robustness: Bi-Lipschitz functions can avoid overly insensitive behavior with respect to their input by controlling the inverse Lipschitz constant. An overly invariant function is also vulnerable to adversarial attacks, as pointed out by Jacobsen et al. (2019). To the best of our knowledge, the application of bi-Lipschitzness in this direction is underexplored, but this concept may provide an effective solution.

### A.3.3 Comparison with Prior Models

In this section, we compare our model BLNN with other models in more details.

**Convex Potential Flow**    If we set $\alpha = 0$, we obtain a BLNN whose output is $\nabla F_\theta^*$, the derivative of the LFT of a $1/\beta$-strongly convex ICNN $F_\theta$ in terms of Algorithm 1. Since the BLNN is still injective, it has an inverse which is only $\nabla F_\theta$. As a consequence, in applications where we need to compute the inverse, this model can provide an interesting solution since we can create invertible functions with known inverse. This model was precisely used by Huang et al. (2021) in the context of normalizing flows. It is equivalent to that of Kinoshita et al. (2023), but the motivation is different. Ours can thus be regarded as an extension of their model as well.

**Residual Network**    Interestingly, our model ultimately takes the form of $\alpha x + g(x)$, which can be compared with a residual network. The main differences are that the skip connection is scaled by $\alpha$ and that the formulation of $g$ is restricted to derivatives of convex functions. These two features were crucial components for a direct parameterization of the bi-Lipschitz constants. Behrmann et al. (2019) composed many layers of the form of a residual network to guarantee high expressive power to their invertible residual network. This superposition of layers is only sub-optimal as it leads to a looseness in the bounds of the bi-Lipschitz constants of the overall function and it is not enough to

represent some functions such as $y = -x$. In a sense, our work shows that by restricting ourselves to the derivative of convex functions, this kind of heuristics is not necessary at all and that this condition leads to tighter control.

# B  Proofs of Statements and Further Discussion

In this appendix, we provide further details and proofs of statements in the main paper.

## B.1  Additional Definitions

We first remind some definitions.

**Definition B.1.** *Let $\gamma > 0$. $F : \mathbb{R}^m \to \mathbb{R}$ is $\gamma$-smooth if $F$ is differentiable and*

$$\|\nabla F(x) - \nabla F(y)\| \leq \gamma \|x - y\|$$

*holds for all $x, y \in \mathbb{R}^l$. That is, $\nabla F$ is $\gamma$-Lipschitz.*

In the remainder of this chapter, we will concisely refer to a *smooth* function when we do not need to specify the smoothness constant. Furthermore, in this work, we will often deal with smooth convex functions. For this type of function, there exist four equivalent characterizations. See Appendix D for the proof.

**Theorem B.2.** *Let $\gamma > 0$ and $F : \mathbb{R}^m \to \mathbb{R}$ a differentiable convex function on a convex domain. Then the following are equivalent:*

1. *$F$ is $\gamma$-smooth in the meaning of Definition B.1:*

$$\|\nabla F(x) - \nabla F(y)\| \leq \gamma \|x - y\|$$

.

2. *The following holds for any $x, y \in \text{dom} F$:*

$$(\nabla F(x) - \nabla F(y))^\top (x - y) \leq \gamma \|x - y\|^2.$$

3. *The following holds for any $x, y \in \text{dom} F$:*

$$F(y) \leq F(x) + \nabla F(x)^\top (y - x) + \frac{\gamma}{2} \|y - x\|^2.$$

4. *(co-coercivity) The following holds for any $x, y \in \text{dom} F$:*

$$(\nabla F(x) - \nabla F(y))^\top (x - y) \geq \frac{1}{\gamma} \|\nabla F(x) - \nabla F(y)\|^2.$$

Convexity and strong convexity is defined as follows:

**Definition B.3.** *$F : \mathbb{R}^m \to \mathbb{R}$ is convex if*

$$F(tx + (1 - t)y) \leq tF(x) + (1 - t)F(y)$$

*holds for all $t \in [0, 1]$ and $x, y \in \text{dom} F$.*

**Definition B.4.** *Let $\mu > 0$. $F : \mathbb{R}^m \to \mathbb{R}$ is $\mu$-strongly convex if $F(x) - \frac{\mu}{2} \|x\|^2$ is convex.*

We also introduce the Lengendre-Fenchel transformation which will be a core process in our construction.

**Definition B.5.** *Let $F : I \to \mathbb{R}$ a convex function over $I \subset \mathbb{R}^m$. Its* Legendre-Fenchel transformation *$F^*$ is defined as follows:*

$$F^*(x) := \sup_{y \in I} \{\langle y, x \rangle - F(y)\}. \tag{3}$$

## B.2 Construction of Bi-Lipschitz Functions: Proof of Theorem 3.4

The first step is to notice that the gradient of a real-valued $L$-strongly convex function $F$ becomes $L$-inverse Lipschitz as pointed out by Kinoshita et al. (2023).

**Proposition B.6.** *Let $F$ be an $\alpha$-strongly convex differentiable function. Then $\nabla F$ is $\alpha$-inverse Lipschitz.*

*Proof.* Since $F$ is $\alpha$-strongly convex,

$$F(y) \geq F(x) + \nabla F(x)^\top (y - x) + \frac{\alpha}{2}\|y - x\|^2.$$

Similarly,

$$F(x) \geq F(y) + \nabla F(y)^\top (x - y) + \frac{\alpha}{2}\|x - y\|^2.$$

By summing both inequalities side by side, we obtain

$$\alpha\|x - y\|^2 \leq (\nabla F(x) - \nabla F(y))^\top (x - y) \leq \|\nabla F(x) - \nabla F(y)\|\|y - x\|,$$

where we used Cauchy-Schwarz inequality for the right inequality. As a result,

$$\|\nabla F(x) - \nabla F(y)\| \geq \alpha\|x - y\|.$$

$\square$

Therefore, an $\alpha$-strongly convex neural network can be first built, and then its gradient calculated in order to construct a neural network which is guaranteed to be $\alpha$-inverse Lipschitz. Now, since the derivative of a smooth function is Lipschitz by definition, we can similarly proceed to construct a function which is both Lipschitz and inverse Lipschitz. That is, we aim to compose a function which is both strongly convex and smooth.

Interestingly, smoothness and strong convexity are closely related through the Legendre-Fenchel transformation.

**Proposition B.7.** *If $F$ is a closed $1/\beta$-strongly convex function. Then its Legendre-Fenchel transformation is $\beta$-smooth.*

See Zhou (2018) for a proof. Importantly, the smoothness of the Legendre-Fenchel transform does not depend on that of $F$ as long as it is strongly convex. This means that this statement holds also for strongly convex neural networks with ReLU activation functions, and consequently non-differentiable.

**Proposition B.8.** *The resulting function $F^*$ of a Legendre-Fenchel transformation is also convex as long as its domain is convex.*

*Proof.* For all $t \in [0, 1]$ and $x_1, x_2 \in \mathrm{dom}F^*$,

$$
\begin{aligned}
F^*(tx_1 + (1-t)x_2) &= \sup_{y \in I}\left\{\langle y, tx_1 + (1-t)x_2\rangle - F(y)\right\}\\
&= \sup_{y \in I}\left\{t\left(\langle y, x_1\rangle - F(y)\right) + (1-t)\left(\langle y, x_2\rangle - F(y)\right)\right\}\\
&\leq t\sup_{y \in I}\left\{\langle y, x_1\rangle - F(y)\right\} + (1-t)\sup_{y \in I}\left\{\langle y, x_2\rangle - F(y)\right\}\\
&= tF^*(x_1) + (1-t)F^*(x_2).
\end{aligned}
$$

$\square$

This leads to the following statement.

**Theorem B.9.** *Let $F$ be a closed $1/\beta$-strongly convex function and $\alpha \geq 0$. Then the following function is $\alpha$-strongly convex and $\alpha + \beta$-smooth:*

$$\bar{F}^*(x) = \sup_{y \in I}\left\{\langle y, x\rangle - F(y)\right\} + \frac{\alpha}{2}\|x\|^2.$$

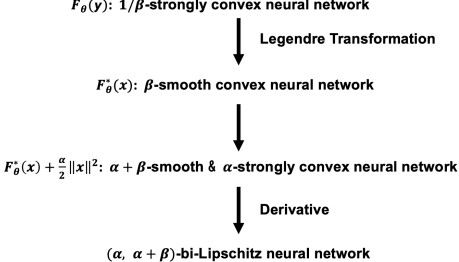

Figure 6: Construction flow of bi-Lipschitz neural network through Legendre-Fenchel transformation and Brenier map.

*Proof.* Since $F$ is a closed $1/\beta$-strongly convex function, we know that $F^*$ is convex and $\beta$-smooth. Now clearly $F^* + \frac{\alpha}{2}\|x\|^2$ is $\alpha$-strongly convex by definition of strong convexity. Moreover, since

$$\left\| \nabla \left( F^*(x) + \frac{\alpha}{2}\|x\|^2 \right) - \nabla \left( F^*(y) + \frac{\alpha}{2}\|y\|^2 \right) \right\| = \|\nabla F^*(x) + \alpha x - (\nabla F^*(y) + \alpha y)\|$$
$$\leq \|\nabla F^*(x) - \nabla F^*(y)\| + \|\alpha x - \alpha y\|$$
$$\leq (\beta + \alpha)\|x - y\|,$$

$F^* + \frac{\alpha}{2}\|x\|^2$ is $\alpha + \beta$-smooth. $\qquad\square$

**Corollary B.10.** *Suppose $\bar{F}^*$ is constructed as Theorem B.9, then $\nabla \bar{F}^*$ is $(\alpha, \alpha + \beta)$-bi-Lipschitz, and $f^* := \nabla \bar{F}^*(x) = \text{argmax}_y \{\langle y, x \rangle - F(y)\} + \alpha x$.*

See Zhou (2018) for a proof of $\nabla F^*(x) = \text{argmax}_y \{\langle y, x \rangle - F(y)\}$. See Figure 6 for a summary of the construction flow.

**Remark B.11.** *This construction can only handle functions with the same input and output dimensions. However, this is a common problem in bi-Lipschitz architectures such as that of Behrmann et al. (2019). It can be addressed by using a composition of bi-Lipschitz functions which will be explained in Appendix F.*

### B.3 Expressive Power of BLNN

In this subsection, we discuss the expressive power of BLNN. It is crucial for this type of construction to clarify this. As we mentioned earlier, layer-wise constraints realize a bi-Lipschitz neural network by restricting the sensitivity of each layer. Nevertheless, this kind of construction is only sub-optimal as it limits the expressive power of the function more than expected. For instance, neural networks with spectral normalization and ReLU activation functions cannot represent $y = |x|$ (Anil et al., 2019), and an invertible residual network cannot express the linear function $y = -x$ (Zhang et al., 2020), which were not mentioned in the original paper (Behrmann et al., 2019). Layer-wise bi-Lipschitz approaches cannot directly inherit the proofs and properties of the original network, which makes the understanding of their behavior more difficult.

As for our model, the $(\alpha, \beta)$-BLNN, we can prove that

1. Before taking the gradient, our model can approximate any $\alpha$-strongly convex and $\alpha + \beta$-smooth functions on a compact domain endowed with the sup norm (Theorem 3.6).

2. After taking the gradient, we can create a sequence of functions that converges point-wise to any function which is $\alpha + \beta$-Lipschitz, $\alpha$-strongly monotone (see Definition B.12) and the derivative of a real-valued function.

The proof is almost straightforward thanks to the fact that our model only makes modification at the output of the core neural network, and we can build on the proofs of previous works (Chen et al., 2019; Huggins et al., 2018).

**Definition B.12.** *A* monotone *function $f : \mathbb{R}^l \to \mathbb{R}^t$ is defined as $\langle f(x) - f(y), x - y \rangle \geq 0$. Furthermore, $f$ is $\alpha$-strongly monotone if $\langle f(x) - f(y), x - y \rangle \geq \alpha\|x - y\|^2$.*

### B.3.1 First Part: Proof of Theorem 3.6

**Theorem B.13.** *For any proper closed $\alpha$-strongly convex and $\alpha + \beta$-smooth function on a compact domain, there is a BLNN without taking the gradient at the end, i.e., $\sup_y \{\langle y, x \rangle - (G_\theta(y) + \|y\|^2/(2\beta))\} + \alpha \|x\|^2/2$ where $G_\theta$ is an ICNN with ReLU or softplus-type activation function, that approximates it within $\epsilon$ in terms of the sup norm.*

*Proof.* From Proposition 3 of Huggins et al. (2018), we already know that for any convex function on a compact domain, there is an ICNN with ReLU or softplus-type activation function that approximates it within $\epsilon$ in terms of the sup norm.[2] The key point in this proof is to show that the Legendre-Fenchel transformation does not deteriorate the approximation quality and that introducing strongly convexity and smoothness constraints does not cause any unexpected limitations on the expressive power.

Let $f$ be a proper closed $\alpha$-strongly convex and $\alpha + \beta$-smooth function. Since $f(x)$ is $\alpha$-strongly convex and $\alpha + \beta$-smooth, $f(x) - \frac{\alpha}{2}\|x\|^2$ becomes a convex $\beta$-smooth function. Indeed, by definition of strong convexity, $f(x) - \frac{\alpha}{2}\|x\|^2$ is convex. Moreover,

$$\langle \nabla f(x) - \alpha x - (\nabla f(y) - \alpha y), x - y \rangle \le \beta \|x - y\|^2.$$

As a result, by definition of smoothness (Theorem B.2), $f(x) - \frac{\alpha}{2}\|x\|^2$ is $\beta$-smooth. Now, since $\tilde{f}(x) := f(x) - \frac{\alpha}{2}\|x\|^2$ is a proper closed convex function, $\tilde{f}^{**} = \tilde{f}$ (Theorem 12.2 of Rockafellar (1997)). That is, the Legendre-Fenchel transforms is an involution. Taking $\tilde{f}^*$, we obtain a $1/\beta$-strongly convex function. Similarly, $\tilde{f}^*(y) - \frac{1}{2\beta}\|y\|^2$ is convex. Therefore, there exists an ICNN $\hat{f}$ so that

$$\sup_y \left| \hat{f}(y) - \left( \tilde{f}^*(y) - \frac{1}{2\beta}\|y\|^2 \right) \right| < \epsilon,$$

or

$$\sup_y \left| \hat{f}(y) + \frac{1}{2\beta}\|y\|^2 - \tilde{f}^*(y) \right| < \epsilon. \tag{4}$$

This precision is preserved after Legendre-Fenchel transformation. Indeed, if we define $\hat{f}^*_\beta$ as the Legendre-Fenchel transformation of $\hat{f}(y) + \frac{1}{2\beta}\|y\|^2$, then

$$\sup_x |\hat{f}^*_\beta(x) - \tilde{f}^{**}(x)| < \epsilon \tag{5}$$

which is equivalent to

$$\sup_x |\hat{f}^*_\beta(x) - \tilde{f}(x)| < \epsilon.$$

This can be derived as follows:

$$\hat{f}^*_\beta(x) = \sup_y \left\{ \langle y, x \rangle - \left( \hat{f}(y) + \frac{1}{2\beta}\|y\|^2 \right) \right\}$$

$$= \sup_y \left\{ \langle y, x \rangle - \tilde{f}^*(x) + \tilde{f}^*(x) - \left( \hat{f}(y) + \frac{1}{2\beta}\|y\|^2 \right) \right\}$$

$$\le \sup_y \left\{ \langle y, x \rangle - \tilde{f}^*(x) \right\} + \sup_y \left\{ \tilde{f}^*(x) - \left( \hat{f}(y) + \frac{1}{2\beta}\|y\|^2 \right) \right\}$$

$$\le \tilde{f}^{**}(x) + \epsilon,$$

where we used the definition of $f^{**}(x)$ and equation (4) for the last inequality. Since we can change the role of $\hat{f}$ and $\tilde{f}^*$ and the above inequality holds for all $x$, we obtain inequality (5). Finally, by adding $\alpha \|x\|^2/2$ to $\hat{f}^*_\beta(x)$, we obtain a function that approximates $f$ within $\epsilon$

$$\sup_x |\hat{f}^*_\beta(x) + \alpha \|x\|^2/2 - f(x)| = \sup_x |\hat{f}^*_\beta(x) - (f(x) - \alpha \|x\|^2/2)|$$

$$= \sup_x |\hat{f}^*_\beta(x) - \tilde{f}(x)|$$

$$\le \epsilon.$$

---

[2] Here, a softplus-type activation function is a function $s$ that satisfies the following conditions: $s \ge \mathrm{ReLU}$, $s$ is convex and $|s(x) - \mathrm{ReLU}(x)| \to 0$ as $|x| \to \infty$ (Chen et al., 2019).

In order words, $\hat{f}$ is the ICNN that creates a BLNN (before derivation) that approximates $f$ within $\epsilon$. $\hfill\square$

As we can conclude, our model with parameters $\alpha$ and $\beta$ (before taking the gradient) can approximate any $\alpha$-strongly convex and $\alpha + \beta$-smooth functions on a compact domain endowed with the sup norm. Importantly, the approximation quality of the BLNN equals that of the model we use to create the core convex function.

### B.3.2 Second Part

The second point of this section is proved by the following theorem.

**Theorem B.14** (Huggins et al. (2018), Theorem 2 adapted). *Suppose $G : \mathbb{R}^d \to \mathbb{R}$ a proper convex function and almost everywhere differentiable. If there is a sequence $F_n : \mathbb{R}^d \to \mathbb{R}$ of BLNN before taking the derivative so that $F_n \to G$. Then, for almost every $x \in \mathbb{R}^d$, $\nabla F_n(x) \to \nabla G(x)$.*

If a monotone function can be represented as the derivative of the real-valued function, then it is the gradient of a convex function. Hence, we can use the above theorem.

**Remark B.15.** *The existence of such sequence $F_n$ is assured by Theorem B.13 since we can let $F_n$ approximate $G$ with a uniform error of $1/n$ on the compact domain $[-n, n]^d$.*

Therefore, an $(\alpha, \beta)$-BLNN can represent any $\alpha + \beta$-Lipschitz $\alpha$-strongly monotone (i.e., $(\alpha, \alpha + \beta)$-bi-Lipschitz) function that is the derivative of another function almost everywhere.

### B.3.3 A Brief Discussion on the Expressive Power of BLNN

In this section, we briefly discuss the difference between the whole set of bi-Lipschitz functions and the expressive power of the BLNN and the possibility of designing an architecture that can approximate any bi-Lipschitz functions based on our model.

There are three relevant classes of functions: (1) bi-Lipschitz functions, (2) monotone bi-Lipschitz functions, and (3) cyclically monotone bi-Lipschitz functions. Our model can represent any function of class (3) since cyclically monotone bi-Lipschitz functions are equivalent to the class of derivatives of (strongly) convex functions (Rockafellar, 1970). In dimensions larger than 1, these three classes are different: $f(x, y) = (-y, x)$ is in (1) but not in (2), and $f(x, y) = (x + 2y, y)$ is in (2) but not in (3). Current bi-Lipschitz models are supposedly in class (1) or (2), like that of Behrmann et al. (2019) and Nolte et al. (2023). Here, we will only focus on bi-Lipschitz functions with the same input and output dimensions, as most of the bi-Lipschitz models fall into this category. Let us now discuss how class (3) can be used to produce functions of class (1) (and (2)).

First, if we suppose the function can be represented by the gradient of another real-valued function, class (1) and (3) are equivalent. Interestingly, under this condition, we can provide an even stronger statement based on Theorem 2.1 of Zlobec (2005). It says that If $f : \mathbb{R}^n \to \mathbb{R}$ is continuously differentiable and its derivative is Lipschitz on a convex set $K$ with some Lipschitz constant $L$, then there are a convex function $C(x)$ on $K$ and $a \geq L$ so that $f(x) = C(x) - ax^\top x/2$. In other words, under the condition that a function can be written as the gradient of another real-valued function, class (3) is equivalent to all Lipschitz functions up to a quadratic concave term. Therefore, our model can still express a quite large family of functions.

Next, the condition of being the gradient of a function can be interpreted as being rotation-free, i.e., $\nabla \times F = 0$. It is known that any vector field can be decomposed into a rotation-free and divergence-free component (Hodge decomposition theorem) under some regularity conditions. In $\mathbb{R}^3$, this means that a function f in class (1) can be decomposed as follows:

$$f = \nabla \times A + \nabla B$$

where $A$ is a vector field, and $B$ a scalar function. From the other direction, we can take $\nabla B$ as a function of class (3), and $A$ as an arbitrary function with bounded gradient. That way, we can reproduce a large variety of functions so that $f$ is bi-Lipschitz without necessarily being representable as the gradient of another function. The choice of $A$ is still ambiguous so that $f$ is effectively

bi-Lipschitz with explicit bi-Lipschitz parameter control, but we believe this approach is promising for the generalization of our method.

In conclusion, while our model may have theoretically restrictive expressive power, it still leaves a large liberty to express more general bi-Lipschitz functions both in theory and practice.

### B.4 Backward Pass of BLNN

In this subsection, we provide the explicit formulation of the gradient of the BLNN with respect to the parameters. The first half considers the simple BLNN and the second a more complete case analysis with a complex architecture where BLNN is only one component of it.

#### B.4.1 Proof of Theorem 3.7

First of all, the gradient of the loss with respect to the parameters can be more concretely formulated as follows:

**Lemma B.16.** *Suppose a loss function* $L : z \mapsto L(z)$, *and the output of the BLNN is* $f^*(x;\theta) := \nabla_x F_\theta^*(x) + \alpha x$ *as defined in Algorithm 1. Then the gradient can be expressed as follows:*

$$\nabla_\theta^\top L(f^*(x;\theta)) = \nabla_z^\top L(z) \left\{ \nabla_x^\top \nabla_\theta F^*(x;\theta) \right\}^\top, \tag{6}$$

*where* $z = f^*(x;\theta)$.

This is only an application of the chain rule. Now, interestingly, $\nabla_\theta F^*(x;\theta)$ can be written as a function of the core ICNN $F$ as the following statements shows.

**Theorem B.17.** *Suppose* $F$ *and* $F^*$ *are both differentiable, then the following holds:*

$$\nabla_\theta F^*(x;\theta) = -\partial_\theta F(y^*(x;\theta);\theta), \tag{7}$$

*where* $\partial_\theta$ *is the partial derivative with respect to* $\theta$.

*Proof.* Since, $y^*(x;\theta) = \mathrm{argmax}_{y \in I} \left\{ \langle y, x \rangle - F(y;\theta) \right\}$, $y^*$ satisfies the following stationary point condition:

$$\nabla_y \left\{ \langle y^*(x;\theta), x \rangle - F(y^*(x;\theta);\theta) \right\} = x - \nabla_y F(y^*(x;\theta);\theta) = 0. \tag{8}$$

Now, taking the derivative of $F^*(x;\theta) = \langle y^*(x;\theta), x \rangle - F(y^*(x;\theta);\theta)$ with respect to $\theta$ leads to:

$$\begin{aligned}
\nabla_\theta F^*(x;\theta) &= \nabla_\theta \left\{ \langle y^*(x;\theta), x \rangle - F(y^*(x;\theta);\theta) \right\} \\
&= (\nabla_\theta^\top y^*(x;\theta))^\top x - \partial_\theta F(y^*(x;\theta);\theta) - (\nabla_\theta^\top y^*(x;\theta))^\top \nabla_y F(y^*(x;\theta);\theta) \\
&= (\nabla_\theta^\top y^*(x;\theta))^\top x - \partial_\theta F(y^*(x;\theta);\theta) - (\nabla_\theta^\top y^*(x;\theta))^\top x \\
&= -\partial_\theta F(y^*(x;\theta);\theta),
\end{aligned}$$

where we used equation (8) for the last equality. □

This results in the following representation of the gradient (Theorem 3.7).

**Theorem B.18.** *Suppose a loss function* $L : z \mapsto L(z)$, *and the output of the BLNN is* $f^*(x;\theta) := \nabla F_\theta^*(x) + \alpha x$ *as defined in Algorithm 1. If* $F$ *and* $F^*$ *are both differentiable, then the gradient can be expressed as follows:*

$$\nabla_\theta^\top L(f^*(x;\theta)) = -\nabla_z^\top L(z) \left\{ \nabla_y^2 F(y^*(x;\theta);\theta) \right\}^{-1} \partial_\theta^\top \nabla_y F(y^*(x;\theta);\theta). \tag{9}$$

*Proof.* From Lemma B.16 and Theorem B.17, we know that

$$\begin{aligned}
\nabla_\theta^\top L(f^*(x;\theta)) &= \nabla_z^\top L(z) \left\{ \nabla_x^\top \nabla_\theta F^*(x;\theta) \right\}^\top \\
&= -\nabla_z^\top L(z) \partial_\theta^\top \nabla_x F(y^*(x;\theta);\theta).
\end{aligned}$$

Continuing the procedure, we obtain:

$$\begin{aligned}
\nabla_\theta^\top L(f^*(x;\theta)) &= -\nabla_z^\top L(z) \partial_\theta^\top \nabla_x F(y^*(x;\theta);\theta) \\
&= -\nabla_z^\top L(z) \nabla_x y^*(x;\theta) \partial_\theta^\top \nabla_y F(y^*(x;\theta);\theta) \\
&= -\nabla_z^\top L(z) \left\{ \nabla_y^2 F(y^*(x;\theta);\theta) \right\}^{-1} \partial_\theta^\top \nabla_y F(y^*(x;\theta);\theta),
\end{aligned}$$

where in the last equality, we used the fact that $y_\theta^* = \nabla_x F_\theta^*$ and $\nabla_y F_\theta$ are inverse with respect to each other (Zhou, 2018). □

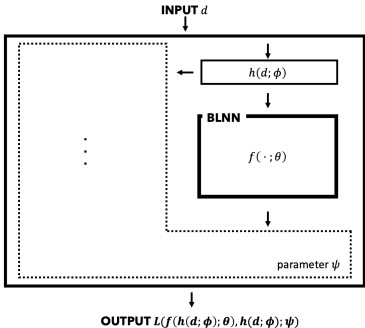

Figure 7: A generalization of an architecture including our model.

Therefore, we can compute the gradient without back-propagating through the whole optimization process of the Legendre-Fenchel transformation but only with the information of the ICNN $F_\theta$. In practice, the following observation facilitates even more coding since we can directly employ the backward algorithm provided in libraries such as Pytorch.

**Corollary B.19.** *Under the assumptions of Thoerem B.18, the gradient of the loss $L$ is equivalent to the following real-valued function:*

$$-\nabla_z^\top L(f^*(x; \theta.\text{requires\_grad\_(F)})) \left\{\nabla_x^2 F(y_\theta^*(x); \theta.\text{requires\_grad\_(F)})\right\}^{-1} \nabla_y F(y_\theta^*(x); \theta),$$

*where $\theta.requires\_grad\_(F)$ indicates that this $\theta$ is considered as a constant and not a parameter to be differentiated with respect to.*

### B.4.2   A more complete case analysis

In practice, a model may possess a more complex architecture where parameters to be optimized do not only come from the ICNN but also from other components. Consider the schema of Figure 7. Mathematically, this can be formulated as follows. Suppose $\theta$, $\phi$ and $\psi$ are parameters, $f(x; \theta)$ is a BLNN, $h(d; \phi)$ is a parameterized function like a neural network, and $L(z, w; \psi)$ is a real-valued function returning the loss. In this problem setting, the overall loss can be expressed as $L(f(h(d; \phi); \theta), h(d; \phi); \psi)$. This means, for a model that incorporates a BLNN, we can classify parameters into three groups:

1. those that define the ICNN used in BLNN,

2. those that define the transformation whose output is transferred to the input of the BLNN,

3. and those that are unrelated to the BLNN.

For instance, this formulation includes VAEs with a BLNN for the underlying neural network of the decoder. The derivative of the first group of parameters was already discussed above, and that of the last one is straightforward. As for the second family, we can proceed in a similar way as Theorem B.18.

**Theorem B.20.** *Suppose a model with loss included $L(f^*(h(d; \phi); \theta); \psi)$. The output of the BLNN is $f^*(x; \theta) := \nabla F_\theta^*(x) + \alpha x$ as defined in Algorithm 1. If $F$ and $F^*$ are both differentiable, then the gradient of $\phi$ can be expressed as follows:*

$$\nabla_\phi L(f^*(h(d; \phi); \theta), h(d; \phi); \psi) = \nabla_z^\top L(z, x; \psi) \left[\left\{\nabla_y^2 F(y^*(x; \theta); \theta)\right\}^{-1} + \alpha I\right] \nabla_\phi^\top h(d; \phi)$$
$$+ \nabla_x^\top L(z, x; \psi) \nabla_\phi^\top h(d; \phi),$$

*where $z = f^*(h(d; \phi); \theta)$ and $x = h(d; \phi)$.*

*Proof.* We can proceed similarly as the previous theorems. By chain rule,

$$\nabla_\phi L(f^*(h(d; \phi); \theta), h(d; \phi); \psi) = \nabla_z^\top L(z, w; \psi) \nabla_\phi^\top f^*(h(d; \phi); \theta) + \nabla_w^\top L(z, w; \psi) \nabla_\phi^\top h(d; \phi),$$

where $z = f^*(h(d;\phi))$ and $w = h(d;\phi)$. The second term does not require further computation. We will focus on the first term.

$$
\begin{aligned}
\nabla_\phi^\top f^*(h(d;\phi);\theta) :=& \nabla_\phi^\top \left\{ \nabla_x F_\theta^*(h(d;\phi)) + \alpha h(d;\phi) \right\} \\
=& \nabla_\phi^\top y^*(h(d;\phi);\theta) + \nabla_\phi^\top \alpha h(d;\phi) \\
=& \nabla_x^\top y^*(x;\theta) \nabla_\phi^\top h(d;\phi) + \nabla_\phi^\top \alpha h(d;\phi) \\
=& \left[ \left\{ \nabla_y^2 F(y^*(x;\theta);\theta) \right\}^{-1} + \alpha I \right] \nabla_\phi^\top h(d;\phi).
\end{aligned}
$$

This provides the desired result. $\qquad\square$

To summarize, gradients used for the update of each type of parameters can be written as follows:

**Corollary B.21.** *Let* $L := L(f^*(h(d;\phi);\theta), h(d;\phi);\psi)$. *Then under assumptions of Theorem B.20,*

$$
\nabla_\theta^\top L = -\nabla_z^\top L(z,x;\psi) \left\{ \nabla_y^2 F(y^*(x;\theta);\theta) \right\}^{-1} \partial_\theta^\top \nabla_y F(y^*(x;\theta);\theta),
$$
$$
\nabla_\phi^\top L = \nabla_z^\top L(z,x;\psi) \left[ \left\{ \nabla_y^2 F(y^*(x;\theta);\theta) \right\}^{-1} + \alpha I \right] \nabla_\phi^\top h(d;\phi) + \nabla_x^\top L(z,x;\psi) \nabla_\phi^\top h(d;\phi),
$$
$$
\nabla_\psi^\top L = \nabla_\psi^\top L(z,x;\psi),
$$

*where* $x = h(d;\phi)$ *and* $z = f(h(d;\phi);\theta)$.

## C Algorithms for Legendre-Fenchel Transformation

In this appendix, we discuss the implementation of the LFT as an optimization problem and properties derived from the choice the optimization algorithm. As a reminder, we suppose that a convex solver generates for a fixed $x$ a sequence of points $\{y_t(x)\}_{t=0,\dots,T}$ based on the objective function $\langle y, x \rangle - F(y)$, converging to the true optimizer $y^*(x) := \operatorname{argmax}_y \{ \langle y, x \rangle - F(y) \}$.

### C.1 Influence of Approximate Optimization on Bi-Lipschitz Constants: Experiments

We first run experiments with the following common convex solvers: steepest gradient descent (GD), Nesterov's accelerated gradient (AGD) (Nesterov, 1983), Adagrad (Duchi et al., 2011), RM-Sprop (Hinton et al., 2012), Adam (Kingma and Ba, 2015) and the Newton method. We calculated the bi-Lipschitz constants of the generated curve $y_t(x)$ for each optimization scheme at each iteration $t$. $F$ was chosen as a two-dimensional ICNN with two hidden layers. The activation was set as the softplus function $\log(1 + \exp(x))$. The derivative of the softplus function is the sigmoid function, which means the overall convex neural network becomes smooth. At the output of the ICNN, we added a regularization term $\|x\|^2/(2 \times 10)$ so that the overall function becomes 10-strongly convex as well. As a result, the Legendre-Fenchel transformation is also smooth and strongly convex with respective constants $c'$ and $c$ that we estimated beforehand. In other words, the Lipschitz and inverse Lipschitz constants of the function $y_t(x)$ with respect to $x$ should converge to $c'$ and $c$, respectively. As for the step size, we chose one so that the corresponding convex solver converged. For a $\mu$-strongly convex objective function, it is known that GD converges with a decreasing step size of $\frac{1}{\mu(t+1)}$. Thus, we chose this for GD but also for Adam and RMSprop since it helped the algorithm to converge. For Adagrad, we set it as $\frac{1}{\mu}$, for the Newton method as 1 and for AGD as $1/c$, the smoothness constant of the objective function. Results are shown in Figures 8 and 9. See Appendix H for further details on the experimental setup.

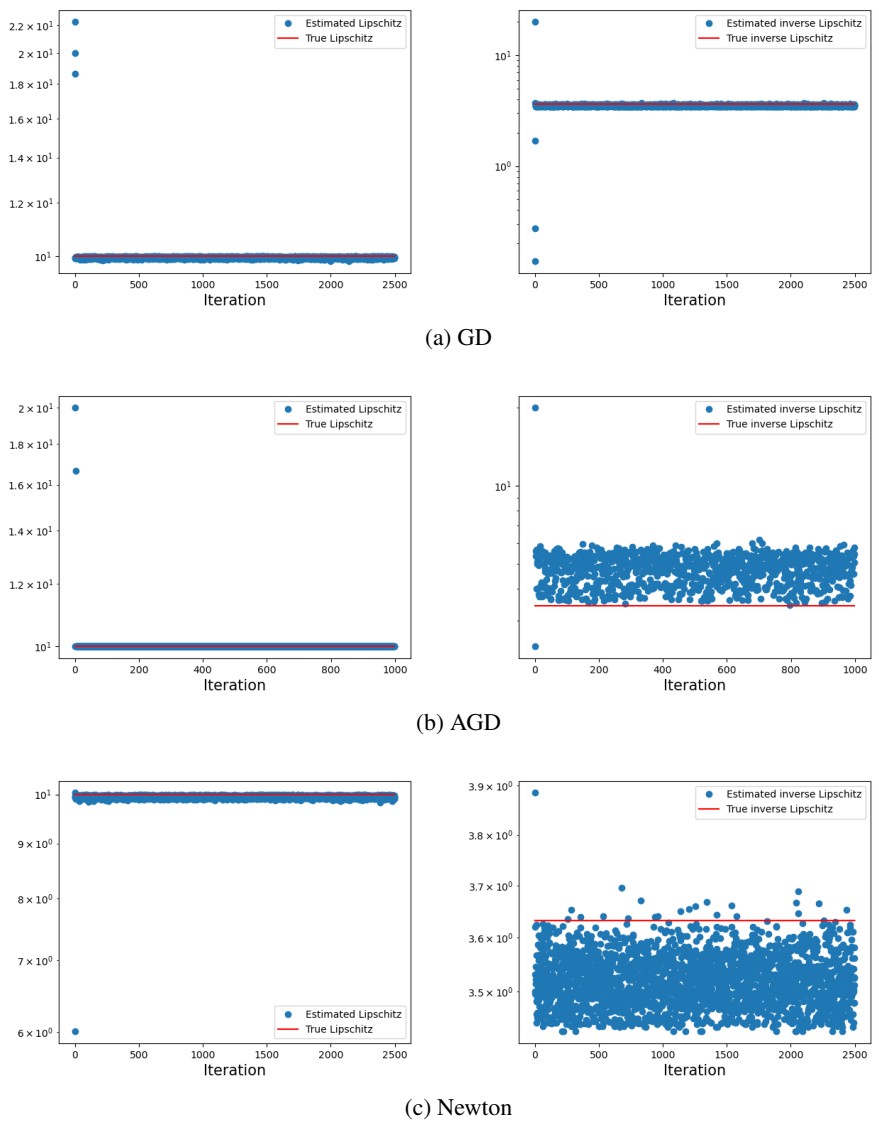

(a) GD

(b) AGD

(c) Newton

Figure 8: Evolution of bi-Lipschitzness (Lipschitz: left, inverse Lipschitz: right) through the iteration of several optimization algorithms: GD (top row), AGD (middle row) and Newton (bottom row).

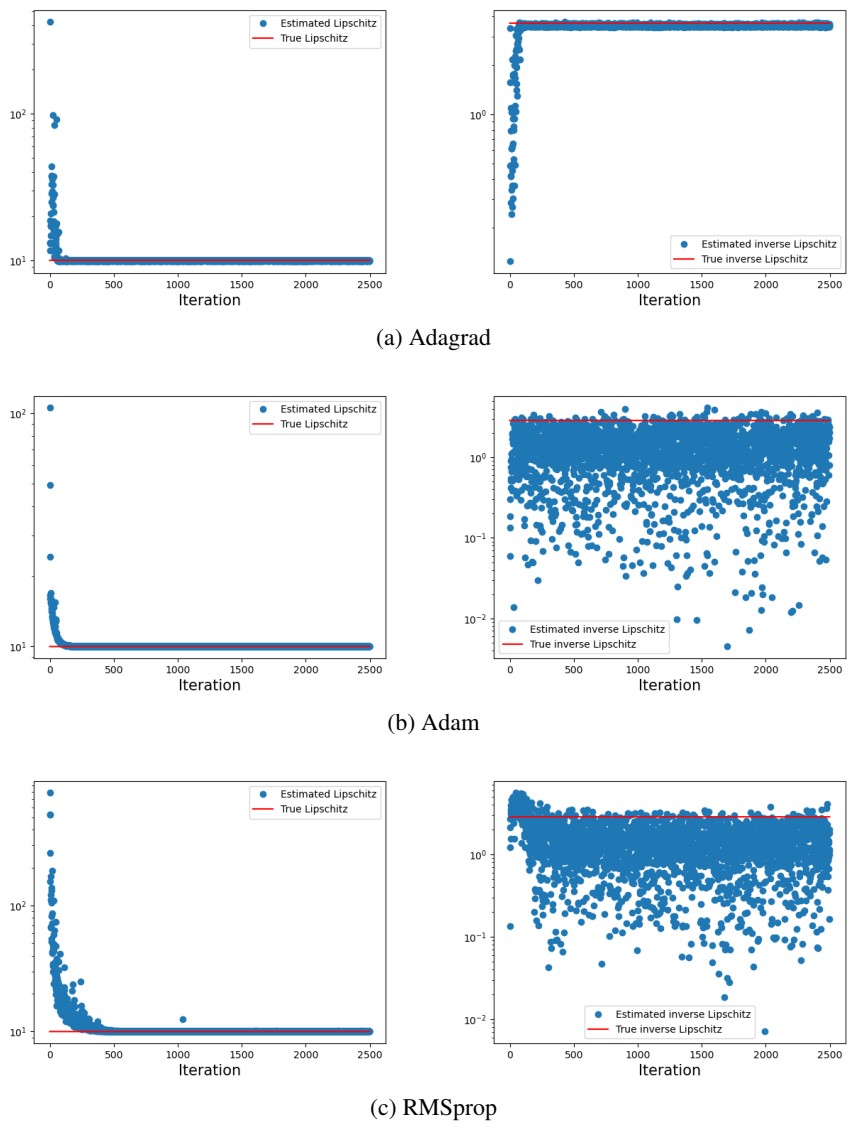

(a) Adagrad

(b) Adam

(c) RMSprop

Figure 9: Evolution of bi-Lipschitzness (Lipschitz: left, inverse Lipschitz: right) through the iteration of several optimization algorithms: Adagrad (top row), Adam (middle row) and RMSprop (bottom row).

As we can observe, the simple gradient descent, AGD, Adagrad and the Newton method perform well in the perspective of conservation of bi-Lipschitzness while RMSprop and Adam are not able to provide such feature, especially for the inverse Lipschitz constant. This incites us to use GD for simplicity or Adagrad as convex solver of the Legendre-Fenchel transformation.

## C.2  Influence of Approximate Optimization on Bi-Lipschitz Constants: Theoretical Analysis

Now that we have an idea of the behavior of well-known convex solvers, we proceed to the theoretical analysis of the approximation quality of optimization schemes concerning the bi-Lipschitz constants predefined in our model. The aim is to clarify when bi-Lipschitzness is preserved and to evaluate the effective bi-Lipschitz constants through the iterations.

### C.2.1 Proof for General Algorithms

Here, we focus on deriving a general non-asymptotic bounds, i.e., the evaluation and evolution of the bi-Lipschitzness of $y_t(x)$ for a finite period of timer for general optimization algorithms. This is a crude bound.

**Theorem C.1.** *Let the symbols defined as in Algorithm 1. Consider an optimization scheme of* $\sup_y \{\langle y, x \rangle - F_\theta(y)\}$ *generating points* $\{y_t(x)\}_t$ *that achieves an error of* $\|y_t(x) - y^*(x)\| \leq \epsilon(t, x)$ *after $t$ iterations, where $y^*(x)$ is the global maximum. Then, for all $x_i, x_j$ such that $\|x_i - x_j\| \geq \delta$ for $\delta > 0$,*

$$\alpha - (\epsilon(t, x_i) + \epsilon(t, x_j))/\delta \leq \frac{\|f_\theta^{(t)}(x_i) - f_\theta^{(t)}(x_j)\|}{\|x_i - x_j\|} \leq \alpha + \beta + (\epsilon(t, x_i) + \epsilon(t, x_j))/\delta,$$

*where $f_\theta^{(t)}(x) := y_t(x) + \alpha x$ is the finite time approximation of $f_\theta^*(x)$.*

*Proof.* Concerning the upper bound,

$$
\begin{aligned}
\|f_\theta^{(t)}(x_i) - f_\theta^{(t)}(x_j)\| =& \|y_t(x_i) + \alpha x_i - (y_t(x_j) + \alpha x_j)\| \\
\leq& \alpha \|x_i - x_j\| + \|y_t(x_i) - y_t(x_j)\| \\
\leq& \alpha \|x_i - x_j\| + \|y_t(x_i) - y^*(x_i)\| + \|y^*(x_i) - y^*(x_j)\| \\
& + \|y^*(x_j) - y_t(x_j)\| \\
\leq& \alpha \|x_i - x_j\| + \epsilon(t, x_i) + \|y^*(x_i) - y^*(x_j)\| + \epsilon(t, x_j) \\
\leq& \alpha \|x_i - x_j\| + \epsilon(t, x_i) + \|\nabla F_\theta^*(x_i) - \nabla F_\theta^*(x_j)\| + \epsilon(t, x_j) \\
\leq& \alpha \|x_i - x_j\| + \epsilon(t, x_i) + \beta \|x_i - x_j\| + \epsilon(t, x_j) \\
=& \alpha \|x_i - x_j\| + \beta \|x_i - x_j\| + \epsilon(t, x_i) + \epsilon(t, x_j),
\end{aligned}
$$

where $F_\theta^*$ is the Legendre-Fenchel transformation of $F_\theta$.

As for the lower bound, we begin by evaluating $\langle y_t(x_i) - y_t(x_j), x_i - x_j \rangle$.

$$
\begin{aligned}
\langle y_t(x_i) - y_t(x_j), x_i - x_j \rangle =& \langle y_t(x_i) - y^*(x_i), x_i - x_j \rangle + \langle y^*(x_i) - y^*(x_j), x_i - x_j \rangle \\
& + \langle y^*(x_j) - y_t(x_j), x_i - x_j \rangle \\
\geq& - \|y_t(x_i) - y^*(x_i)\| \|x_i - x_j\| \\
& + \langle \nabla F_\theta^*(x_i) - \nabla F_\theta^*(x_j), x_i - x_j \rangle \\
& - \|y^*(x_j) - y_t(x_j)\| \|x_i - x_j\| \\
\geq& - \epsilon(t, x_i) \|x_i - x_j\| + \langle \nabla F_\theta^*(x_i) - \nabla F_\theta^*(x_j), x_i - x_j \rangle \\
& - \epsilon(t, x_j) \|x_i - x_j\| \\
\geq& - \epsilon(t, x_i) \|x_i - x_j\| - \epsilon(t, x_j) \|x_i - x_j\|,
\end{aligned}
$$

where for the last inequality we used that for any convex function $\langle \nabla F_\theta^*(x_i) - \nabla F_\theta^*(x_j), x_i - x_j \rangle \geq 0$.

Now since $f_\theta^{(t)}(x) := y_t(x) + \alpha x$

$$
\begin{aligned}
\langle f_\theta^{(t)}(x_i) - f_\theta^{(t)}(x_j), x_i - x_j \rangle =& \langle y_t(x_i) + \alpha x_i - (y_t(x_j) + \alpha x_j), x_i - x_j \rangle \\
=& \langle y_t(x_i) - y_t(x_j), x_i - x_j \rangle + \alpha \|x_i - x_j\|^2 \\
\geq& \alpha \|x_i - x_j\|^2 - \epsilon(t, x_i) \|x_i - x_j\| - \epsilon(t, x_j) \|x_i - x_j\|.
\end{aligned}
$$

By Cauchy-Schwarz inequality, $\langle f_\theta^{(t)}(x_i) - f_\theta^{(t)}(x_j), x_i - x_j \rangle \leq \|f_\theta^{(t)}(x_i) - f_\theta^{(t)}(x_j)\| \|x_i - x_j\|$. Thus, we finally obtain

$$\|f_\theta^{(t)}(x_i) - f_\theta^{(t)}(x_j)\| \geq \alpha \|x_i - x_j\| - \epsilon(t, x_i) - \epsilon(t, x_j).$$

In order to derive the inequalities of the statement, it suffices to divide all sides by $\|x_i - x_j\|$ and use $\|x_i - x_j\| \geq \delta$ on the two obtained bounds. $\qquad \square$

Since $\epsilon(t, x) \to 0$ as $t \to \infty$ for any optimization scheme that converges, we can recover an arbitrarily close approximation of $\alpha$ and $\alpha + \beta$, the inverse Lipschitz and Lipschitz constants, with enough number of iterations. This is a general bound that can be applied to any algorithm with known behavior. Below, we provide two concrete evaluations applied to the gradient descent.

**Corollary C.2.** *Suppose $F_\theta$ is $\mu$-strongly convex and $x - \|\partial_{\mathrm{sub}} F_\theta(y_t)\| \leq G(x)$. If we employ gradient descent with $\eta_t = 1/(\mu(t+1))$ as a step size and a fixed initial point $y_0 = 0$, $\epsilon(t, x) = 2G(x)/(\mu\sqrt{t})$. Therefore,*

$$\alpha - 2(G(x_i) + G(x_j))/(\delta\mu\sqrt{t}) \leq \frac{\|f_\theta^{(t)}(x_i) - f_\theta^{(t)}(x_j)\|}{\|x_i - x_j\|} \leq \alpha + \beta + 2(G(x_i) + G(x_j))/(\delta\mu\sqrt{t}).$$

*Proof.* This follows from Lemma 1 of Rakhlin et al. (2012). $\qquad\square$

A similar bound can be provided when $x - \|\partial_{\mathrm{sub}} F_\theta(y_t)\| \leq G(x)$ is satisfied almost surely. See Lemma 2 of Rakhlin et al. (2012).

**Corollary C.3.** *Suppose $F_\theta$ is $\mu$-strongly convex and $\gamma$-smooth. If we employ gradient descent with $\eta_t = 1/\gamma$ as a step size and a fixed initial point $y_0$, we can choose $\epsilon(t, x) = \left(1 - \frac{\mu^2}{\gamma^2}\right)^{t/2} \|y_0(x) - y^*(x)\|^2$.*

*Proof.* Since $F$ is $\gamma$-smooth, $h(y) = \langle y, x \rangle - F(y)$ is $\gamma$-smooth and $\mu$-strongly concave. As a result,

$$
\begin{aligned}
\|y_{t+1} - y^*\|^2 &= \|y_t + \eta \nabla h(y_t) - y^*\|^2 \\
&= \|y_t - y^*\|^2 - 2\eta\langle \nabla h(y_t), y_t - y^* \rangle + \eta^2 \|\nabla h(y)\|^2 \\
&= \|y_t - y^*\|^2 - 2\eta\langle \nabla h(y_t) - \nabla h(y^*), y_t - y^* \rangle + \eta^2 \|\nabla h(y_t) - \nabla h(y^*)\|^2 \\
&\leq \|y_t - y^*\|^2 - 2\eta\frac{1}{\gamma}\|\nabla h(y_t) - \nabla h(y^*)\|^2 + \eta^2 \|\nabla h(y_t) - \nabla h(y^*)\|^2 \\
&= \|y_t - y^*\|^2 - \frac{1}{\gamma^2}\|\nabla h(y_t) - \nabla h(y^*)\|^2 \\
&\leq \|y_t - y^*\|^2 - \frac{\mu^2}{\gamma^2}\|y_t - y^*\|^2 \\
&= \left(1 - \frac{\mu^2}{\gamma^2}\right) \|y_t - y^*\|^2 \\
&= \left(1 - \frac{\mu^2}{\gamma^2}\right)^{t+1} \|y_0 - y^*\|^2,
\end{aligned}
$$

where we used that $\nabla h(y^*) = 0$ for the third equality, the co-coercivity of smooth convex functions for the first inequality, and inverse Lipschitzness for the second inequality. $\qquad\square$

This is a crude estimation of the bi-Lipschitz constants. As long as we assume that all values can only take discrete ones with intervals bigger than $\delta$, then this theorem provides a meaningful evaluation. For example, during training this does not pose a problem since the number of training data is usually finite. Technically speaking, we will not have any problem in practice either as the computer also deals with discrete values.

The limitation of the above theorem appears once we authorize infinite precision as we could take two values arbitrarily close to each other, i.e., $\delta \to 0$, leading the lower and upper bound to explode. This makes the model prone to adversarial attacks. One way to remedy this is to evaluate the Legendre-Fenchel transformation for some discrete values and interpolate with some bi-Lipschitz functions (e.g., linear) so that the whole function satisfies the expected bounds.

Ideally, we would like to derive a bound of the type

$$h_1(\|x_i - x_j\|, t) \leq \frac{\|f_\theta^{(t)}(x_i) - f_\theta^{(t)}(x_j)\|}{\|x_i - x_j\|} \leq h_2(\|x_i - x_j\|, t), \tag{10}$$

where $h_1$ and $h_2$ converge as $\|x_i - x_j\| \to 0$ and converge to 0 as $t \to \infty$. We could not achieve this in Theorem C.1, which is predictable as we did not take into account the similarity of the optimization paths of $x_i$ and $x_j$ and derived the bounds by isolating each point. The next step is thus to take into account this closeness. Nevertheless, this kind of analysis is more involved and we do not know if it is applicable to all optimization schemes. This could explain the existence of some algorithms where the bi-Lipschitzness do not evolve as desired as we have pointed out in Figure 9.

### C.2.2 Proof of Theorem 3.5

The following theorem shows the first half of the statement of Theorem 3.5. That is, we show that in the limit there is no bias for GD in terms of bi-Lipschitz constants.

**Theorem C.4.** *Let the symbols defined as in Algorithm 1. Consider an optimization scheme of* $\sup_y \{\langle y, x \rangle - F(y)\}$ *generating points* $\{y_t(x)\}_t$ *at the t-th iterations. If* $F$ *is* $\mu$*-strongly convex,* $\gamma$*-smooth and twice differentiable and we employ gradient descent with a step size* $\eta_t$ *so that the discrete optimization converges to the global maximum* $y^*(x)$*, then for all* $x$

$$\frac{1}{\gamma} \preceq \nabla_x^\top y_\infty(x) \preceq \frac{1}{\mu}.$$

*Proof.* The recurrence relation of GD is as follows:

$$y_{t+1}(x) = y_t(x) + \eta_t \{x - \nabla_y F(y_t(x))\}.$$

Since $F$ is twice differentiable, we can take the Jacobian of both sides:

$$\nabla_x^\top y_{t+1}(x) = \nabla_x^\top y_t(x) + \eta_t \{I - \nabla_y^2 F(y_t(x)) \nabla_x^\top y_t(x)\}.$$

By replacing $y_{t+1}(x)$ and $y_t(x)$ by $y_\infty(x)$ as the sequence is converging by hypothesis, we obtain

$$\begin{aligned}
\nabla_x^\top y_\infty(x) =& \nabla_x^\top y_\infty(x) + \eta_t \{I - \nabla_y^2 F(y_\infty(x)) \nabla_x^\top y_\infty(x)\} \\
=& \{I - \eta_t \nabla_y^2 F(y_\infty(x))\} \nabla_x^\top y_\infty(x) + \eta_t I
\end{aligned}$$

Therefore, since $\nabla_y^2 F(y)$ is invertible as $\mu \preceq \nabla_y^2 F(y)$,

$$\nabla_x^\top y_\infty(x) = \{\nabla_y^2 F(y_\infty(x))\}^{-1}.$$

By using $\mu \preceq \nabla_y^2 F(y) \preceq \gamma$, we arrive at the desired result. $\qquad\square$

**Remark C.5.** *Note that* $y_\infty(x)$ *and* $y^*(x)$ *are different in the sense that the former is defined by the limit* $(t \to \infty)$ *of the recurrence relation of the optimizer, while the latter is defined as the global maximum of* $\sup_y \{\langle y, x \rangle - F(y)\}$ *which also satisfies* $\nabla_y F(y^*(x)) = x$*. In short, they differ in the formulation and explicit dependence on* $x$*.*

Since $F$ is $\mu$-strongly convex and $\gamma$-smooth, its Legendre-Fenchel transform is $1/\gamma$-strongly convex and $1/\mu$-smooth, leading to a $(1/\gamma, 1/\mu)$-bi-Lipschitz function by taking the derivative of the latter. This behavior is inherited by the optimization scheme in the limit of $t \to \infty$ as the above theorem shows. The point is that there is no bias, which may have occurred if the influence of the step size persisted in the final result. A similar result can be obtained for AGD. For the others, the analysis is more complicated as the gradient is accumulated throughout the iterations. Since our main interest is non-asymptotic behavior, we will not further develop this.

We will now prove the second half of the theorem and show that equation (10) can be actually established for some parameter settings at least for Lipschitzness.

**Theorem C.6.** *Let the symbols defined as in Algorithm 1. Consider an optimization scheme of* $\sup_y \{\langle y, x \rangle - F(y)\}$ *generating points* $\{y_t(x)\}_t$ *at the t-th iterations and* $y^*(x)$ *is the global maximum. If* $F$ *is* $\mu$*-strongly convex and* $\gamma$*-smooth and we employ gradient descent with* $\eta_t = 1/\mu(t+1)$ *as a step size and* $y_0(x_i) = y_0$ *as initial point, then for all* $x_i, x_j$

$$\|y_{t+1}(x_i) - y_{t+1}(x_j)\| \leq \sqrt{1 - 2\eta_t \mu + \eta_t^2 \gamma^2} \|y_t(x_i) - y_t(x_j)\| + \eta_t \|x_i - x_j\|.$$

*As a result,*

$$\|y_{t+1}(x_i) - y_{t+1}(x_j)\| \leq h(t)\|x_i - x_j\|,$$

*where*

$$\lim_{t \to \infty} h(t) = \frac{1}{\mu}.$$

*Proof.* Let us first prove the recurrence relation of the statement. Throughout the proof, we use the following notations:

$$\Delta y_t := y_t(x_i) - y_t(x_j),$$
$$\Delta x := x_i - x_j,$$
$$\Delta D_t := \nabla F(y_t(x_i)) - \nabla F(y_t(x_j)).$$

Since

$$\|\Delta y_{t+1}\| = \|\Delta y_t - \eta_t \Delta D_t + \eta_t \Delta x\| \le \|\Delta y_t - \eta_t \Delta D_t\| + \eta_t \|\Delta x\|, \tag{11}$$

it suffices to evaluate $\|\Delta y_t - \eta_t \Delta D_t\|$. Now,

$$\begin{aligned}
\|\Delta y_t - \eta_t \Delta D_t\|^2 &= \|\Delta y_t\|^2 - 2\eta_t \langle \Delta y_t, \Delta D_t \rangle + \eta_t^2 \|\Delta D_t\|^2 \\
&\le \|\Delta y_t\|^2 - 2\eta_t \mu \|\Delta y_t\|^2 + \eta_t^2 \|\Delta D_t\|^2 \\
&\le \|\Delta y_t\|^2 - 2\eta_t \mu \|\Delta y_t\|^2 + \eta_t^2 \gamma^2 \|\Delta y_t\|^2 \\
&= \left(1 - 2\eta_t \mu + \eta^2 \gamma^2\right) \|\Delta y_t\|^2,
\end{aligned}$$

where for the first inequality we used that $F$ is $\mu$-strongly convex, i.e., $\langle \nabla F(y) - \nabla F(y'), y - y' \rangle \ge \mu \|y - y'\|^2$ and for the second inequality we used that $F$ is $\gamma$-smooth.

As $\mu \le \gamma$ always holds,

$$1 - 2\eta_t \mu + \eta^2 \gamma^2 = 1 - 2\eta_t \mu + \eta^2 \mu^2 + \eta^2(\gamma^2 - \mu^2) = (1 - \eta_t \mu)^2 + \eta^2(\gamma^2 - \mu^2) \ge 0.$$

As a result

$$\|\Delta y_t - \eta_t \Delta D_t\| \le \sqrt{1 - 2\eta_t \mu + \eta^2 \gamma^2} \|\Delta y_t\|.$$

By substituting this to equation (11), we obtain

$$\|\Delta y_{t+1}\| \le \sqrt{1 - 2\eta_t \mu + \eta_t^2 \gamma^2} \|\Delta y_t\| + \eta_t \|\Delta x\|,$$

which was the desired inequality.

Next, we assume $\|\Delta x\| > 0$ and divide both sides by $\|\Delta x\|$. Denoting $\alpha_t := \mu \|\Delta y_{t+1}\| / \|\Delta x\|$, we arrive at

$$\alpha_{t+1} \le \sqrt{1 - 2\eta_t \mu + \eta_t^2 \gamma^2} \alpha_t + \eta_t \mu.$$

We want to show that $\lim_{t \to \infty} \alpha_t = 1$.

By the above-mentioned argument when $t \ge 1$, we can further develop this as follows:

$$\begin{aligned}
\alpha_{t+1} &\le \sqrt{1 - 2\eta_t \mu + \eta_t^2 \gamma^2} \alpha_t + \eta_t \mu \\
&= \sqrt{(1 - \eta_t \mu)^2 + \eta_t^2(\gamma^2 - \mu^2)} \alpha_t + \eta_t \mu \\
&\le \sqrt{\left\{(1 - \eta_t \mu) + \frac{1}{2} \frac{\eta_t^2(\gamma^2 - \mu^2)}{1 - \eta_t \mu}\right\}^2} \alpha_t + \eta_t \mu \\
&= \left\{(1 - \eta_t \mu) + \frac{1}{2} \frac{\eta_t^2(\gamma^2 - \mu^2)}{1 - \eta_t \mu}\right\} \alpha_t + \eta_t \mu.
\end{aligned}$$

Since $\eta_t \mu = 1/(t+1)$, we get

$$\alpha_{t+1} = \left\{\frac{t}{t+1} + \frac{\gamma^2/\mu^2 - 1}{2} \frac{1}{t(t+1)}\right\} \alpha_t + \frac{1}{t+1}.$$

By multiplying both sides by $t + 1$, defining $A_t := t\alpha_t$ and $\kappa := \gamma^2/\mu^2 - 1 (\ge 0)$, we can conclude that

$$A_{t+1} \le \left(1 + \frac{\kappa}{2} \frac{1}{t^2}\right) A_t + 1. \tag{12}$$

Let us prove that from equation 12, we have

$$A_t \le t + \frac{\kappa}{2} \sum_{k=1}^{t-1} \frac{1}{k} \prod_{k < i \le t-1} \exp\left(\frac{\kappa}{2} \frac{1}{i^2}\right) \tag{13}$$

for all $t = 1, 2, \ldots$. We will proceed by mathematical induction.

When $t = 1$, $A_1 = \alpha_1$. Since $\|\Delta y_1\| = \|\Delta y_0 - \eta_0 \Delta D_0 + \eta_0 \Delta x\|$ and the initial point was a constant independent of $x_i$, $\|\Delta y_1\| = \eta_0 \|\Delta x\| = \frac{1}{\mu} \|\Delta x\|$. Thus, $A_1 = \alpha_1 = 1$. This means, equation (13) is satisfied.

Now suppose equation (13) holds for $t = m$. Then by equation 12,

$$
\begin{aligned}
A_{m+1} &\leq \left(1 + \frac{\kappa}{2} \frac{1}{m^2}\right) A_m + 1 \\
&\leq \left(1 + \frac{\kappa}{2} \frac{1}{m^2}\right) \left(m + \frac{\kappa}{2} \sum_{k=1}^{m-1} \frac{1}{k} \prod_{k < i \leq m-1} \exp\left(\frac{\kappa}{2} \frac{1}{i^2}\right)\right) + 1 \\
&\leq m + \frac{\kappa}{2} \frac{1}{m} + \left(1 + \frac{\kappa}{2} \frac{1}{m^2}\right) \left(\frac{\kappa}{2} \sum_{k=1}^{m-1} \frac{1}{k} \prod_{k < i \leq m-1} \exp\left(\frac{\kappa}{2} \frac{1}{i^2}\right)\right) + 1 \\
&\leq m + 1 + \frac{\kappa}{2} \frac{1}{m} + \exp\left(\frac{\kappa}{2} \frac{1}{m^2}\right) \left(\frac{\kappa}{2} \sum_{k=1}^{m-1} \frac{1}{k} \prod_{k < i \leq m-1} \exp\left(\frac{\kappa}{2} \frac{1}{i^2}\right)\right) \\
&\leq m + 1 + \frac{\kappa}{2} \frac{1}{m} + \left(\frac{\kappa}{2} \sum_{k=1}^{m-1} \frac{1}{k} \prod_{k < i \leq m} \exp\left(\frac{\kappa}{2} \frac{1}{i^2}\right)\right) \\
&\leq m + 1 + \left(\frac{\kappa}{2} \sum_{k=1}^{m} \frac{1}{k} \prod_{k < i \leq m} \exp\left(\frac{\kappa}{2} \frac{1}{i^2}\right)\right).
\end{aligned}
$$

Therefore, by mathematical induction equation (13) holds for all $t = 1, 2, \ldots$.

Finally, since

$$
\sum_{i=1}^{j} \frac{1}{i^2} \leq \sum_{i=1}^{\infty} \frac{1}{i^2} = \frac{\pi^2}{6} \quad \forall j = 1, 2, \ldots
$$

and

$$
\sum_{i=1}^{j} \frac{1}{i} \leq O(\log j),
$$

we can conclude that

$$
\begin{aligned}
\alpha_t &= \frac{A_t}{t} \\
&\leq \frac{t + \frac{\kappa}{2} \sum_{k=1}^{t-1} \frac{1}{k} \prod_{k < i \leq t-1} \exp\left(\frac{\kappa}{2} \frac{1}{i^2}\right)}{t} \\
&= \frac{t + \frac{\kappa}{2} \sum_{k=1}^{t-1} \frac{1}{k} \exp \sum_{k < i \leq t-1} \left(\frac{\kappa}{2} \frac{1}{i^2}\right)}{t} \\
&\leq \frac{t + \frac{\kappa}{2} \sum_{k=1}^{t-1} \frac{1}{k} \exp \frac{\kappa}{2} \frac{\pi^2}{6}}{t} \\
&\leq \frac{t + O(\log t)}{t} \\
&\to 1 \quad (t \to \infty).
\end{aligned}
$$

$\square$

This theorem proves that the generated curve $y_t(x)$ at a fixed time $t$ is Lipschitz with a constant that converges to the true value with a convergence speed of $O(\log t / t)$. This theorem is interesting as it not only guarantees that $y_t(x)$ is bi-Lipschitz through the whole iteration and converges to the desired values but also is equipped with a non-asymptotic bound offering a concrete convergence speed and a

value at each iteration. The advantage of this proof is that setting the step size as $\eta_t = 1/(\mu(t+1))$ is realistic, since in our setting we know the strong convexity constant which corresponds to $1/\beta$. In practice, the convergence of the bi-Lipschitz constants seems to be faster, which means that there may exist tighter bounds but we let this investigation for future work since our goal was to just assure a convergence with good properties.

As for the lower bound, we could not prove a similar property for GD. Nevertheless, we can guarantee that it will not diverge to $-\infty$ when $\|x_i - x_j\| \to 0$ as

$$\langle y_t(x_i) - y_t(x_j), x_i - x_j \rangle \leq -\|y_t(x_i) - y_t(x_j)\| \|x_i - x_j\|$$

and $\|y_t(x_i) - y_t(x_j)\|$ was just proved to be upper-bounded by a constant. We let further theoretical investigation for future work. Since GD shows and assures rather good performance, we will mainly use this algorithm in this paper.

### C.2.3 Analogue of Theorem 3.5 with optimal step size

An analogue of Theorem 3.5 can be shown with a different step size, i.e., $\eta = 1/\gamma$, where $\gamma$ is the smoothness constant of the objective function. This step size is optimal as it assures the fastest convergence. However, in practice, it is actually useless as estimating the smoothness constant $\gamma$ for the step size is computationally demanding and during training this constant is constantly changing. Only after training, it suffices to estimate $\gamma$ once, and all forward passes can be executed with this step size and convergence speed. The theorem is as follows:

**Theorem C.7.** *Let the symbols defined as in Algorithm 1. Consider an optimization scheme of $\sup_y \{\langle y, x \rangle - F(y)\}$ generating points $\{y_t(x)\}_t$ at the $t$-th iterations and $y^*(x)$ is the global maximum. If $F$ is $\mu$-strongly convex and $\gamma$-smooth and we employ gradient descent with $\eta_t = 1/\gamma$ as a step size and $y_0(x_i) = y_0$ as initial point, then for all $x_i$, $x_j$*

$$\|y_{t+1}(x_i) - y_{t+1}(x_j)\| \leq \left(1 - \frac{\mu}{\gamma}\right) \|y_t(x_i) - y_t(x_j)\| + \eta_t \|x_i - x_j\|.$$

*As a result,*

$$\|y_{t+1}(x_i) - y_{t+1}(x_j)\| \leq h(t) \|x_i - x_j\|,$$

*where*

$$\lim_{t \to \infty} h(t) = \frac{1}{\mu}.$$

*Proof.* Let us first prove the following inequality:

$$\langle \nabla F(y) - \nabla F(y'), y - y' \rangle \geq \frac{\mu\gamma}{\mu + \gamma} \|y - y'\|^2 + \frac{1}{\mu + \gamma} \|\nabla F(y) - \nabla F(y')\|^2. \qquad (14)$$

If $\mu = \gamma$, then by strong convexity of $F$,

$$\langle \nabla F(y) - \nabla F(y'), y - y' \rangle \geq \mu \|y - y'\|^2 = \frac{\mu\gamma}{\mu + \gamma} \|y - y'\|^2 + \frac{\mu^2}{\mu + \gamma} \|y - y'\|^2.$$

However, since $\mu = \gamma$, we have $\mu \|y - y'\| = \|\nabla F(y) - \nabla F(y')\|$. Thus, the above inequality leads to

$$\langle \nabla F(y) - \nabla F(y'), y - y' \rangle \geq \mu \|y - y'\|^2 = \frac{\mu\gamma}{\mu + \gamma} \|y - y'\|^2 + \frac{1}{\mu + \gamma} \|\nabla F(y) - \nabla F(y')\|^2.$$

When $\gamma > \mu$, since $F$ is $\mu$-strongly convex and $\gamma$-smooth, $F(y) - \frac{\mu}{2}\|y\|^2$ is convex and $\gamma - \mu$ smooth. This is straightforward from Definition 2 of Theorem B.2. Now, by applying Definition 4 of Theorem B.2 to $F(y) - \frac{\mu}{2}\|y\|^2$, we obtain

$$\langle \nabla F(y) - \mu y - (\nabla F(y') - \mu y'), y - y' \rangle \geq \frac{1}{\gamma - \mu} \|\nabla F(y) - \mu y - (\nabla F(y') - \mu y')\|^2.$$

By developing both sides, we arrive at

$$\langle \nabla F(y) - \nabla F(y'), y - y' \rangle - \mu \|y - y'\|^2$$
$$\geq \frac{1}{\gamma - \mu} \left\{ \|\nabla F(y) - \nabla F(y')\|^2 - 2\mu \langle \nabla F(y) - \nabla F(y'), y - y' \rangle + \mu^2 \|y - y'\|^2 \right\}.$$

Rearranging the terms, we have

$$\frac{\gamma + \mu}{\gamma - \mu} \langle \nabla F(y) - \nabla F(y'), y - y' \rangle \geq \frac{\mu\gamma}{\gamma - \mu} \|y - y'\|^2 + \frac{1}{\gamma - \mu} \|\nabla F(y) - \nabla F(y')\|^2,$$

which leads to

$$\langle \nabla F(y) - \nabla F(y'), y - y' \rangle \geq \frac{\mu\gamma}{\mu + \gamma} \|y - y'\|^2 + \frac{1}{\mu + \gamma} \|\nabla F(y) - \nabla F(y')\|^2.$$

Let us now prove the recurrence relation of the statement. Throughout the proof, we use the following notations:

$$\Delta y_t := y_t(x_i) - y_t(x_j),$$
$$\Delta x := x_i - x_j,$$
$$\Delta D_t := \nabla F(y_t(x_i)) - \nabla F(y_t(x_j)).$$

Since

$$\|\Delta y_{t+1}\| = \|\Delta y_t - \eta_t \Delta D_t + \eta_t \Delta x\| \leq \|\Delta y_t - \eta_t \Delta D_t\| + \eta_t \|\Delta x\|, \qquad (15)$$

it suffices to evaluate $\|\Delta y_t - \eta_t \Delta D_t\|$. Now,

$$\|\Delta y_t - \eta_t \Delta D_t\|^2 = \|\Delta y_t\|^2 - 2\eta_t \langle \Delta y_t, \Delta D_t \rangle + \eta_t^2 \|\Delta D_t\|^2$$

$$\leq \|\Delta y_t\|^2 - 2\eta_t \left( \frac{\mu\gamma}{\mu + \gamma} \|\Delta y_t\|^2 + \frac{1}{\mu + \gamma} \|\Delta D_t\|^2 \right) + \eta_t^2 \|\Delta D_t\|^2$$

$$= \left( 1 - \frac{2\eta_t \mu\gamma}{\mu + \gamma} \right) \|\Delta y_t\|^2 + \eta_t \left( \eta_t - \frac{2}{\mu + \gamma} \right) \|\Delta D_t\|^2.$$

where we used inequality (14) for the inequality.

Since we set $\eta_t = 1/\gamma$,

$$\|\Delta y_t - \eta_t \Delta D_t\|^2 \leq \left( 1 - \frac{2\mu}{\mu + \gamma} \right) \|\Delta y_t\|^2 + \frac{1}{\gamma} \left( \frac{1}{\gamma} - \frac{2}{\mu + \gamma} \right) \|\Delta D_t\|^2$$

$$\leq \left( 1 - \frac{2\mu}{\mu + \gamma} \right) \|\Delta y_t\|^2 + \frac{1}{\gamma} \left( \frac{1}{\gamma} - \frac{2}{\mu + \gamma} \right) \mu^2 \|\Delta y_t\|^2$$

$$= \left( 1 - \frac{\mu}{\gamma} \right)^2 \|\Delta y_t\|^2,$$

where for the second inequality we used that $\frac{1}{\gamma} - \frac{2}{\mu+\gamma} \leq 0$ ($\mu \leq \gamma$) and $\|\nabla F(y) - \nabla F(y')\| \geq \mu\|y - y'\|$ (inverse Lipschitzness). Substituting this to equation (15), we conclude that

$$\|\Delta y_{t+1}\| \leq \left( 1 - \frac{\mu}{\gamma} \right) \|\Delta y_t\| + \eta_t \|\Delta x\|$$

as $\mu/\gamma \leq 1$.

Since $\|\Delta y_1\| = \|\Delta y_0 - \eta_0 \Delta D_0 + \eta_0 \Delta x\|$ and the initial point was a constant independent of $x_i$, $\|\Delta y_1\| = \eta_0 \|\Delta x\| = \frac{1}{\gamma} \|\Delta x\|$, and

$$\|\Delta y_t\| \leq \frac{1}{\gamma} \sum_{i=0}^{t-1} \left( 1 - \frac{\mu}{\gamma} \right)^i \|\Delta x\|.$$

Finally,

$$\lim_{t \to \infty} \|\Delta y_t\| \leq \frac{1}{\gamma} \sum_{i=0}^{\infty} \left( 1 - \frac{\mu}{\gamma} \right)^i \|\Delta x\| = \frac{1/\gamma}{1 - (1 - (\mu/\gamma))} = \frac{1}{\mu} \|\Delta x\|.$$

$\square$

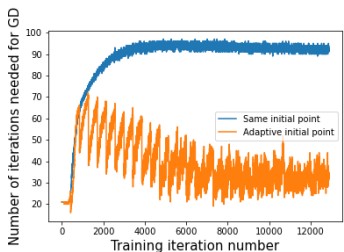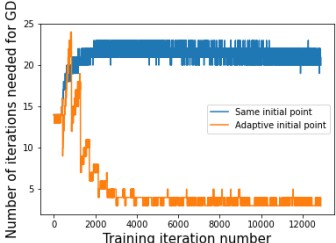

Figure 10: Evolution of the number of iterations needed for the gradient descent of the Legendre-Fenchel transformation to satisfy the stopping condition during training, with two regimes for the choice of initial points: a fixed or adaptive initial points during training on the FashionMNIST dataset from Appendix G.6.2.

## C.3 On the Choice of the Initial Point

In practice, as the iteration proceeds and $\theta$ evolves, the objective function changes. That is, the value of $\mathrm{argmax}_y \left\{ \langle y, x \rangle - F_\theta(y) \right\}$ for the same training data $x$ differs between each training epoch. However, it is often observed that the evolution of the weights is relatively slow. This means that the objective function does not change too much compared to the previous time. As a result, in order to accelerate the computation, we can use the final estimation of $\mathrm{argmax}_y \left\{ \langle y, x \rangle - F_\theta(y) \right\}$ of the previous epoch as the initial point of the next epoch. This may be a really simple trick, and there could exist more involved methods, but Figure 10 illustrates that this approach is already effective. See Appendix H for further details on the experimental setup. We also leave this research of better approaches for future work.

# D Equivalent Definitions of Smoothness under Convexity

In this Appendix, we prove Theorem B.2 by showing a more general theorem extended to dual norms.

## D.1 Theorem and Proof

Dual norms are defined as follows:

**Definition D.1.** *The* dual norm *of a norm* $\| \cdot \|$ *is defined as*

$$\|y\|_* = \sup \left\{ \langle x, y \rangle \mid \|x\| = 1 \right\}.$$

Let us now prove the following theorem. When the norm in question is $L_2$, its dual is also the same and we arrive at the initial statement.

**Theorem D.2.** *Let* $\gamma > 0$, $\| \cdot \|$ *and* $\| \cdot \|_*$ *a pair of dual norms, and* $F : \mathbb{R}^l \to \mathbb{R}^t$ *a differentiable convex function on a convex domain. Then the following are equivalent:*

D1. *The following holds for any* $x, y \in \mathrm{dom}F$:

$$\|\nabla F(x) - \nabla F(y)\|_* \leq \gamma \|x - y\|$$

.

D2. *The following holds for any* $x, y \in \mathrm{dom}F$:

$$\langle \nabla F(x) - \nabla F(y), x - y \rangle \leq \gamma \|x - y\|^2.$$

D3. *The following holds for any* $x, y \in \mathrm{dom}F$:

$$F(y) \leq F(x) + \nabla F(x)^\top (y - x) + \frac{\gamma}{2} \|y - x\|^2.$$

D4. *(co-coercivity) The following holds for any* $x, y \in \mathrm{dom}F$:

$$(\nabla F(x) - \nabla F(y))^\top (x - y) \geq \frac{1}{\gamma} \|\nabla F(x) - \nabla F(y)\|_*^2.$$

*Proof.*
D1.$\Rightarrow$ D2.

When $x = y$, the inequality trivially holds. When $x \neq y$, this is straightforward by the Cauchy-Schwarz inequality, since

$$
\begin{aligned}
\langle \nabla F(x) - \nabla F(y), x - y \rangle &= \|x - y\| \langle \nabla F(x) - \nabla F(y), \frac{x - y}{\|x - y\|} \rangle \\
&\leq \|x - y\| \sup \left\{ \langle z, \nabla F(x) - \nabla F(y) \rangle \mid \|z\| = 1 \right\} \\
&= \|x - y\| \|\nabla F(x) - \nabla F(y)\|_* \\
&\leq \gamma \|x - y\|^2,
\end{aligned}
$$

where we used D1. in the last inequality.
D2.$\Rightarrow$ D3.

Consider the function $G(t) := F(x + t(y - x))$, which is well-defined as the domain of $F$ is convex. Since $F$ is differentiable and $F(y) = G(1)$,

$$
\begin{aligned}
F(y) &= G(0) + \int_0^1 \frac{\mathrm{d}}{\mathrm{d}t} G(t) \mathrm{d}t \\
&= G(0) + \int_0^1 \langle \nabla F(x + t(y - x)), y - x \rangle \mathrm{d}t \\
&= F(x) + \langle \nabla F(x), y - x \rangle + \int_0^1 \langle \nabla F(x + t(y - x)) - \nabla F(x), y - x \rangle \mathrm{d}t \\
&\leq F(x) + \langle \nabla F(x), y - x \rangle + \int_0^1 t\gamma \|x - y\|^2 \mathrm{d}t \\
&= F(x) + \langle \nabla F(x), y - x \rangle + \frac{\gamma}{2} \|x - y\|^2,
\end{aligned}
$$

where we used D2. for the inequality.
D3.$\Rightarrow$ D4.

Consider the following function:

$$
F_x(z) := F(z) - \langle \nabla F(x), z \rangle,
$$

which is a $\gamma$-smooth function. Moreover, since $F$ is convex, $x$ is the minimizer of $F_x$. Consequently, the following PL inequality generalized to dual norms holds:

$$
\begin{aligned}
F_x(x) &= \inf_w F_x(w) \\
&\leq \inf_w \left\{ F_x(z) + \langle \nabla F_x(z), w - z \rangle + \frac{\gamma}{2} \|w - z\|^2 \right\} \\
&= \inf_{\|v\|=1} \inf_t \left\{ F_x(z) + \langle \nabla F_x(z), tv \rangle + \frac{\gamma}{2} \|tv\|^2 \right\} \\
&= \inf_{\|v\|=1} \inf_t \left\{ F_x(z) + t\langle \nabla F_x(z), v \rangle + \frac{\gamma}{2} t^2 \right\} \\
&= \inf_{\|v\|=1} \left\{ F_x(z) - \frac{1}{2\gamma} \left( \langle \nabla F_x(z), v \rangle \right)^2 \right\} \\
&= F_x(z) - \frac{1}{2\gamma} \|\nabla F_x(z)\|_*^2,
\end{aligned}
$$

where we used D3. for the inequality. Since this holds for all $z$, by substituting $z = y$, we obtain

$$
\frac{1}{2\gamma} \|\nabla F(y) - \nabla F(x)\|_*^2 = \frac{1}{2\gamma} \|\nabla F_x(y)\|_*^2 \leq F_x(y) - F_x(x) = F(y) - F(x) - \langle \nabla F(x), y - x \rangle.
$$

Since we can interchange $x$ and $y$, we also have

$$
\frac{1}{2\gamma} \|\nabla F(y) - \nabla F(x)\|_*^2 \leq F(x) - F(y) - \langle \nabla F(y), x - y \rangle.
$$

Summing side by side these two inequalities, we arrive at

$$\frac{1}{\gamma}\|\nabla F(y) - \nabla F(x)\|_*^2 \leq \langle \nabla F(x) - \nabla F(y), x - y \rangle.$$

D4.$\Rightarrow$ D1.

When $x = y$, the inequality trivially holds. When $x \neq y$, by Cauchy-Schwarz inequality,

$$\frac{1}{\gamma}\|\nabla F(y) - \nabla F(x)\|_*^2 \leq \langle \nabla F(x) - \nabla F(y), x - y \rangle \leq \|\nabla F(x) - \nabla F(y)\|_* \|x - y\|.$$

$\square$

# E    Uncertainty Estimation

## E.1    Background

Deep neural networks are nowadays used in many applications such as self-driving cars and large language models. However, in the real world, they are constantly subjected to a large amount of ambiguous scenarios, and assuring fail-safes is a top priority. One approach to realize this is to design the agent so that it quantifies the reliability of its own outputs and behavior, while achieving high performance. However, it is well-known that usual deep neural networks over-confidently extrapolate to unknown data or poorly perform in uncertainty tasks. As a result, the quantification of uncertainty has become the subject of many research, leading to Bayesian neural networks (Neal, 2012), Monte Carlo dropouts (Gal and Ghahramani, 2016) and deep ensembles (Lakshminarayanan et al., 2017). Unfortunately, these methods quickly become computationally expensive as they require to process multiple forward passes or to retain distribution samples over a large number of parameters. Furthermore, such ensemble methods cannot in the essence avoid that all constitutive members make the same mistake or place high confidence on the same out-of-distribution data.

Therefore, a line of work that concentrates on the uncertainty estimation using a single neural network has recently started to draw attention (Liu et al., 2020a; Van Amersfoort et al., 2020). In these works, they are interested in quantifying the uncertainty through a single forward pass and analyzing its behavior. On their own, they can not only provide new methods with low computational cost but also address the challenges of ensemble methods as successful individual agents can again be integrated in those models.

Thanks to these works, some properties necessary for an accurate uncertainty quantification have been discovered. Notably, bi-Lipschitzness is an indispensable inductive bias. Intuitively, bi-Lipschitzness guarantees the neural network to be distance aware, i.e., distance is moderately preserved between the input and the feature space, resulting in correct detection of out-of-distribution points. Without this property, it is known that the problem of *feature collapse* occurs, which refers to the phenomenon that out-of-distribution and in-distribution points overlap in the feature space, making out-of-distribution detection impossible in principle (Van Amersfoort et al., 2020).

## E.2    DUQ

We will focus on a recent model proposed by Van Amersfoort et al. (2020) as it is a model that has been shown to perform as well as deep ensemble methods that are the state-of-the-art in this area. It also performs well on the FashionMNIST dataset.

Their model is called Deep Uncertainty Quantification (DUQ) and is mainly used for classification tasks. The idea is to create a class $c$ represented by a centroid $e_c$ and to calculate the distance $K_c$ between a data point and all centroids. The data will be classified as the label of the closest centroid to that point. They draw inspiration from the radial basis function kernel. The mathematical formulation of $K_c$ is as follows:

$$K_c(f_\theta(x), e_c) = \exp\left\{-\frac{\frac{1}{n}\|W_c f_\theta(x) - e_c\|_2^2}{2\sigma^2}\right\},$$

where $f_\theta$ is a general neural network also called feature extractor, $W_c$ is a weight matrix, and $\sigma$ is a hyper parameter called length scale.

The loss function is the sum of the binary cross entropy over all classes. During the training, the centroids are updated by the following rule:

$$
\begin{aligned}
N_{c,t} &= \gamma N_{c,t-1} + (1-\gamma)n_{c,t}, \\
m_{c,t} &= \gamma m_{c,t-1} + (1-\gamma)\sigma_i W_c f_\theta(x_{c,t,i}), \\
e_{c,t} &= m_{c,t}/N_{c,t},
\end{aligned}
$$

where $n_{c,t}$ is the number of data points assigned to class $c$, $x_{c,t,i}$ is the $i$-th element of the minibatch corresponding to class $c$, and $\gamma$ is a hyper parameter (van den Oord et al., 2017).

Bi-Lipschitzness was incorporated through a gradient penalty term as follows:

$$
\lambda \left\{ \left\| \nabla_x \sum_c K_c \right\|^2 - 1 \right\}.
$$

The discussion about this regularization can be found in Section 2. By deleting this regularization term, and replacing the neural network $f_\theta$ by our bi-Lipschitz neural network, we obtain a model called here DUQ+BLNN.

# F  Extensions

## F.1  Extension 1: Different Input and Output Dimensions

With only one BLNN, we can only provide an output that has the same dimension as the input. This is a common problem in bi-Lipschitz models. In this extension, we provide an architecture that extends to a more realistic case with different input and output dimensions. While we will lose many theoretical guarantees we have provided so far, some will be retained. Furthermore, throughout this paper, we will show that this method works well in practice.

The idea follows Wang et al. (2021) and Kinoshita et al. (2023). Let $f_1^*(\cdot;\theta_1)$ and $f_2^*(\cdot;\theta_2)$ be two BLNN with input (and output) dimension $d_1$ and $d_2$, respectively. By composing them as

$$
f_2^*(Df_1^*(\cdot;\theta_1);\theta_2), \tag{16}
$$

where $D$ is a $d_2 \times d_1$ matrix with all diagonal components to 1, we obtain a function with input dimension $d_1$ and output dimension $d_2$. Note that if $d_1 \leq d_2$ then the whole bi-Lipschitz property is preserved.

On the other hand, if $d_1 \geq d_2$ then the Lipschitzness remains at least. Nevertheless, this parameterization (16) can represent any bi-Lipschitz function $l : \mathbb{R}^{d_1} \to \mathbb{R}^{d_2}$ with $d_1 \geq d_2$. It suffices to set $f_2^*$ as the identity map and consider the extension $\tilde{l} : \mathbb{R}^{d_1} \to \mathbb{R}^{d_1}$ whose image coincides with that of $l$ in the $\mathbb{R}^{d_2}$-dimensional subspace. This remains bi-Lipschitz like $l$. Therefore, this representation is reasonable to some extent.

The gradient computation can also be explicitly formulated as follows:

**Theorem F.1.** *Suppose a model with loss included*

$$
L := L(f_2^*(Df_1^*(h(d;\phi);\theta_1);\theta_2), h(d;\phi);\psi),
$$

*where $f_1^* := \nabla F_1^*(x_1;\theta_1) + \alpha_1 x_1$ and $f_2^* := \nabla F_2^*(x_2;\theta_2) + \alpha_2 x_2$ are two $C^2$ BLNN defined in Algorithm 1 with input dimensions $d_1$ and $d_2$, respectively. $D$ is a $d_2 \times d_1$ diagonal matrix with all diagonal elements set to 1. If $F$ and $F^*$ are both differentiable, then the gradients with respect to each parameter can be expressed as follows:*

$$
\begin{aligned}
\nabla_{\theta_1}^\top L &= -\nabla_z^\top L \Xi_2 D \left\{ \nabla_y^2 F_1(y_1^*(x;\theta_1);\theta_1) \right\}^{-1} \partial_{\theta_1}^\top \nabla_y F_1(y_1^*(x;\theta_1);\theta_1), \\
\nabla_{\theta_2}^\top L &= -\nabla_z^\top L(z,x;\psi) \left\{ \nabla_y^2 F_2(y_2^*(w;\theta_2);\theta_2) \right\}^{-1} \partial_{\theta_2}^\top \nabla_y F_2(y_2^*(w;\theta_2);\theta_2), \\
\nabla_\phi L &= \nabla_z^\top L(z,x;\psi)\Xi_2 D\Xi_1 \nabla_\phi h(d;\phi) + \nabla_x^\top L(z,x;\psi)\nabla_\phi^\top h(d;\phi), \\
\nabla_\psi^\top L &= \nabla_\psi^\top L(z,x;\psi),
\end{aligned}
$$

*where $\Xi_1 := \left\{ \nabla_y^2 F_1(y_1^*(x;\theta_1);\theta_1) \right\}^{-1} + \alpha_1 I$, $\Xi_2 := \left\{ \nabla_y^2 F_2(y_2^*(w;\theta_2);\theta_2) \right\}^{-1} + \alpha_2 I$, $z := f_2^*(Df_1^*(h(d;\phi);\theta_1);\theta_2)$, $w := Df_1^*(h(d;\phi);\theta_1)$ and $x := h(d;\phi)$.*

## F.2 Extension 2: Non-Homogeneous Functions

As pointed out by Jin and Lavaei (2020), in some cases, we may have knowledge of specific physical constraints or requirements to design a non-homogeneous control of the sensitivity like in physics-informed neural networks (Raissi et al., 2019). However, the current model imposes the same bi-Lipschitz constraint for each dimension. In the linear case, this corresponds to bounding all the eigenvalues of the matrix with the same constants from above and below respectively. For example, consider the following function:

$$f(x_1, x_2) = (2x_1, 100x_2).$$

If we can only choose 1 parameter for Lipschitzness, then it would be 100. However, this ignores the non-homogeneity of each dimension and their sparseness. In the above example, the Lipschitzness with respect to $x_1$ is only 2. Therefore, we would like to be able to control the bi-Lipschitzness of the function with respect to each dimension of the input.

This can be executed by introducing a diagonal matrix $A$ and $B$ instead of $\alpha$ and $\beta$ in our model. That is, instead of adding $\alpha \|x\|^2/2$ and $\|y\|^2/(2\beta)$ (see Algorithm 1), we add $x^\top A^2 x/2$ and $y^\top B^{-2} y/2$, respectively, which makes possible a more flexible control of the sensitivity of the overall function. The drawback is that we obtain more parameters to tune, but this can be avoided by designing these matrices as parameters to learn throughout the training too.

In the same idea, it may also be interesting to impose the convexity requirement on a limited number of variables. The Legendre-Fenchel transformation is then executed on those variables solely. As a result, we obtain partially bi-Lipschitz functions. This idea of incomplete convexity can be realized by the partially input convex neural network (PICNN) proposed by Amos et al. (2017). A PICNN $f(x, y; \theta)$, convex only with respect to $y$ with $L$ layers, can be formulated as follows:

$$
\begin{aligned}
u_{i+1} =& \tilde{g}_i \left( \tilde{W}_i u_i + \tilde{b}_i \right), \\
z_{i+1} =& g_i \left( W_i^{(z)} \left( z_i \odot [W_i^{(zu)} u_i + b_i^{(z)}]_+ \right) \right. \\
& \left. + W_i^{(y)} \left( y \odot \left( W_i^{(yu)} u_i + b_i^{(y)} \right) \right) + W_i^{(u)} u_i + b_i \right) \quad (i = 0, \ldots, L-1), \\
f(x, y; \theta) =& z_L, \ u_0 = x,
\end{aligned}
$$

where $\odot$ denotes the Hadamar product, only $W_i^{(z)}$ are non-negative, $g_i$ are convex non-decreasing and $W_0^{(z)} = 0$.

## F.3 Extension 3: General Norms

In this paper, we mainly discussed notions of Lispchitzness, inverse Lispchitzness, strong convexity and smoothness in terms of the $L^2$-norm. However, in some cases, it may be more interesting to work in other norms such as the $L_\infty$ norms (Zhang et al., 2022). In this extension, we briefly discuss how other norms can be introduced in our theoretical framework and its advantages. The previous extension can be regarded as such an example where the $L_2$ norm was changed to a weighted $L_2$ norm. Here, we treat the more general case. In this section, $\| \cdot \|$ will denote a general norm.

The dual norm can be defined as follows.

**Definition F.2.** *Let $\| \cdot \|$ be a norm. Its dual $\| \cdot \|_*$ is defined as follows:*

$$\|y\|_* := \sup \{ \langle x, y \rangle \mid \|x\| = 1 \}$$

For example, the dual of the $L_p$-norm for any $p \geq 1$ is the $L_q$-norm such that $1/p + 1/q = 1$. Definitions of strong convexity and smoothness are accordingly modified as follows:

**Definition F.3.** *Let $\mu > 0$. $F : \mathbb{R}^m \to \mathbb{R}$ is $\mu$-strongly convex with respect to a norm $\| \cdot \|$ if for all $t \in [0, 1]$*

$$F(tx + (1-t)y) \leq tF(x) + (1-t)F(y) - \frac{\mu}{2}t(1-t)\|x - y\|^2.$$

**Definition F.4.** *Let $\gamma > 0$. $F : \mathbb{R}^m \to \mathbb{R}$ is $\gamma$-smooth with respect to a norm $\| \cdot \|$ if $F$ is differentiable and*

$$F(y) \leq F(x) + \langle \nabla F(x), y - x \rangle + \frac{\gamma}{2}\|x - y\|^2.$$

Then, the following theorem holds:

**Theorem F.5** (Shalev-Shwartz and Singer (2010), Lemma 18)**.** *If $F$ is a closed and $\mu$-strongly convex function with respect to a norm $\|\cdot\|$, then its Legendre-Fenchel transformation $F^*$ is $1/\mu$-smooth with respect to the dual norm $\|\cdot\|_*$.*

Following Theorem D.2, the Legendre-Fenchel transformation of a closed and $\alpha$-strongly convex function with respect to a norm $\|\cdot\|$ satisfies:

$$\|\nabla F^*(x) - \nabla F^*(y)\| \leq \beta \|x - y\|_*. \tag{17}$$

This means that by taking the derivative, we obtain a monotone Lipschitz function with respect to the dual norm. If $\|\cdot\| = \|\cdot\|_1$, then $\|\cdot\|_* = \|\cdot\|_\infty$ and we arrive at a Lipschitzness in terms of the $L_\infty$ norm which is preferred in some situations (Zhang et al., 2022). We can thus create convex smooth functions with a relatively wide choice on the norm.

In many applications, since all norms are equivalent in a finite dimensional vector space, it may not be interesting to be able to deal with different norms. [3] In other words, controlling the sensitivity of a function $f$ in terms of a norm $\|\cdot\|$ can be executed simply through the calculation of the $L_2$-norm as

$$C_1 \|x - y\| \leq \|x - y\|_2 \leq C_2 \|x - y\|.$$

where $C_1$ and $C_2$ are constant independent of $x$. However, it turns out that usually $C_1$ and $C_2$ are dependent on the dimension. For example, between the $L_p$ and $L_q$ norms where $p \geq q$, the following holds:

$$\|x\|_p \leq \|x\|_q \leq n^{1/q - 1/p} \|x\|_p.$$

As a result, in high dimensional spaces, the equivalence of norms may become impractical as we will need to deal with extremely large or small values when translating the requirements into $L_2$ norm, making the training unstable. The main advantage of our approach is that we can avoid this as we can directly characterize the function in any desired norm without any dependency on the dimension.

On the other hand, the control of the lower bound (inverse Lipschitzness) becomes more difficult. Indeed, when $p = 1$ or $p > 2$, the $L_p$ norm is not strongly convex with respect to its own norm (Acu et al., 2023). As a result, we cannot similarly proceed like our approach for the $L_2$ norm and introduce the generalized inverse Lipschitzness.

Note that adding the squared norm $\alpha \|x\|_2^2 / 2$ to a function $F^*$ satisfying equation (17), will lead for the lower bound to

$$\alpha \|x - y\|_2^2 \leq \langle \nabla(F^*(x) + \alpha \|x\|_2/2) - \nabla(F^*(y) + \alpha \|y\|_2/2), x - y \rangle$$

but for the upper bound to

$$\langle \nabla(F^*(x) + \alpha \|x\|_2/2) - \nabla(F^*(y) + \alpha \|y\|_2/2), x - y \rangle \leq \beta \|x - y\|^2 + \alpha \|x - y\|_2^2$$
$$\leq (C_2 \alpha + \beta) \|x - y\|^2,$$

where the constant $C_2$ of the left hand side is highly dependent on the dimension. In high dimension, this may attenuate the importance of $\beta$, making control of the Lipschitzness difficult and unclear in practice.

The above discussion is rather theoretical. We let concrete applications, as well as deeper investigations for future work. However, this may be a promising avenue since we can use other norms and the dependency on the dimension is completely dropped.

## F.4   Extension 4: Improved Expressive Power through Superposition and Combination

**Superposition**   As mentioned in the main paper, while the expressive power of our bi-Lipschitz unit is constrained to $\alpha + \beta$-Lipschitz, $\alpha$-strongly monotone functions which are themselves the derivative of a real-valued function, this can be alleviated by superposing several BLNNs. For example, we have tested to fit the sign function, and the composition of 5 BLNNs could achieve an accuracy around 0.

---

[3] Any two norms $\|\cdot\|_{(1)}$ and $\|\cdot\|_{(2)}$ on a finite dimensional space X over $\mathbb{C}$ are equivalent. That is, there exist constants $C_1$ and $C_2$ so that for all $x \in X$, $C_1 \|x\|_{(1)} \leq \|x\|_{(2)} \leq C_2 \|x\|_{(1)}$.

Table 4: Out-of-distribution detection task of CIFAR10 vs SVHN with DUQ and BLNNconv.

| Models | Accuracy | Loss | AUROC SVHN |
|---|---|---|---|
| DUQ | 0.929 | 0.04 | 0.83 |
| BLNNconv then DUQ | **0.930** | **0.04** | **0.89** |

**Combination** Another solution to improve the expressive power of our BLNN is to combine it with other architectures. While there may be various approaches, our model could be used as a pre-processing module for difficult tasks since the BLNN is by definition bi-Lipschitz which means that the geometric properties of the input data is relatively preserved in the output as well. Therefore, for some problems, we could implement a model where we first process the data through a BLNN and then transfer it to other networks. For example, we conducted an experiment using the convolutional version of BLNN (BLNNconv) on the problem of uncertainty estimation (as in Subsection 4.2) with the CIFAR-10 vs. SVHN dataset to illustrate the scalability of our method (Van Amersfoort et al., 2020). For this problem, we implemented the model so that we first process the data through BLNNconv and then transfer it to the DUQ. The result compared to DUQ can be found in Table 4. Our model is not only scalable to large-scale networks but also improves out-of-detection performance (the AUROC of SVHN). Using BLNNconv instead of the fully connected BLNN also improved the computation time (e.g., 2.5 times faster for the 5 first iterations).

### F.5 A Simpler Architecture

In this work, the Legendre-Fenchel transformation was used in order to provide a direct and single parameterization of the Lipschitzness in contrast to prior methods that controlled it on a layer-wise basis. At the expense of losing this benefit, it is also possible to characterize Lipschitzness at each layer. This may be more practical and faster to apply in some contexts. In this case, we will have to control the spectral norm of all the weights. Since we calculate the derivative, the relation between the overall Lipschitz constant and that of each layer is complex though. We provide it in the following proposition as reference.

**Proposition F.6.** *Define the ICNN $G_\theta$ as follows*

$$z_{i+1} = g_i(W_i^{(z)} z_i + W_i^{(y)} y + b_i) \ (i = 0, \ldots, k-1),$$
$$G_\theta(y) = z_k,$$

*where $g_i$ is a 1-Lipschitz function, with bounded and Lipschitz derivative. Then the Lipschitz constant of $\nabla G_\theta$ can be obtained by solving the following set of inequalities, where $\alpha_k$ is the Lipschitz constant:*

$$\alpha_1 = \mathrm{Lip}(\nabla g_1) \|W_1^{(y)}\|_2^2$$
$$\beta_1 = \|W_1^{(y)}\|_2$$
$$\alpha_{i+1} \leq \|\nabla g_i\|_\infty \|W_i^{(z)}\|_2 \alpha_i$$
$$\quad + \left\{ \mathrm{Lip}(\nabla g_i) \|W_i^{(z)}\|_2^2 \beta_i + 2\mathrm{Lip}(\nabla g_i) \|W_i^{(y)}\|_2 \|W_i^{(z)}\|_2 \right\} \beta_i$$
$$\quad + \mathrm{Lip}(\nabla g_i) \|W_i^{(y)}\|_2^2$$
$$\beta_{i+1} \leq \|W_i^{(z)}\|_2 \beta_i + \|W_i^{(y)}\|_2 \quad (i = 1, \ldots, k-1).$$

*Proof.* Let us denote, $z_i(y)$ the $i$-th layer of the ICNN with input $y$. By construction,

$$\|\nabla_y z_{i+1}(y_1) - \nabla_y z_{i+1}(y_2)\|$$
$$= \left\| \nabla g_i \left( W_i^{(z)} z_i(y_1) + W_i^{(y)} y_1 + b_i \right) \left( W_i^{(z)} \nabla_y z_i(y_1) + W_i^{(y)} \right) \right.$$
$$\left. - \nabla g_i \left( W_i^{(z)} z_i(y_2) + W_i^{(y)} y_2 + b_i \right) \left( W_i^{(z)} \nabla_y z_i(y_2) + W_i^{(y)} \right) \right\|.$$

By defining $h_i(y) := \nabla g_i \left( W_i^{(z)} z_i(y) + W_i^{(y)} y + b_i \right)$, we can further develop as follows:

$$\|\nabla_y z_{i+1}(y_1) - \nabla_y z_{i+1}(y_2)\| \leq \left\| h_i(y_1) W_i^{(z)} \nabla_y z_i(y_1) - h_i(y_2) W_i^{(z)} \nabla_y z_i(y_2) \right\|$$
$$+ \left\| h_i(y_1) W_i^{(y)} - h_i(y_2) W_i^{(y)} \right\|.$$

The second term of the right hand side becomes:

$$\left\| h_i(y_1) \ W_i^{(y)} - h_i(y_2) W_i^{(y)} \right\|$$
$$\leq \text{Lip}(\nabla g_i) \left\| W_i^{(z)} z_i(y_1) + W_i^{(y)} y_1 + b_i - \left( W_i^{(z)} z_i(y_2) + W_i^{(y)} y_2 + b_i \right) \right\| \|W_i^{(y)}\|$$
$$\leq \text{Lip}(\nabla g_i) \|W_i^{(y)}\| \left( \|W_i^{(z)}\| \|z_i(y_1) - z_i(y_2)\| + \|W_i^{(y)}\| \|y_1 - y_2\| \right).$$

As for the first term,

$$\left\| h_i(y_1) \, W_i^{(z)} \nabla_y z_i(y_1) - h_i(y_2) W_i^{(z)} \nabla_y z_i(y_2) \right\|$$
$$= \left\| h_i(y_1) W_i^{(z)} \nabla_y z_i(y_1) - h_i(y_2) W_i^{(z)} \nabla_y z_i(y_1) \right.$$
$$\left. + h_i(y_2) W_i^{(z)} \nabla_y z_i(y_1) - h_i(y_2) W_i^{(z)} \nabla_y z_i(y_2) \right\|$$
$$\|h_i(y_1) - h_i(y_2)\| \|W_i^{(z)}\| \|\nabla_y z_i\| + \|\nabla g_i\|_\infty \|W_i^{(z)}\| \|\nabla_y z_i(y_1) - \nabla_y z_i(y_2)\|$$
$$\leq \text{Lip}(\nabla g_i) \left( \|W_i^{(z)}\| \|z_i(y_1) - z_i(y_2)\| + \|W_i^{(y)}\| \|y_1 - y_2\| \right) \|W_i^{(z)}\| \|\nabla_y z_i\|$$
$$+ \|\nabla g_i\|_\infty \|W_i^{(z)}\| \|\nabla_y z_i(y_1) - \nabla_y z_i(y_2)\|,$$

where in the last inequality we used the definition of $h_i(y)$.

On the other hand, we have

$$\|\nabla_y z_i\| \leq \|W_{i-1}^{(z)}\| \|\nabla_y z_i\| + \|W_{i-1}^{(y)}\|.$$

By replacing $\|\nabla_y z_i\|$ by $\beta_i$, we obtain

$$\beta_i \leq \|W_{i-1}^{(z)}\| \beta_{i-1} + \|W_{i-1}^{(y)}\|.$$

Since $z_i(y)$ is differentiable,

$$\|z_i(y_1) - z_i(y_2)\| \leq \beta_i.$$

Finally, by setting $\alpha_i := \|\nabla_y z_{i+1}(y_1) - \nabla_y z_{i+1}(y_2)\| / \|y_1 - y_2\|$ and substituting the results into the first inequality, we obtain the desired result. $\qquad \square$

**Remark F.7.** *Note that in the above theorem $g_i$ has to be not only convex non-decreasing but also 1-Lipschitz with bounded and Lipschitz derivative. The layer-wise formulation requires thus further conditions on $g_i$ that were unnecessary in the BLNN (Algorithm 1). This means that our original model has more freedom and flexibility than the above simple approach.*

# G    Additional Experiments

See Appendix H for further details on the setup of each experiment.

## G.1    Simple Estimation of Bi-Lipschitz Constants at Initialization

First of all, we verify that BLNN behaves as expected with different values of $\alpha$ and $\beta$. We set all $g_i$ as ReLU or softplus functions. The Legendre-Fenchel transformation step is executed with GD (since it is simple and has good theoretical guarantees) until 1000 iterations are reached or the stopping condition, i.e., $\|\nabla_y \left( \langle y, x \rangle - F(y) \right)\| < 10^{-3}$, is satisfied. The BLNN has input dimension 2, and $\alpha$ is set to 4 as varying it does not severely impact the fundamental quality of the theoretical bounds.

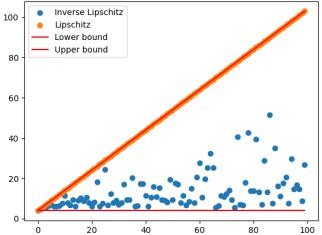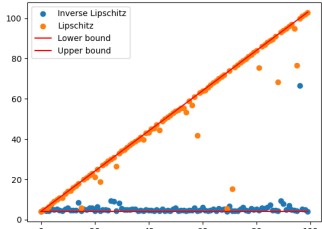

Figure 11: Estimated Lipschitz and inverse Lipschitz constants of BLNN with gradient descent with softplus activation functions. The left figure is with 3 hidden layers and the right with 10. The $x$ axis corresponds to $j$ with $\beta = 0.05 + 99.95 \cdot j/100$.

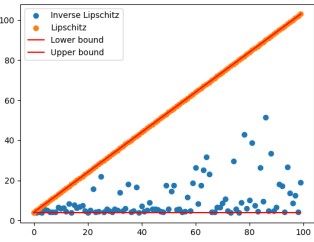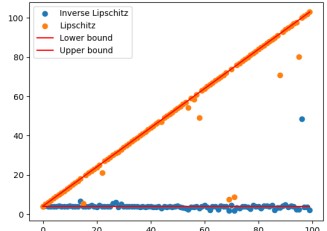

Figure 12: Estimated Lipschitz and inverse Lipschitz constants of BLNN with gradient descent with ReLU activation functions. The left figure is with 3 hidden layers and the right with 10. The $x$ axis corresponds to $j$ with $\beta = 0.05 + 99.95 \cdot j/100$

On the other hand, $\beta$ was changed following the sequence $\{0.05 + 99.95 \cdot j/100\}_{j=0}^{100}$. See Figures 11 and 12 for results.

As we can observe, our model combined with gradient descent can generate functions with bi-Lipschitz constants that are inside the predefined bounds as expected for both softplus and ReLU activation functions. That is why, it seems reasonable to use the simple gradient descent in order to execute the Legendre-Fenchel transformation.

Theoretically, since the initialization of weights are executed at random, the Lipschitz and inverse Lipschitz constants can be any value between $\alpha$ and $\alpha + \beta$. Curiously though, the estimated Lipschitzness is close to the upper bound for almost every model. Note that we are generating the network and then directly estimating the bi-Lipschitz constants, which means this phenomenon is possibly closely related to the initialization of weights. To better understand this phenomenon, consider the linear network $x \mapsto Wx$ where $W \in \mathbb{R}^{n \times n}$. In our architecture, this can be regarded as a simplification of $\nabla F_\theta$, where $F_\theta$ is the core ICNN. If we add $x/\beta$ to this network and compute the inverse operation, corresponding to $\nabla F_\theta^*$, we obtain $y \mapsto (W + I/\beta)^{-1}y$. By letting $\sigma_{\min}$ the smallest singular value of $W$, the Lipschitz constant becomes $(\sigma_{\min} + 1/\beta)^{-1}$. As a result, if the initialization provides a $W$ with small $\sigma_{\min}$ close to 0, we inevitably obtain a function with Lipschitz constant close to the upper bound $\beta$, independently of the distribution of the other eigenvalues. This seems to happen in Figures 11 and 12. In other words, the construction employing the default initialization is biased, or more concretely, the derivative of an ICNN $F_\theta$ with initialized weights is with high probability $\epsilon$-inverse Lipschitz with $\epsilon << 1$. This is supported by Figure 13 which calculated the bi-Lipschitz constants of a $(4, 60)$-BLNN initialized over 100 different trials. While the inverse Lipschitz constant is moderately distributed, the Lipschitz constant is concentrated on the theoretical maximum, i.e., 64.

We also modified the initialization procedure of the non-negative weights $W_i^{(z)}$ to confirm our hypothesis. The default approach was to draw each element from a uniform distribution proposed by Glorot and Bengio (2010) and clamp all negative elements to 0. Figure 14 replaces this to a uniform distribution on an interval of $[0, 1]$, and Figure 15 to $[1.0, 1.1]$.

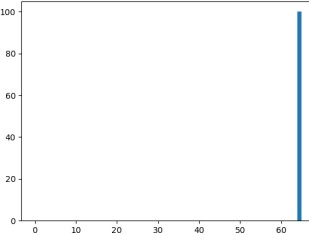 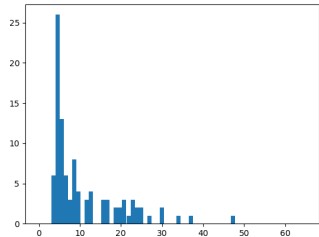

Figure 13: Histograms of Lipschitz and inverse Lipschitz constants of 100 $(4, 60)$-BLNNs with non-negative weights $W_i^{(z)}$ initialized following the uniform distribution of Glorot and Bengio (2010). Negative elements were clamped to 0.

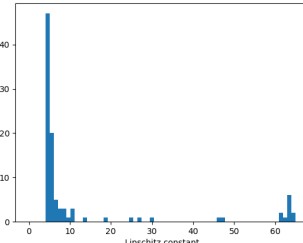 

Figure 14: Histograms of Lipschitz and inverse Lipschitz constants of 100 $(4, 60)$-BLNNs with non-negative weights $W_i^{(z)}$ initialized following a uniform distribution over $[0, 1.0]$.

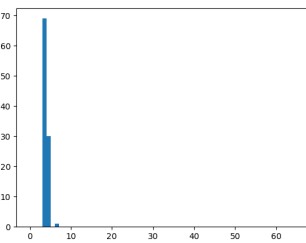 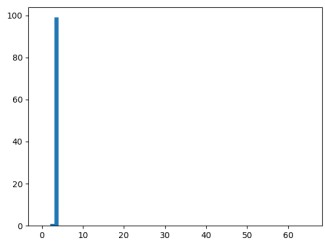

Figure 15: Histograms of Lipschitz and inverse Lipschitz constants of 100 $(4, 60)$-BLNNs with non-negative weights $W_i^{(z)}$ initialized following a uniform distribution over $[1.0, 1.1]$.

As we can conclude, the choice of the initialization impacts the distribution of the starting bi-Lipschitz constants for a $(\alpha, \beta)$-BLNN while theoretically any values are possible. In the default setting, we have a strong bias for the Lipschitz constant. Nevertheless, the fact that the bi-Lipschitz constants are close to the bounds can be regarded as the representation of a function with rich features. For example, if a matrix has the greatest eigenvalue and the smallest eigenvalue close to the limits, it implies that the other eigenvalues can assume any values in-between, allowing for considerable flexibility. This is a direct consequence of our meticulous construction of bi-Lipschitz neural networks respecting the desired constraints using convex neural networks and Legendre-Fenchel transformation.

The adequate initialization of the weights of an ICNN in order to obtain BLNNs with a richer distribution of bi-Lipschitz constants is a whole new problem and outside the scope of our work. We will not investigate further but it seems to be an interesting research topic since an appropriate initialization may facilitate training. In this paper, we used the original setup.

Table 5: Tightness of Lipschitz bound with different methods. Each model was built with an upper-bound constraint on the Lipschitzness $L$, and the true Lipschitzness of the model after training was evaluated. The percentage between this value and $L$ is reported in the table, with a mean and standard deviation over five different seeds.

| Models (Nbr. of param.) | $L = 2$ | $L = 5$ | $L = 10$ | $L = 50$ |
|---|---|---|---|---|
| SN (4.3K) | 96.8%±1.3 | 80.1%±5.0 | 68.9 %±5.4 | 32.5%±5.8 |
| AOL (4.3K) | 96.2%±0.7 | 65.9%±4.7 | 46.5%±4.4 | 15.4%±2.8 |
| Orthogonal (4.3K) | 98.7%±0.4 | 75.1%±12.0 | 48.9%±11.7 | 14.6%±3.7 |
| SLL (4.2K) | 97.5 %±0.4 | 77.6%±6.5 | 50.0%±8.5 | 15.7%±6.5 |
| Sandwich (4.5K) | 97.5%±0.7 | 84.3%±2.0 | 60.2%±4.1 | 16.4%±3.6 |
| LMN (4.4K) | **100.0%±0.0** | **100.0%±0.0** | 98.5%±0.3 | 26.0%±3.6 |
| BiLipNet (5.0K) | **100.0%± 0.0** | 98.0%±0.8 | 58.9%±3.6 | 6.8%±0.6 |
| Ours (4.3K) | **100.2%±0.3** | **100.0%±0.0** | **99.4%±1.2** | **99.9%±0.1** |

## G.2 Tightness of the Bounds: Underestimation Case

This subsection provides additional results of experiment 4.1 of the main paper. As a reminder, this experiment verifies the tightness of the proposed architecture by fitting the following function that has a discontinuity at the point $x = 0$:

$$f(x) = \begin{cases} x & (x \leq 0) \\ x + 1 & (x > 0) \end{cases}$$

Therefore, when constraining the Lipschitzness of the model by a positive finite $L$, a model with tight Lipschitz bounds should achieve that upper bound around $x = 0$ in order to reproduce the behavior of $f$. We compare our model with other Lipschitz constrained architectures that were introduced in Appendix A. We take as comparison the spectral normalization (SN) (Miyato et al., 2018), AOL (Prach and Lampert, 2022), Orthogonal (Trockman and Kolter, 2021), SLL (Araujo et al., 2023) and Sandwich (Wang and Manchester, 2023). They can all regulate the Lipschitzness by an upper bound, but it is executed on a layer-wise basis.

Models are built so that $L$ is set to 2, 5, 10 and 50. For the compared prior works, this is realized by making all layers 1-Lipschitz, and scaling the input by $L$, which is the strategy often employed in practice. After learning was completed, the empirical Lipschitz constant was computed. The percentage between the empirical Lipschitz constant and the true upper bound $L$ can be found in Table 5. In order to be able to legitimately compare results, the number of parameters for each model were all of the same order. The difference in tightness is even clearer when setting the upper bound to 50 and plotting the result as shown in Figure 16. Only our method can capture the presence of a discontinuous jump around 0. We can conclude that our method clearly achieves the tightest bound for Lipschitzness as well as the most coherent fit of the function.

## G.3 Flexibility of the Model: Overestimation Case

This subsection provides additional results of experiment 4.1 of the main paper. As a reminder, this experiment verifies the flexibility of our model. While we already provided a theoretical guarantee of the expressive power of our model, it is important to show that it can learn without problem functions with smaller Lipschitz and larger inverse Lipschitz constants than those we set as parameters. At the same time, this addresses one concern of the previous experiment since the tightness of our model may be interpreted as a result of the strong influence the regularization terms we add, effectively generating only $\alpha + \beta$-Lipschitz functions. Therefore, we lead this experiment, where we overestimated the Lipschitz constant of the target function. The Lipschitz constant was set to 50 but the function we want to learn is a linear function with slope 1. Experimental setting was the same as the previous subsection. Results are shown in Figures 17 and 18.

Our model can without any problem fit the curve $y = 50x$ and $y = x$ even in the overestimation scenario. Interestingly, this is not the case for prior work. On the one hand, SN and AOL cannot learn the linear function with slope 50. On the other hand, others cannot learn the simple identity function even after convergence. While the overall trend is along $y = x$, we can observe small fluctuations.

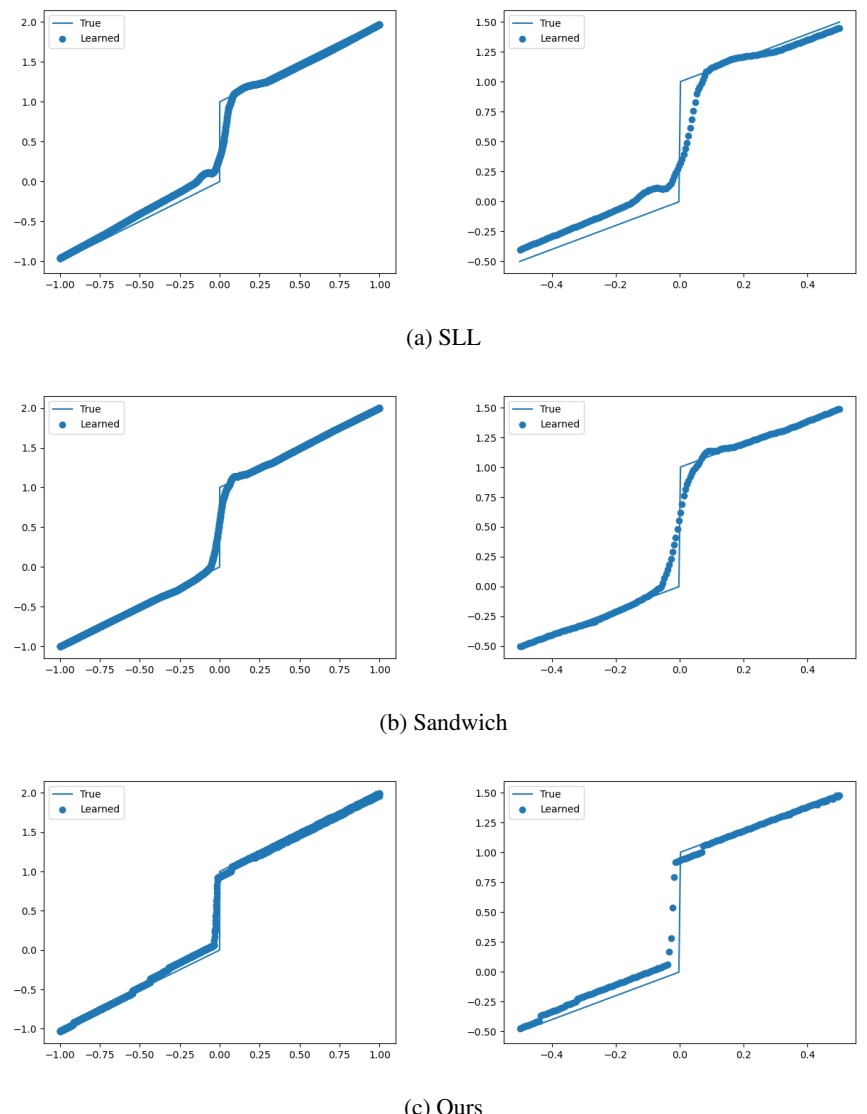

(a) SLL

(b) Sandwich

(c) Ours

Figure 16: Results of fitting the curve with SLL (first row), Sandwich (second row) and our method (third row) with a specified Lispchitzness of 50. The right column is a zoom of the right figure.

This may be due to the large scaling factor $L$ we multiply at the input, leading to overfitting or interpolation which usually depends on the smoothness. That is, with high Lipschitzness, the function is likely to dynamically fluctuate around unseen points during training.

In order to examine the influence of this scaling factor on the training we run the experiment again with even larger $L$, namely, 100 and 1000. The evolution of the loss and the fitted function for $L = 1000$ can be found in Figures 19 and 20, respectively. As we can observe, only our method is hardly affected by the size of $L$ and achieves a good generalization performance. While we do not have a rigorous mathematical explanation for this phenomenon, we believe our approach provides a better regularization than the previous methods. In other words, these layer-wise methods seem to provide a loss landscape that has many spurious minima where the function is driven. This phenomenon is aggravated with higher $L$. The difference in the smooth decrease of the loss function also supports this. The presence of these fluctuations due to the large scaling factor is thought to be related with the way we incorporated $L$ into the model. For previous work, we had no choice but

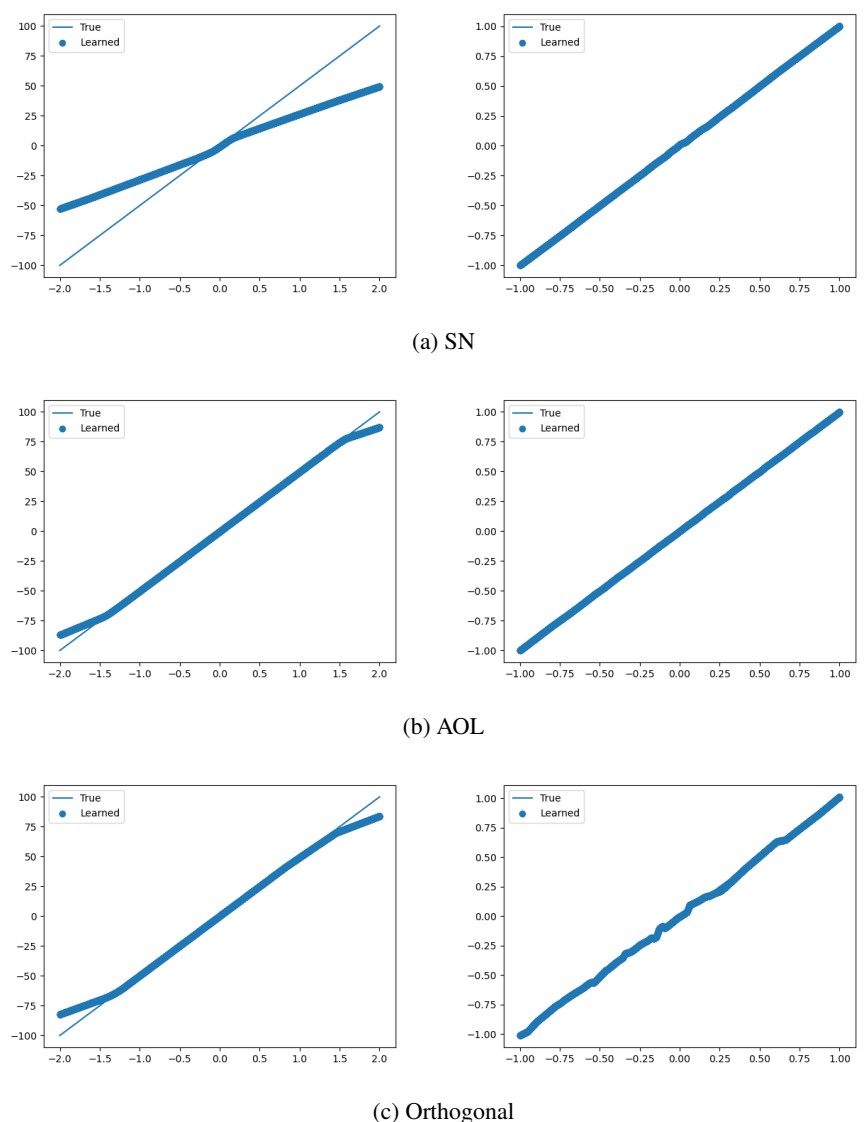

(a) SN

(b) AOL

(c) Orthogonal

Figure 17: Results of fitting a linear function ($y = 50x$ for left column and $y = x$ for right column) with SN (first row), AOL (second row) and Orthogonal (third row) with a specified Lipschitzness of 50.

to scale the function by multiplying with $L$. However, our method is adding a regularization term without touching the core function. This is an interesting avenue for future research.

In short, our model can perfectly learn both the function with slope 50 and 1 even though we greatly overestimated the Lipschitz constant.

### G.4   Summary of the Two Previous Experiments

This experiment summarizes the two previous experiments. We propose to run a regression task of the function $y = 50x$ with SN (representing the layer-wise methods) and our model, where the Lipschitz bound $L$ is changed from 25 to 125. If the model provides tight bounds and perfect expressive power, then the loss should equal 0 for $L \geq 50$ as theoretically $y = 50x$ can be reconstructed, and should increase once $L$ becomes smaller than 50 as the maximal slope we can reproduce is limited

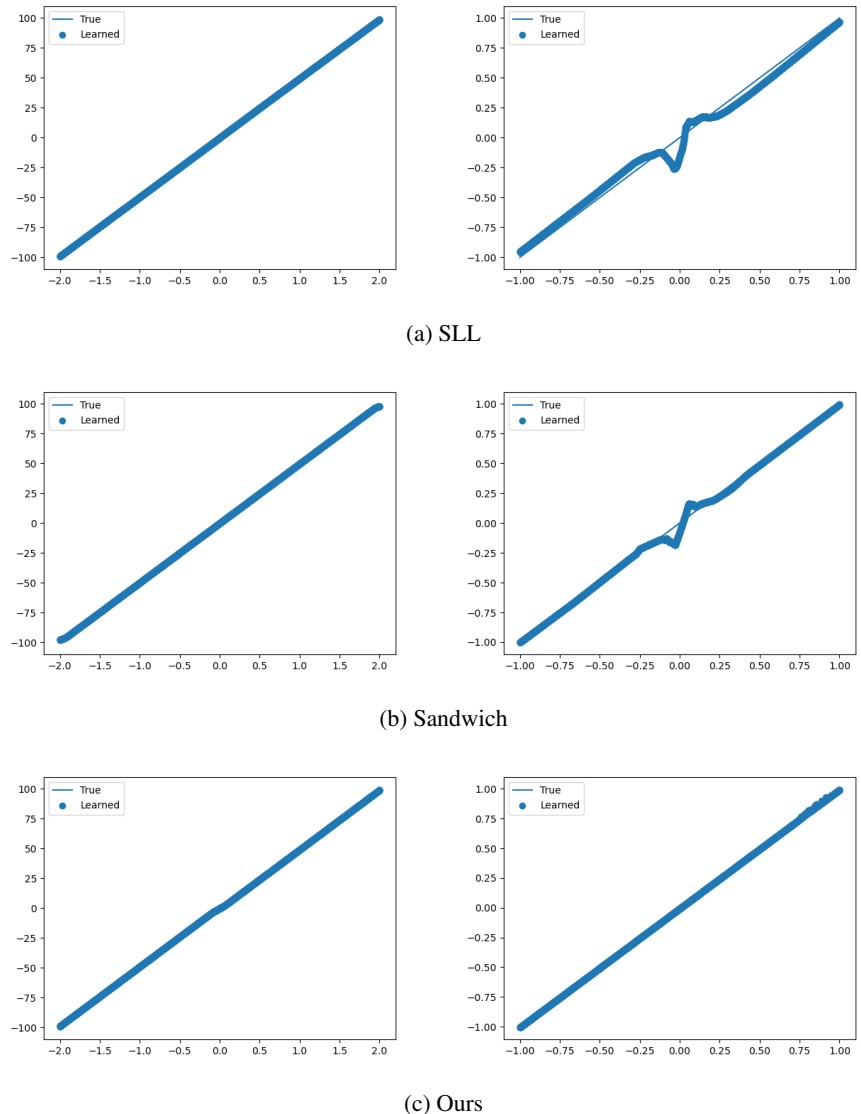

(a) SLL

(b) Sandwich

(c) Ours

Figure 18: Results of fitting a linear function ($y = 50x$ for left column and $y = x$ for right column) with SLL (first), Sandwich (second row) and our method (third row) with a specified Lipschitzness of 50.

to $L$. Moreover, we expect that the optimization should proceed faster when $L$ is near $50$ as the initialization of the function should also be close to $y = 50x$. Results are shown in Figure 21.

As we can observe, SN, a layer-wise model known to provide conservative bounds, only achieves a 0 loss from around $L = 100$, while ours exactly exhibits an increase of the loss from $L = 50$. This clearly shows that a layer-wise model has an expressive power that is largely lower than expected in the initial works. Notably, the convergence speed drastically decreases around $L = 50$ for our approach. This suggests choosing bi-Lipschitz constants as tight as possible tailored to the specific problem to ensure faster convergence.

Interestingly, this difference of behavior around the true Lipschitz constant of our model could provide a new method to estimate the bi-Lipschitz constants. Indeed, by starting with a high Lipschitz constant $\alpha + \beta$ and decreasing it, we can find the largest point where the loss starts to increase. This value can become a good estimation of the Lipschitzness of the true function. We can similarly proceed for inverse Lipschitzness. We leave further investigation for future work.

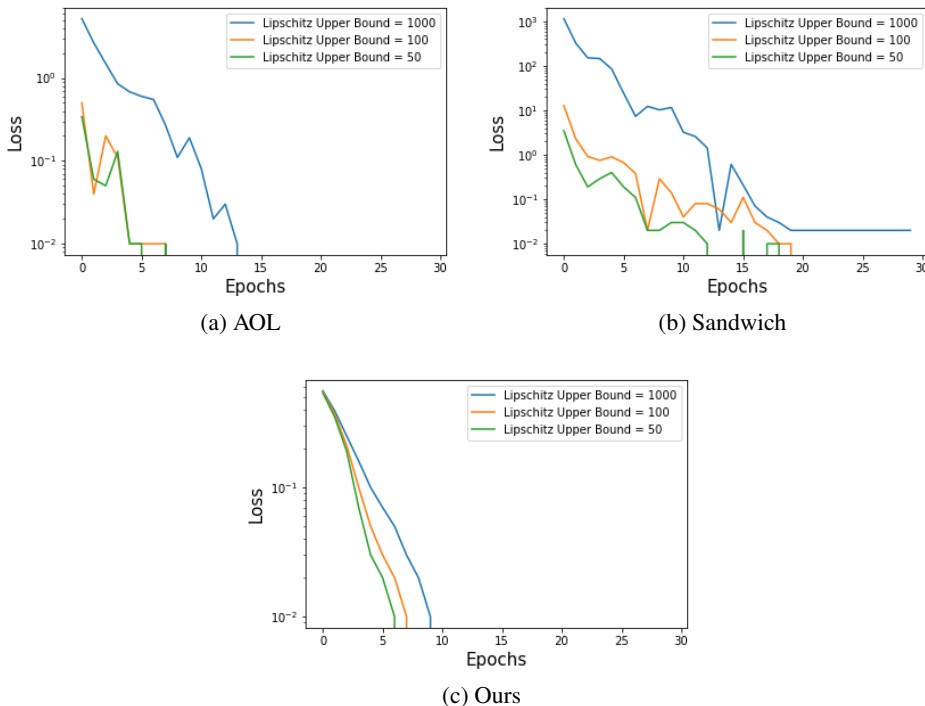

(a) AOL

(b) Sandwich

(c) Ours

Figure 19: The evolution of the loss with different upper bound constraints on the Lipschitz constant with AOL (top left), Sandwich (top right) and our method (bottom).

### G.5    Better Control for Annealing

Since our model provide tight bounds and good regularization performance as shown in the precedent experiments, the annealing of its parameters could be effective in some suitable settings. Let us consider the case where we want to learn the exponential function. It is not Lipschitz in general since its slope is monotonically diverging to $\infty$ as $x \to \infty$ and not inverse Lipschitz either since $e^x \to 0$ as $x \to -\infty$. We will start by a low Lipschitz parameter of the model and increase it by evaluating its effective Lipschitzness during training. The annealing will proceed in an elastic way. If the estimation is close to the upper bound then we will relax the bound since it means that our bound is too strong. That way, if we have a tight bound, the annealing process should progress faster and the overall function should rapidly converge to $\exp(x)$. The inverse Lipschitz constant is set to 0 and kept fixed. We compared our model with the Sandwich model. See Figure 22. We can verify that the annealing process is faster than the layer-wise model. Ours constantly wants to increase the Lipschitz constant while the Sandwich layer abandons at some point and cannot keep up the increase of the Lipschitzness.

### G.6    Uncertainty Estimation

#### G.6.1    Two Moons

This subsection provides additional results of experiment 4.2 of the main paper. As a reminder, we compare the uncertainty when learning the two moons dataset with not only DUQ and DUQ+BLNN but also the Deep Ensembles method and a DUQ with no bi-Lipschitz regularization. Results are plotted in Figure 23. Uncertainty is calculated as the distance to the closest centroid (see Appendix E for the mathematical formulation). Blue indicates high uncertainty, and yellow low uncertainty.

As mentioned in the main paper, DUQ+BLNN achieves a tighter bound which is possible because we are able to tune the bi-Lipschitz constant unlike DUQ. The experimental setup for DUQ was chosen like the original paper (Van Amersfoort et al., 2020).

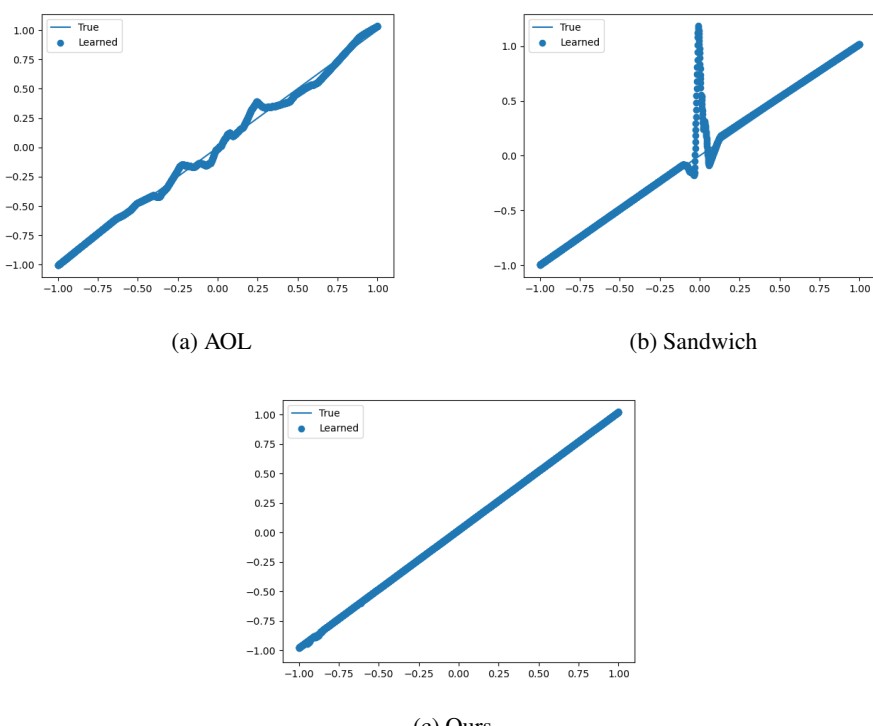

Figure 20: Results of fitting the linear function $y = x$ with AOL (top left), Sandwich (top right) and our method (bottom) with a specified Lipschitzness of 1000.

Table 6: Uncertainty quantification of two moons dataset with DUQ+BLNN with different $\alpha$ and $\beta$. Mean and standard deviation over five different seeds.

| $\alpha\backslash\beta$ | 0.1 | 1.0 | 2.0 | 3.0 | 4.0 | 5.0 |
|---|---|---|---|---|---|---|
| 0.0 | 0.716±.111 | 0.874±.006 | 0.904±.045 | 0.990±.002 | 0.972±.043 | 0.957±.048 |
| 1.0 | 0.879±.004 | 0.991±.002 | 0.994±.004 | 0.992±.004 | 0.895±.195 | 0.993±.007 |
| 2.0 | 0.878±.006 | 0.990±.003 | 0.993±.004 | 0.936±.087 | **0.995±.003** | 0.518±.047 |
| 5.0 | 0.503±.006 | 0.500±.000 | 0.541±.080 | 0.503±.004 | 0.501±.002 | 0.503±.004 |

Table 6 provides details on the accuracy of the learned model with different $\alpha$ and $\beta$, used for the grid search. As for the accuracy, the value of $\alpha$ does not matter as long as $\beta$ is chosen large enough. Indeed, a BLNN with $(\alpha, \beta) = (0.0, 3.0)$ theoretically includes that with $(\alpha, \beta) = (1.0, 2.0)$. That is why they exhibit similar performance. Nevertheless, a too loose setting of these parameters, especially for $\beta$, still seems to affect the training, and a as tight as possible choice is beneficial as we can see for $(\alpha, \beta) = (2.0, 5.0)$. In a sense, decreasing $\beta$ which is mainly in charge of the Lipschitz constant helps better generalization as pointed out in prior work as well (Bartlett et al., 2017). In the same direction, a too large value of $\alpha$ leads to a nonsensical result as we can understand when $\alpha = 5$. In this case, the model is randomly guessing which means that it is unsure everywhere. This is also translated in the uncertainty estimation as shown in Figure 24. The area where the model is unsure increases as $\alpha$ is increased, meaning that it becomes more sensitive to unknown data (see Figure 25). Comparatively, setting $\alpha = 0$ does not affect so much since the injectivity constraint is already beneficial in this type of task. Finally, a too low value of $\beta$ means that the inverse Lipschitz and Lipschitz constants are close to each other and the function behaves more like a linear function. Consequently, learning becomes difficult as we can conclude from the table with $\beta = 0.1$. Similar trends could be also observed in more complex tasks such as FashionMNIST vs MNIST and FashionMNIST vs NotMNIST.

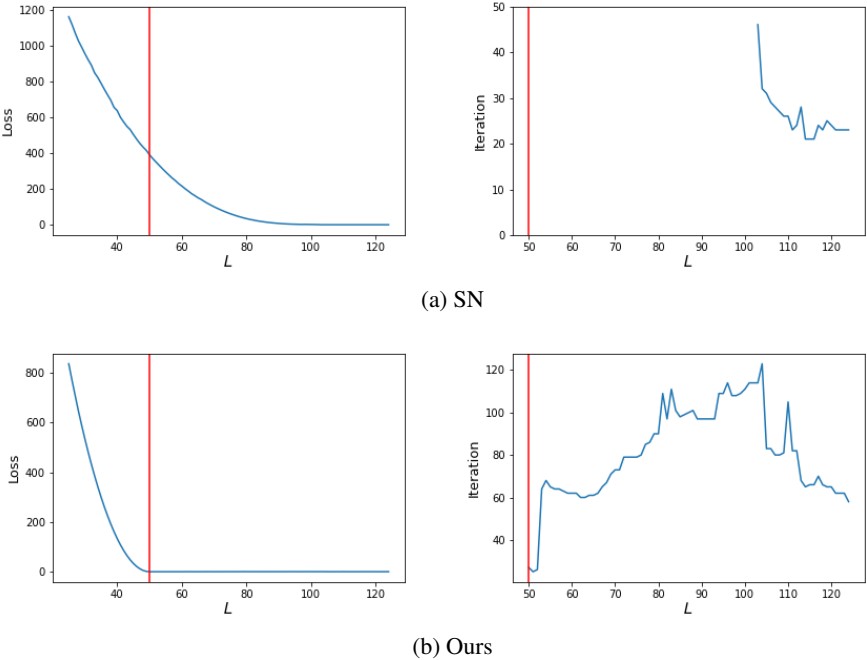

(a) SN

(b) Ours

Figure 21: Results of fitting the linear function $y = 50x$ with different $L$ for SN and our model. The left figure is the loss in function of $L$, and the right figure is the first iteration that the loss was below $0.5$ for each $L$ (if there is no point, it means that a loss below $0.5$ was never reached for this value of $L$). The red line emphasizes the point of $L = 50$.

Table 7: Out-of-distribution detection task with down-sampled data. FashionMNIST vs MNIST and FashionMNIST vs NotMNIST dataset with DUQ and DUQ+BLNN. For the BLNN, $\alpha = 0.2$ and $\beta = 0.4$. Mean and standard deviation over five trials.

| Model | Accuracy | BCE (loss function) | AUROC MNIST | AUROC NotMNIST |
|---|---|---|---|---|
| DUQ | 0.8564±.0066 | 0.078±.004 | 0.870±.040 | 0.852±.010 |
| DUQ+BLNN | **0.8595±.0032** | 0.078±.004 | **0.876±.008** | **0.960±.009** |

### G.6.2 FashionMNIST

This subsection provides additional results of experiment 4.2 of the main paper. In this experiment, we use real world data of FashionMNIST (Xiao et al., 2017), MNIST (LeCun and Cortes, 2010) and NotMNIST (Bulatov, 2011). The task is to learn to classify FashionMNIST but at the same time to detect out-of-distribution points from MNIST and NotMNIST. This task to distinguish FashionMNIST from MNIST datasets is known to be a complicated task (Van Amersfoort et al., 2020). We compute the AUROC for the detection performance. Results for a downsampled dataset and full size dataset are shown in Table 7 and 8, respectively. A visualization of the ROC curve can be found in Figures 26 and 27.

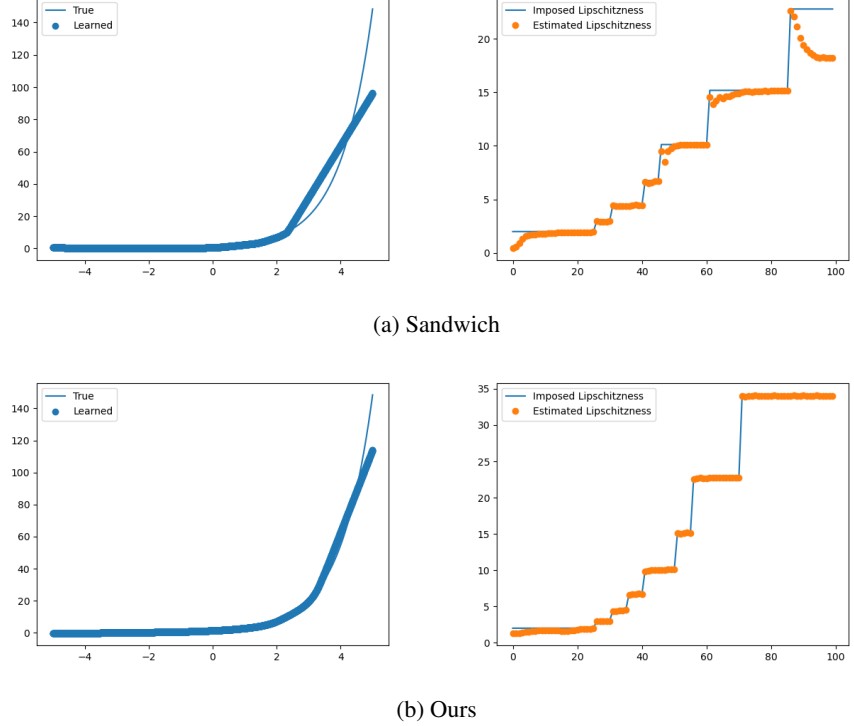

(a) Sandwich

(b) Ours

Figure 22: Results of fitting $y = \exp(x)$ with Sandwich (top row) and our method (bottom row). The right figures are the evolution of the upper bound and the estimated Lipschitz constant over the iteration, and the left column the learned function. The $x$-axis of the right pictures represents the iteration number.

Table 8: Out-of-distribution detection task with full size data. FashionMNIST vs MNIST and FashionMNIST vs NotMNIST dataset with DUQ and DUQ+BLNN. For the BLNN, $\alpha = 0$ and $\beta = 3.0$. Mean and standard deviation over five trials.

| Model | Accuracy | BCE (loss function) | AUROC MNIST | AUROC NotMNIST |
|---|---|---|---|---|
| DUQ | 0.8893±.0037 | 0.064±.005 | 0.862±.035 | 0.900±.024 |
| DUQ+BLNN | **0.8985±.0040** | **0.060±.000** | **0.896±.008** | **0.964±.005** |

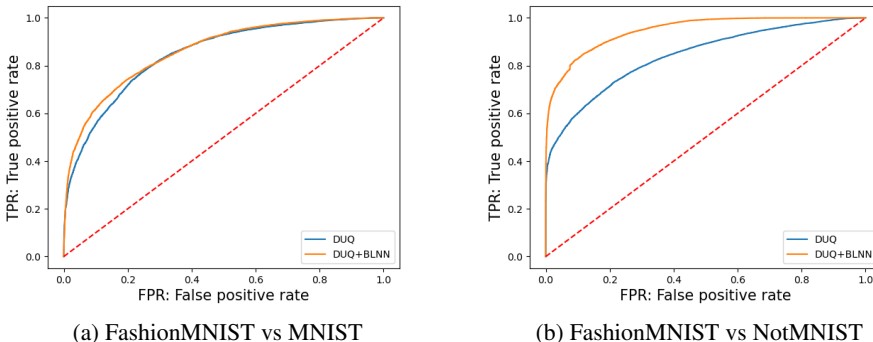

(a) FashionMNIST vs MNIST

(b) FashionMNIST vs NotMNIST

Figure 26: ROC between dwonsampled FashionMNIST vs MNIST (left) and FashionMNIST vs NotMNIST (right).

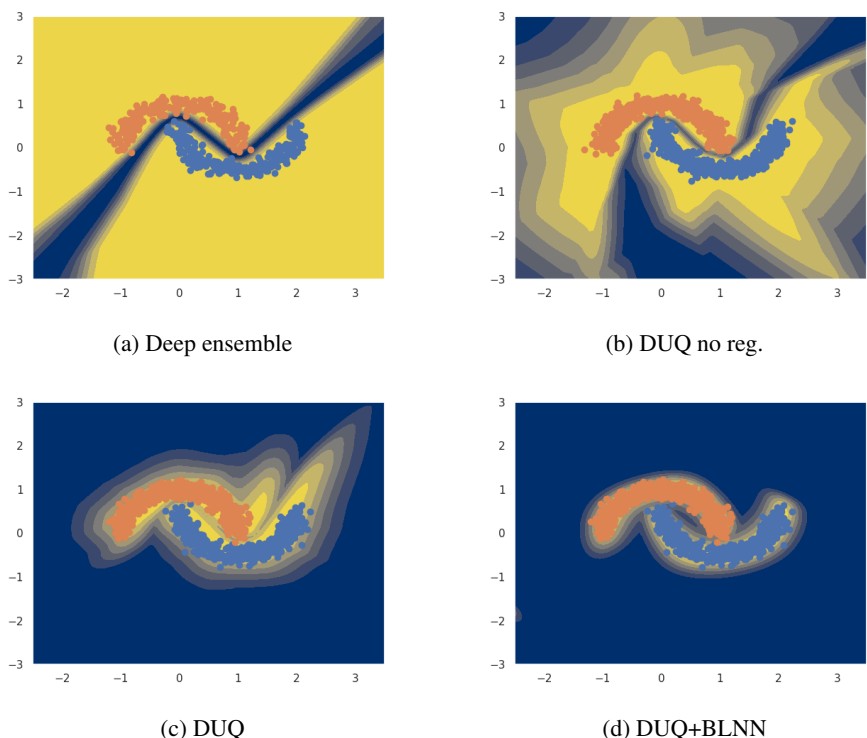

(a) Deep ensemble                               (b) DUQ no reg.

(c) DUQ                                    (d) DUQ+BLNN

Figure 23: Uncertainty estimation with the two moons data set. Left figure is with a simple neural network without any constraints, and the right is with our model constraining bi-Lipschitzness.

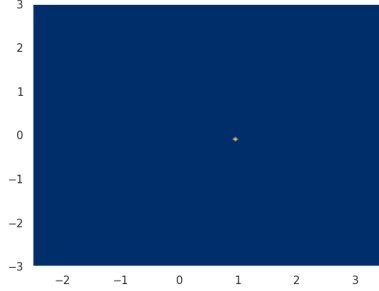

Figure 24: Uncertainty quantification of two moons dataset with DUQ+BLNN with a high $\alpha$. $\alpha = 5.0$ and $\beta = 3.0$. We do not show the points so that the highly certain area is visible.

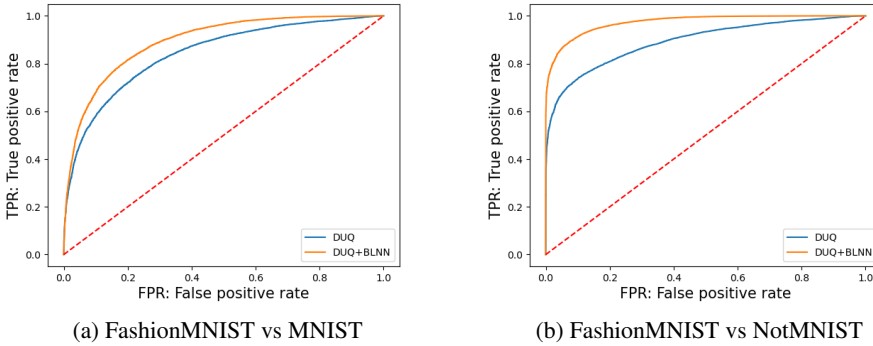

(a) FashionMNIST vs MNIST                    (b) FashionMNIST vs NotMNIST

Figure 27: ROC between full size FashionMNIST vs MNIST (left) and FashionMNIST vs NotMNIST (right).

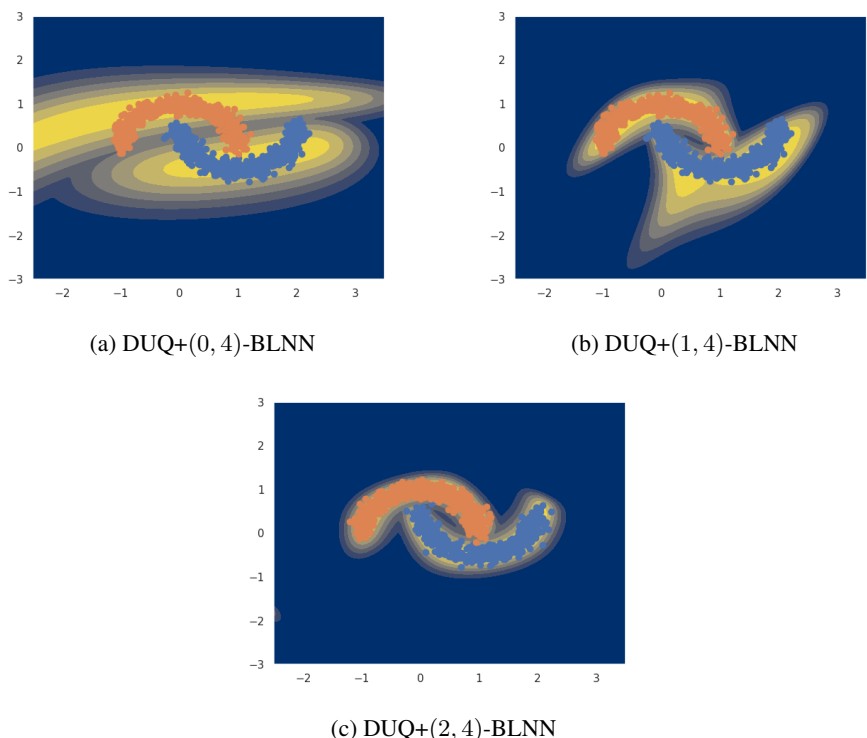

(a) DUQ+(0, 4)-BLNN

(b) DUQ+(1, 4)-BLNN

(c) DUQ+(2, 4)-BLNN

Figure 25: Uncertainty quantification of two moons dataset with DUQ+BLNN with different $\alpha$.

Table 9: Comparison of our model with state-of-the-art monotone models. Means and standard deviations over three trials. Results of prior models are from the original papers. C = COMPAS, BF = BlogFeedBack, LD = LoanDefaulter, HD = HeartDisease, AM = AutoMPG, Acc. = accuracy.

| Models | C (Acc.) | BF (RMSE) | LD (Acc.) | HD (Acc.) | AM (MSE) |
|---|---|---|---|---|---|
| Certified | $68.8 \pm 0.2$ | $0.158 \pm 0.001$ | $65.2 \pm 0.1$ | - | - |
| Constrained | $69.2 \pm 0.2$ | $0.154 \pm 0.001$ | $65.3 \pm 0.0$ | $89.0 \pm 0.0$ | $8.37 \pm 0.1$ |
| LMN | $69.3 \pm 0.1$ | $0.160 \pm 0.001$ | $65.4 \pm 0.0$ | $89.6 \pm 1.9$ | $7.58 \pm 1.2$ |
| SMNN | $69.3 \pm 0.9$ | $\mathbf{0.150 \pm 0.001}$ | $65.0 \pm 0.1$ | $88.0 \pm 4.0$ | $7.44 \pm 1.2$ |
| Ours | $\mathbf{69.4 \pm 0.6}$ | $0.157 \pm 0.001$ | $\mathbf{65.5 \pm 0.1}$ | $\mathbf{90.2 \pm 1.6}$ | $\mathbf{7.13 \pm 1.2}$ |

### G.7 Partially Monotone Settings

Table 9 provides the experimental results with full data. We also compare with the certified monotone neural network of Liu et al. (2020b) (Certified) and the constrained monotone neural network of Runje and Shankaranarayana (2023) (Constrained). See Tables 2 and 3 of Kim and Lee (2024) for a comprehensive comparison with other general models.

## H Experimental Setups

In this appendix, we provide further experimental details, including values of hyper-parameter, specific composition of used architectures and optimization schemes, which were omitted in the main text as a concern of clarity. All experiments were executed with Pytorch. We mainly used the GPU NVIDIA A100 with a memory of 80GB for the computation. Additional information for each experiments can be found in each subsection below and in the supplementary material.

Table 10: General details on the architectures of Figures 8 and 9.

| Model | Hidden Dimension | Number of Layers |
|-------|------------------|------------------|
| Ours  | 10               | 2                |

## H.1 Computational Complexity Comparison

In this experiment, corresponding to Figure 2, we presented a comparison of the computational complexity for a single iteration between a traditional feedforward network and various BLNN variants.: BLNN (with brute force backpropagation), BLNN with Theorem 3.7, PBLNN with only one variable constrained to be bi-Lipschitz. Respective parameter sizes are 1.11M, 1.42M, 1.42M and 1.38M. The input size was randomly generated data of size $3 \times 32 \times 32$, simulating a CIFAR dataset, and the batch size was varied in the set 1, 5, 25, 75, 100. For this experiment, BLNN was implemented according to the template of the DEQ library [4]. FLOPs was computed following another github library [5]. The feedforward neural network had ReLU activation function and 4 layers with hidden dimension 210, and the BLNN variants had softplus activation function and 3 layers with hidden dimension 150. $\alpha$ and $\beta$ were both set to 1.

## H.2 Algorithms for LFT

### H.2.1 Influence of Approximate Optimization on Bi-Lipschitz Constants: Experiments

In this experiment, we were interested in the evolution of the bi-Lipschitz constants under different optimization schemes.

**Corresponding Figures and Tables**   Figures 8 and 9.

**Data**   We generated 5000 2-dimensional points uniformly sampled at random in the interval $[-1, 9] \times [-1, 9]$.

**Architecture**   Our BLNN was created with $\beta = 10$. The value of $\alpha$ does not matter since we focused on the output of the optimization process which occurs before adding the regularization term $\alpha/2\|x\|^2$. We used the softplus function as the activation function. The effective Lipschitz and inverse Lipschitz constants were estimated by a simple sampling with sample size 5000 chosen from the data.

**Optimization Schemes**   We used several optimization schemes, namely, steepest gradient descent (GD), Nesterov's accelerated gradient (AGD) (Nesterov, 1983), Adagrad (Duchi et al., 2011), RM-Sprop (Hinton et al., 2012), Adam (Kingma and Ba, 2015) and the Newton method For Adagrad, RMSprop and Adam, all parameters were set as default except the learning rate.

### H.2.2 On the Choice of the Initial Point

In this experiment, we were interested in the influence of the initial point on the convergence speed of the Legendre-Fenchel transformation. We used as a starting point either the final point of the previous epoch for each training data or the point $(1, \ldots, 1)^\top$ independently of the history. This experiment was run with the architecture explained in Subsection H.4.2.

**Corresponding Figures and Tables**   Figure 10.

## H.3 Bi-Lipschitz Control

The codes were implemented in Python 3.11.4 with PyTorch 2.0.1+cu117.

---

[4] https://github.com/locuslab/deq
[5] https://github.com/MrYxJ/calculate-flops.pytorch

### H.3.1 Simple Estimation of Bi-Lipschitz Constants at Initialization

In this experiment, we were interested in verifying in more detail whether an $(\alpha, \beta)$-BLNN respects the pre-defined bounds of bi-Lipschitzness, namely, $\alpha$ and $\alpha + \beta$. We used GD for the computation of the Legendre-Fenchel transformation since it showed good performance and practical usefulness in the previous sections.

**Corresponding Figures and Tables**   Figures 11, 12, 13, 14 and 15.

**Data**   We randomly created 200 2-dimensional points, and estimated the effective bi-Lipschitz constants with them.

**Architecture**   For a fixed activation function (ReLU or softplus) and number of hidden layers (3 or 10), we created 100 architectures with different $\alpha$ and $\beta$. $\alpha$ was set to 4 for all 100 models, and $\beta$ as $0.05 + 99.95/100i$ where $i \in \{0, \ldots, 99\}$. The value of the hidden dimension was set to 10.

### H.3.2 Tightness of the Bounds: Underestimation Case

In this experiment, we were interested in the tightness of the Lipschitz bound provided by each framework. The experiment was inspired from Figure 3 of Wang and Manchester (2023). Codes were also imported and modified from their github [6] combined with others[7][8]

**Corresponding Figures and Tables**   Tables 1 and 5 and Figures 3 and 16.

**Data**   We created 300 training points $(x, f(x))$, where $x$ was sampled at random in the interval $[-2, 2]$. The test dataset consisted of 2000 points, with $x$ in the interval $[-1, 1]$.

**Optimization**   Based on Wang and Manchester (2023), we used the Adam optimizer with a learning rate of 0.01, and other parameters were set as default. The objective function was the mean squared error.

**Architecture**   In addition to our model, we used seven layer-wise Lipschitz architectures, namely, spectral normalization (Miyato et al., 2018), AOL (Prach and Lampert, 2022), Orthogonal (Trockman and Kolter, 2021), SLL (Araujo et al., 2023), Sandwich (Wang and Manchester, 2023), LMN (Nolte et al., 2023) and BiLipNet (Wang et al., 2024). See Appendix A for the mathematical formulation of each Lipschitz approach. Each layer was designed to be 1-Lipschitz, and we employed ReLU as activation function for all of them, including ours, except that for LMN. Since the Lipschitz constant of the overall network is 1, we included a scaling factor before the first layer which multiplies the input by $L$. As a result, we obtain $L$-Lipschitz neural networks. In the experiments, $L$ was set to 2, 5, 10 and 50. For our model, $L$ corresponds to $\alpha + \beta$. We set $\alpha = 1$ and changed $\beta$ accordingly. See Table 11 for further details on the architecture. The effective Lipschitz constant was estimated by a simple sampling with sample size 1000 within the range $[-1, 1]$. As for BiLipNet, we only used their monotone Lipschitz layer and no orthogonal layer in order to avoid losing tightness by composing many simple layers.

### H.3.3 Flexibility of the Model: Overestimation Case

In this experiment, we were interested in the influence of the overestimation of the Lipschitz constant when building the model. The dataset, architecture and optimization scheme were mostly the same as H.3.2. We just changed the function to be fitted to a linear one with slope 1 and 50. The architectures were designed to have a Lipschitz bound $L$ of 50, 100 or 1000.

**Corresponding Figures and Tables**   Figures 4, 17, 18, 19, 20.

---

[6]https://github.com/acfr/LBDN
[7]https://github.com/ruigangwang7/StableNODE
[8]https://github.com/niklasnolte/monotonic_tests

Table 11: General details on the architectures of Table 5 and Figure 16.

| Model | Hidden Dimension | Number of Layers | Size of Model |
|---|---|---|---|
| SN | 45 | 4 | 4.3K |
| AOL | 45 | 4 | 4.3K |
| Orthogonal | 45 | 4 | 4.3K |
| SLL | 45 | 2 | 4.2K |
| Sandwich | 65 | 2 | 4.5K |
| LMN | 64 | 3 | 4.4K |
| BiLipNet | 40 | 2 | 5.0K |
| Ours | 64 | 2 | 4.3K |

### H.3.4 Summary of the Two Previous Experiments

In this experiment, we were interested in the relation between the imposed Lipschitz upper bound and the loss function when learning the function $y = 50x$. The dataset, architecture and optimization scheme were mostly the same as the previous section. We just changed the activation function of the BLNN from ReLU to softplus. We also reported the first time when the loss reached a value below 0.50.

**Corresponding Figures and Tables**   Figures 1 and 21.

### H.3.5 Better Control for Annealing

In this experiment, we were interested in the behavior of our model under an annealing scheme. The dataset, architecture and optimization scheme were mostly the same as H.3.2. We just changed the function to be fitted to $y = e^x$. Moreover, we set $\alpha = 0$ for our model. The training was executed for 200 epochs.

**Corresponding Figures and Tables**   Figure 22

**Annealing**   We employed a simple annealing scheme. We started with a Lipschitz constant of $\gamma = 2$. For the Sandwich model, it is the scaling factor at the input, and for ours, it is $\alpha + \beta = \beta$ since $\alpha = 0$. We estimated the Lipschitz constant at each iteration, and if the effective Lipschitzness was close to the imposed one, i.e., $\gamma$, with an accuracy of 0.05, we relaxed the condition by multiplying $\gamma$ by 1.5. This update was executed every five epochs, if the condition was satisfied, in order to provide to the model the time to learn a new appropriate representation.

## H.4 Uncertainty Estimation

In this chapter, the experiments were inspired from Figure 3 of Van Amersfoort et al. (2020). Codes were also imported and modified from their github. [9] The codes were implemented in Python 3.10.13 with PyTorch 2.0.1+cu117.

### H.4.1 Two Moons

In this experiment, we were interested in the uncertainty estimation when learning the two moons dataset. We compared the performance of our model with other existing models.

**Corresponding Figures and Tables**   Figures 5, 23, 24, 25 and Table 6.

**Data**   The two moon dataset was generated using the sklearn toolkit with a noise of 0.1. We used 1500 points for training and 200 for tests. Batch size was 64.

**Optimization**   We used SGD with a learning rate of 0.01, momentum of 0.9 and weight decay of $10^{-4}$. We used the binary cross entropy as a loss function.

---

[9] https://github.com/y0ast/deterministic-uncertainty-quantification

Table 12: General details on the architectures of Figure 23."reg." stands for regularization.

| | Hidden Dimension | Num. of Layers | Output Dim. | Centroid Size | Num. of Parameters |
|---|---|---|---|---|---|
| DUQ | 40 | 3 | 40 | 10 | 4.2K |
| DUQ no reg. | 40 | 3 | 40 | 10 | 4.2K |
| DUQ+BLNN | 20 | 2/2 | 40 | 10 | 3.4K |
| Deep Ensembles | 20 | 4 | 2 | - | $0.9K \times 5$ |

Table 13: General details on the architectures of Table 7 and Figure 26. For DUQ+BLNN, since the input and output dimensions were not equal we used the architecture explained in Extension 1 of Appendix F. While these two parts share almost the same structure, we clarify the differences by the following notation "info. of first BLNN/info. of second BLNN".

| | Hidden Dimension | Num. of Layers | Output Dim. | Centroid Size | Num. of Parameters |
|---|---|---|---|---|---|
| DUQ | 49 | 3 | 256 | 100 | 280.9K |
| DUQ+BLNN | 40 | 3/1 | 256 | 100 | 293.7K |

**Architecture** We used four models, namely, a deep ensemble method (Lakshminarayanan et al., 2017), the original DUQ, DUQ with no regularization and DUQ with our method (DUQ+BLNN). The latter means that we used the DUQ framework but we deleted the gradient penalty of the loss function and directly replaced the vanilla neural network with our BLNN. We used the softplus function for all architectures. See Table 12 for further details on the architecture. As for our model, the best values of $\alpha$ and $\beta$ were found through a grid search: $\alpha = 2$ and $\beta = 4$. Results with other $\alpha$ and $\beta$ are shown in Table 6.

### H.4.2 Fashion-MNIST (downsampled)

In this experiment, we were interested in applying BLNN to the task of out-of-distribution detection of FashionMNIST vs MNIST and FASHIONMNIST vs NotMNIST datasets, and comparing its performance with existing models. This is a downsampled version of the next experiment.

**Corresponding Figures and Tables** Tables 2, 7 and Figure 26.

**Data** The FashionMNIST dataset was generated using the torchvision toolkit, with 60000 training data and 2000 test data. Batch size was set to 128. The dataset was downsamlped by a pooling data from $28 \times 28$ to $14 \times 14$.

**Optimization** We used SGD with a learning rate of 0.05, momentum of 0.9 and weight decay of $10^{-4}$. We used the binary cross entropy as a loss function.

**Architecture** We compared two models DUQ and DUQ with our method (DUQ+BLNN). The latter means that we used the DUQ framework but we deleted the gradient penalty of the loss function and directly replaced the vanilla neural network with our BLNN. We used the softplus function for DUQ+ours and ReLU for the other architectures. See Table 13 for further details on the architecture. As for our model, the best values of $\alpha$ and $\beta$ were found through a grid search: $\alpha = 0.2$ and $\beta = 0.4$. For both models, data was down-sampled by a max-pooling layer from $28 \times 28$ to $14 \times 14$.

### H.4.3 Fashion-MNIST (full size)

In this experiment, we were interested in applying BLNN to the task of out-of-distribution detection of FashionMNIST vs MNIST and FASHIONMNIST vs NotMNIST datasets, and comparing its performance with existing models. This is the full size version of the previous experiment.

**Corresponding Figures and Tables** Table 8 and Figure 27.

Table 14: General details on the architectures of Table 8 and Figure 27.

| Model | Hidden Dimensions | Num. of Layers | Output Dim. | Centroid Size | Num. of Parameters |
|-------|-------------------|----------------|-------------|---------------|---------------------|
| DUQ | 196/49 | 3 | 256 | 100 | 432.3K |
| DUQ+BLNN | 150 | 3 | 50 | 100 | 449.2K |

**Data** The FashionMNIST dataset was generated using the torchvision toolkit, with 60000 training data and 2000 test data. Batch size was set to 128. We used the original size of the dataset, namely $28 \times 28$.

**Optimization** We used SGD with a learning rate of 0.05, momentum of 0.9 and weight decay of $10^{-4}$. We used the binary cross entropy as a loss function.

**Architecture** We compared two models DUQ and DUQ with our method (DUQ+BLNN). The latter means that we used the DUQ framework but we deleted the gradient penalty of the loss function and directly replaced the vanilla neural network with our BLNN. We used the softplus function for DUQ+ours and ReLU for the other architectures. See Table 14 for further details on the architecture. As for our model, the best values of $\alpha$ and $\beta$ were found through a grid search: $\alpha = 0.0$ and $\beta = 3.0$. Particularly, we only used one BLNN in this experiment compared to the previous one, and projected it to a smaller sub-space of size 50 by a diagonal rectangular matrix with all components set to 1.

### H.5 Partially Monotone Settings

In this experiment, we were interested in applying PBLNN to several tasks where data exhibited monotone behaviors with respect to some variables. The codes were imported and modified from (Nolte et al., 2023).[10] They were implemented in Python 3.10.14 with PyTorch 2.0.1+cu117.

**Corresponding Figures and Tables** Table 3.

**Data** COMPAS (Angwin et al., 2016) is a classification task with 13 features, where 4 of them have monotone inductive bias. It is important to highlight that this dataset is known to be racially biased (Angwin et al., 2016). BlogFeedBack (Nolte et al., 2023) is a prediction task with 276 features, where 8 of them have monotone bias. LoanDefaulter (Nolte et al., 2023) a classification task with 28 features, where 5 of them have monotone inductive bias. HeartDisease (Janosi et al., 1988) is also a classification task with 13 features, including 2 with monotone inductive bias. AutoMPG (Quinlan, 1993) contains 7 features where 2 of them shows monotone inductive bias. CIFAR101 is an augmented dataset used by Nolte et al. (2023).

**Optimization** We used the same training scheme as Nolte et al. (2023).

**Architecture** We compared our model, the PBLNN, with that of Nolte et al. (2023) and that of Kim and Lee (2024). As for the former, we constructed the PBLNN based on the inductive bias. For HeartDisease and AutoMPG, we took as the final output the average of the output of one PBLNN with bi-Lipschitzness imposed on the monotone variables. For COMPAS and LoanDefaulter, we created independent 1-dimensional PBLNNs whose bi-Lipschitzness is imposed on each variable that are monotone and then took the average. For CIFAR101, we used 101 PBLNNs with respect to the last variable. See Table 15 and the codes for further details.

---

[10]https://github.com/niklasnolte/monotonic_tests

Table 15: General details on the architectures of PBLNN used for Table 3.

| Dataset | Hidden Dimensions | Num. of Layers | Lipschitzness | Inverse Lipschitzness |
|---|---|---|---|---|
| COMPAS | $45\times4$ | 4 | 0.3 | 0 |
| BlogFeedBack | $3\times 8$ | 2 | 1 | 0 |
| LoanDefaulter | $10\times 5$ | 2 | 0.5 | 1 |
| HeartDisease | 10 | 1 | 10 | 2 |
| AutoMPG | 20 | 3 | 1 | 0 |
| CIFAR101 | $10\times101$ | 3 | 1 | 1 |

