# OpenReview forum: "A provable control of sensitivity of neural networks through a direct parameterization of the overall bi-Lipschitzness"
_NeurIPS.cc/2024/Conference — NeurIPS 2024 poster_

### Official Review · Reviewer_Hqwz · 2024-07-12

**Soundness:** 3
**Presentation:** 3
**Contribution:** 3
**Rating:** 6
**Confidence:** 3

**Summary:**

This paper investigates and proposes a novel bi-Lipschitz neural network architecture. This architecture provides a simple, direct and tight control of the Lipschitz and inverse Lipschitz constants through the use of two parameters, the ideal minimum, equipped with theoretical guarantees. To devise their architecture the authors exploit convex neural networks and the Legendre-Fenchel duality. The authors also propose a variant of their bi-Lipschitz architecture that is more scalable by exploiting partially input convex neural networks. Finally, the authors propose a set of experiments to showcase the utility of our model in concrete machine learning applications, namely, uncertainty estimation and monotone problem settings and show that it can improve previous methods.

**Strengths:**

- The paper is well written. After writing a clearly structured related work (with an extensive background and related work proposed in Appendix A), the authors propose their new design and explicitly explain how the forward pass of their network is computed as well as the expressivity and how the backpropagation can be done.
- The authors acknowledge that the computational cost of their approach can pose serious limitation and propose to overcome this problem with partially input convex neural networks.

**Weaknesses:**

- It would be interesting of the authors could provide experiments with both their architectures with respect to computational cost, and highlight time of training etc.

**Questions:**

NA.

**Limitations:**

Lack of experiments wrt to computational cost and maybe an experiment with a more larger dataset than the current ones use in the paper.

---

> ### Author Rebuttal · Authors · 2024-08-06
>
> Thank you very much for spending your time on carefully reviewing our paper. We really appreciate all the advice you provide to improve our paper. Please find below answers to your questions. We also summarized the comparison of the time and space complexity of our models in an independent thread.
>
> > It would be interesting of the authors could provide experiments with both their architectures with respect to computational cost, and highlight time of training etc.
>
> Thank you very much for this advice. Since a few reviewers were also interested in this topic, both theoretical and experimental discussions are summarized in the global rebuttal. We would appreciate it if you could verify this. This will be added in the updated version of our paper. Please note that we did not directly compare training times in seconds but instead compared the number of floating-point operations (FLOPs) for each architecture, as training time is heavily dependent on factors such as the machine, code, and libraries used.
>
> > maybe an experiment with a more larger dataset than the current ones use in the paper.
>
> We conducted an additional experiment using the convolutional version of our model (BLNNconv) on the problem of uncertainty estimation (as in Subsection 4.2) with the CIFAR-10 vs. SVHN dataset to illustrate the scalability of our method. For this problem, we implemented the model so that we first process the data through BLNNconv and then transfer it to the DUQ. The result compared to DUQ is as below. Our model is not only scalable to large-scale networks but also improves out-of-detection performance (the AUROC of SVHN). Using BLNNconv instead of the fully connected BLNN also improved the computation time (e.g., 2.5 times faster for the 5 first iterations).
>
> |      | Accuracy  | Loss | AUROC SVHN |
> |-------|-----------|----------|-----------|
> | DUQ   | 0.929    | 0.04 | 0.83|
> | BLNNconv | 0.930    | 0.04 | **0.89**   |
>
> We hope we addressed all your questions and concerns. We will be glad to provide further explanation and clarification if necessary.

---

> > ### Comment · Reviewer_Hqwz · 2024-08-10
> >
> > Thank you for the rebuttal. I believe this paper to provide a good contribution and maintain my score.

---

> ### Author Response · Authors · 2024-08-12
>
> Thank you very much for considering our rebuttal! We greatly appreciate your positive position on the contributions of our paper.

---

### Official Review · Reviewer_XcLq · 2024-07-15

**Soundness:** 4
**Presentation:** 3
**Contribution:** 3
**Rating:** 6
**Confidence:** 3

**Summary:**

This paper proposes a novel neural network architecture called BLNN (Bi-Lipschitz Neural Network) that allows direct control and parameterization of the overall bi-Lipschitzness of the network. The main contributions include: i) a framework that allows tight control of Lipschitz and inverse Lipschitz constants of networks via using convex neural networks and the Legendre-Fenchel transformation, ii) comprehensive theoretical analysis, iii) empirical evaluation showing the nice performance of BLNN on tasks like function fitting, out-of-distribution detection, and monotone regression.

**Strengths:**

**Originality:**

The paper presents a novel approach to constructing bi-Lipschitz neural networks that is distinctly different from existing methods. The use of convex neural networks and Legendre-Fenchel transformation to directly parameterize overall bi-Lipschitzness is quite novel. The extension (e.g. partially bi-Lipschitz networks, etc) is also new.


**Quality:**

The quality of the paper is good. The authors provide detailed proofs and analyses for their key claims, including the bi-Lipschitz properties of their construction and the expressive power of the resulting networks. The experiments cover various scenarios, from simple function fitting to uncertainty estimation and monotone regression. The results are quite competitive.


**Clarity:**

The paper is generally well-structured and clearly written. However, given the technical nature and the length of the paper, understanding the paper fully is still a tough task.

**Significance:**

The paper's contributions are significant in its solid theoretical developments. The significance is further underscored by the improved performance on tasks like out-of-distribution detection and monotone function learning. In conclusion, this paper presents a novel approach to an important problem in deep learning theory and practice.

**Weaknesses:**

1. Computational Complexity: A detailed analysis of time and space complexity compared to traditional networks can be helpful.

2. Scalability and Practical Implications: There's insufficient exploration of how the method scales to very large networks or complex datasets (e.g. TinyImageNet).

3. Hyperparameter Sensitivity: More discussions on this issue will be beneficial.

4. The paper could be more explicit about scenarios where the theoretical guarantees might not hold, and could explore potential extensions to other network architectures beyond feedforward networks.

**Questions:**

1. How does the proposed method perform on larger, more complex datasets like TinyImageNet or ImageNet?

2. Can the authors clarify the computational complexity of their approach?

3. Can the authors provide a more comprehensive study on hyperparameter sensitivity?

4. Can the authors comment on other network structures (e.g. implicit models, DEQs, etc)?

**Limitations:**

It seems that an improved discussion on potential negative societal impacts or broader ethical considerations is still missing.

---

> ### Author Rebuttal · Authors · 2024-08-06
>
> Thank you very much for spending your time on carefully reviewing our paper. We really appreciate all the advice you provide to improve our paper. Please find below answers to your questions. We also summarized the comparison of the time and space complexity of our models in an independent thread.
>
> (W=Weakness, Q=Question)
>
> >W1+Q2
>
> We apologize for omitting this important point. Please find in the global rebuttal both theoretical and experimental discussions on this topic. This will be added in the updated paper.
>
> > W2+Q1
>
> The main goal of our work was to create a new paradigm for the bi-Lipschitz control and to analyse its behavior. Nevertheless, we understand this is an important direction of exploration. We conducted an additional experiment using the convolutional version of BLNN (BLNNconv) on the problem of uncertainty estimation (as in Subsection 4.2) with the CIFAR-10 vs. SVHN dataset to illustrate the scalability of our method. We implemented the model so that we first process the data through BLNNconv and then transfer it to the DUQ. The result compared to DUQ is as below. Our model is not only scalable to large-scale networks but also improves out-of-detection performance (the AUROC of SVHN). Using BLNNconv instead of the fully connected BLNN also improved the computation time (e.g., 2.5 times faster for the 5 first iterations). Moreover, the amortization method of Reviewer u3yR may be a fundamental solution to the computational complexity of our model (which can already at least scale to CIFAR10) but out of the scope for this current work.
>
> | | Accuracy   | Loss | AUROC SVHN |
> |-|-|-|-|
> | DUQ   | 0.929    | 0.04 | 0.83|
> | BLNNconv | 0.930    | 0.04 | **0.89**   |
>
>
> > W3+Q3
>
> Thank you very much for this advice. Here, we would like to focus on the inverse Lipschitz and Lipschitz hyperparameters ($\alpha$ and $\beta$). Note that other hyperparameters such as batchsize, depth…, follow the basic properties of the core neural network ICNN, and therefore, not directly related to the goal of this work (see e.g., Schalbetter, Adrian. *Input convex neural networks for energy optimization in an occupied apartment*. MS thesis. ETH Zurich, 2020.).
>
> We would like to start the discussion by reminding that there is already an analysis with different $\alpha$ and $\beta$ for uncertainty estimation in the Appendix Table 5 p.50. We discuss in detail how the value of $\alpha$ and $\beta$ influence the performance of the model. The influence is rather intuitive, and we would like to verify this with additional experiments with a downscaled FashionMNIST for further clarity. The observations are as follows:
> 1. Changes in performance due to hyperparameter changes are continuous (no hyper-sensitivity).
> |   $\alpha$   | $\beta$   | Accuracy | Loss| AUROC MNIST| AUROC NotMNIST|
> |-|-|-|-|-|-|
> | 0.2  | 0.99    | 0.8600|0.08|0.84|0.95|
> | 0.2|1.0| 0.8630|0.08|0.84|0.95|
>
> 2. An increase of the domain of search leads to better performance due to higher expressivity.
> |   $\alpha$   | $\beta$   | Accuracy | Loss|
> |-|-|-|-|
> |0.0|0.1| 0.5680|0.24|
> |0.0|0.2 | 0.7485|0.14|
> |0.0|0.3 |0.7825|0.11|
>
> 3. But too loose smoothing leads to worse performance.
> |   $\alpha$   | $\beta$   | Accuracy | Loss| AUROC MNIST| AUROC NotMNIST|
> |-|-|-|-|-|-|
> |1.0|4.0| 0.1070|0.10|0.50|0.50|
> |1.0|1.0| 0.8455|0.11|0.81|0.98|
>
> 4. Too low sensitivity leads to worse out-of-distribution detection.
> |   $\alpha$   | $\beta$   | AUROC NotMNIST|
> |-|-|-|
> |0.0|0.2 |0.42|
> |0.3|0.2 |0.83|
>
> Moreover, in Appendix G.3 we also investigated the influence of Lipschitz constant on convergence speed in many different Lipschitz architectures (Figure 18 p.50). While increasing the Lipschitz constant (i.e., decreasing the smoothness of the network) leads to slower convergence, this decrease in speed is the smallest for ours among the compared models. In that sense, our method is more stable with respect to these hyperparameters.
>
> > W4+Q4
>
> Concerning the first part of W4, our theoretical guarantee is almost valid for usual practical scenarios. For example, the setting of Theorem 3.5 agrees with our experiments.
>
> As for the potential extensions to other network architectures, we would like to first remind that our work is primarily foundational, providing a novel paradigm for the creation of bi-Lipschitz architectures and addressing some non-negligeable issues of the field. Therefore, this paper focuses on the fundamental design and properties of our model related to the control of bi-Lipschitzness and its theoretical analysis. We acknowledge that there are some applications outside the bi-Lipschitz control framework that we could not investigate. That is also why we have tried to provide an extensive discussion and analyses in the appendix and kept the formulation as general as possible to promote further extensions and other interpretations for future work.
>
> Now, in the context of DEQ and implicit models, the LFT can be indeed re-formulated as finding the solution $z$ of $z=x-\nabla F(z) +z$ which corresponds to a DEQ. In that sense, our model can be regarded as a *bi-Lipschitz* DEQ. We believe this novel interpretation is interesting for future work to increase the generality of our model.
>
> Furthermore, as mentioned in the first answer, we can extend our BLNN to convolutional layers as well. We just have to change the core neural network of the BLNN, which is the ICNN, to the convolutional ICNN proposed in the original work of Amos et al. (2017).
>
> > It seems that […] discussion on potential negative societal impacts […] is still missing.
>
> We will ensure that a more detailed discussion is included in the revised version.
>
> We hope we addressed all your questions and concerns. We will be glad to provide further explanation and clarification if necessary.
> Moreover, clarity is one of our major concerns. If you have any recommendations or some parts of the paper that were difficult to understand, we will be glad to improve them for the next version.

---

> > ### Comment · Reviewer_XcLq · 2024-08-10
> >
> > Thanks for the detailed response. I maintain my positive evaluation of this paper.

---

> > > ### Author Response · Authors · 2024-08-10
> > >
> > > Thank you very much for considering our rebuttal and for your positive position on the acceptance of our paper!

---

### Official Review · Reviewer_u3yR · 2024-07-17

**Soundness:** 3
**Presentation:** 4
**Contribution:** 3
**Rating:** 7
**Confidence:** 4

**Summary:**

This paper proposes to control the bi-Lipschitzness of a neural-network by parameterizing the output by the Legendre-Fenchel-Dual. This involves parameterizing a strongly convex function and computing the minimum of that function in the forward pass. Several benchmarks are studied in simple regression tasks and uncertainty quantification.

**Strengths:**

-The framework is interesting because it parameterizes bi-Lipschitz networks in a way that is not layer-wise and instead takes advantage of the Legendre-Fenchel transform LFT / convex conjugate of parameterized strongly-convex functions (ICNN), which only modifies the output of the output.

-Computing the LFT of a given function can be costly, however the paper offers a non-asymptotic bound for the Lipschitz constant and tractable gradient.

-The experimental results show a considerable improvement in tightness and regularity over other Lipschitz controlled networks like spectral normalization, AOL and Sandwich layers on small regression tasks. In particular BiLipNet behaves a lot better when the Lipschitz constant is overestimated in existing parameterizations.

**Weaknesses:**

-Computing the LFT seems to be quite expensive, hence why the experiments are only on simple 2d problems and fashion-MNIST. For this reason I'm doubtful that it will be used for any large-scale network training pipelines where tight Lipschitz control and estimation is challenging.

-The provable approximation class is limited to alpha-strongly monotone functions and is the derivative of some function almost everywhere. Lipschitz layers like AOL, SLL and Sandwich layer are all solutions to the LipSDP framework which only requires the activations themselves to be alpha-strongly monotone for alpha >= 0 (Fazlyab et al., 2019).

**Questions:**

-Is it also necessary that BLNN is a strongly monotone function? It seems that many of the regression experiments involve monotone target functions (figure 2 and 3), but I'm not sure if that is because BLNN is not capable of representing monotone functions or just not a great representer due to your approximation theorem. If it can represent non-monotone functions, it would be interesting to see a simple regression comparison to SLL, AOL, etc. Answering this question will greatly help my evaluation.

-The Lipschitz parameterization of SLL, AOL, and Sandwich layers commonly uses compositions of 1-Lipschitz layers for the application of certified robustness. How would the BLNN parameterization compare to existing 1-Lipschitz layer networks in the certified robustness setting? I’d imagine BLNN might be much more expressive than composed 1-Lipschitz layers which could have a big impact.

-Have you considered amortizing the LFT computation as done in this paper? https://arxiv.org/abs/2210.12153

-I'm curious if there is any possibility of extending BLNN to convolutional layers? These settings are interesting for larger image classification problems like CIFAR10 and Imagenet.

**Limitations:**

Limitation are adequately addressed.

---

> ### Author Rebuttal · Authors · 2024-08-06
>
> Thank you very much for spending your time on carefully reviewing our paper. We really appreciate all the questions highlighting the significant potential and future directions of our work. Please find below answers to your questions and some clarifications to other important points of your review.
>
> (W=Weakness, Q=Question)
>
> >Q4
>
> Yes, we can extend our BLNN to convolutional layers. We just have to change the core neural network of the BLNN, which is the ICNN, to the convolutional ICNN proposed in the original work of Amos et al. (2017). We will definitely clarify this in the revised version. In the next answer, we provide experimental results with this architecture.
>
> > W1
>
> We conducted an additional experiment using the convolutional version of BLNN (BLNNconv) on the problem of uncertainty estimation (as in Subsection 4.2) with the CIFAR-10 vs. SVHN dataset to illustrate the scalability of our method. For this problem, we implemented the model so that we first process the data through BLNNconv and then transfer it to the DUQ. The result compared to DUQ is as below. Our model is not only scalable to large-scale networks but also improves out-of-detection performance (the AUROC of SVHN). Using BLNNconv instead of the fully connected BLNN also improved the computation time (e.g., 2.5 times faster for the 5 first iterations). Please also refer to the global rebuttal for this topic.
>
> |      | Accuracy  | Loss | AUROC SVHN |
> |-|-|-|-|
> | DUQ   | 0.929    | 0.04 | 0.83|
> | BLNNconv | 0.930    | 0.04 | **0.89**   |
>
> > W2
>
> This is indeed a limitation of our algorithm but also an improvement compared to prior works in some aspects. From the theoretical perspective, several (bi-)Lipschitz layers do not provide a provable approximation class, and, from the practical perspective, some bi-Lipschitz models are also restricted to strongly monotone functions (e.g., BiLipNet). We would like to add that most existing layer-wise approaches have difficulty in creating partially bi-Lipschitz networks (e.g., controlling the spectral norm of linear matrices cannot handle this case) but ours can easily realize this with the PBLNN. Finally, while out of the scope of this work, there are some straightforward ways to increase the expressive power of BLNN. Please see the next answer.
>
> > Q1
>
> The following answer is also related to W1 and W2, and is therefore a bit longer.
>
> Yes, the vanilla BLNN is necessarily a strongly monotone function. However, our model still has a higher expressive power and tighter bounds than other (bi-)Lipschitz units (= the most basic architecture guaranteed to be bi-Lipschitz without using composition of several bi-Lipschitz entities). Indeed, most existing models can only create simple (bi-)Lipschitz units with low expressive power (e.g., only linear) and they have to compose those units to achieve higher expressive power, but this leads to looser bounds. In comparison, our model can **with only one unit** achieve high expressive power and keep tight bounds. This is an improvement realized thanks to a parameterization not layer wise. Experiments of figures 2 and 3 were designed to illustrate this fact.
>
> Now, we can of course, like any previous (bi-)Lipschitz methods, start to employ our BLNN unit as a component of a deeper network. For example, we can successively compose several BLNNs to improve its expressive power. With this approach, it **can indeed represent non-monotone functions**. We have tested to fit the sign function and the composition of 5 BLNNs achieves an accuracy around 0.
>
> Some other extensions to increase the flexibility were already mentioned such as the PBLNN, and the BLNN used as a pre-processing stage for the next networks as we did in the answer to W1. Future research should work on how to use this novel unit in various fields inside and outside bi-Lipschitz problems.
>
> Nevertheless, this direction is slightly out of the scope for this work. Indeed, our work is primarily foundational, providing a novel paradigm for the creation of bi-Lipschitz architectures and addressing some non-negligeable issues of the field. Therefore, this paper focuses on the fundamental design and properties of our model related to the control of bi-Lipschitzness and its theoretical analysis. One of the main contributions of this work resides thus in the construction of such a general framework for improved bi-Lipschitz control equipped with theoretical guarantees. While we ran several experiments related to bi-Lipschitz control to illustrate the benefits of our model, we acknowledge that there are some applications outside the bi-Lipschitz control framework that we could not investigate. This is also why we have tried to provide an extensive discussion and analyses in the appendix and kept the formulation as general as possible to promote further extensions and encourage other interpretations for future work.
>
> > Q2
>
> Thank you for pointing out this interesting possibility. We also believe that BLNN might perform better in this setting by composing many BLNNs. However, this is out of the scope for this paper as explained in the above answer. This is still one of the first avenues we will investigate for future works.
>
> >Q3
>
> Amortizing the LFT computation as the cited paper may be a fundamental solution to the computational complexity of our model (which can already at least scale to CIFAR10). However, this approach is currently out of the scope of this work as the main goal of our paper is to provide a bi-Lipschitz architecture with guaranteed bi-Lipschitz control and build its foundation starting with well-known basic algorithms such as the gradient descent. This amortization technique is more involved as we have to carefully analyse how such amortization influences the provable bi-Lipschitz bounds of the architecture, which is not trivial.
>
> We hope we addressed all your questions and concerns. We will be glad to provide further explanation and clarification if necessary.

---

> > ### Comment · Reviewer_u3yR · 2024-08-12
> >
> > Thank you for clarifying the representation power of BLNN and its application to convolutional. I agree that the parameterization is quite convenient when considering compositions of BLNN blocks over other Lipschitz constrained layers. I think the framework is generally quite interesting so I will raise my score.
> >
> > Regarding my question about amortization of LFT: I would certainly not expect you to produce these results during the rebuttal phase, but I thought it could be a helpful reference for alleviating the computational costs of BLNN.

---

> > > ### Author Response · Authors · 2024-08-13
> > >
> > > Thank you very much for considering our rebuttal and for your positive evaluation of our work! We greatly appreciate your insight regarding the amortization of LFT, as we also believe this could be a solution to improve the scalability of our method. We will mention this in the main paper and further analyze it in future work.

---

### Author Rebuttal · Authors · 2024-08-06

# Time and Space Complexity of the BLNN and its Variants

This global rebuttal discusses the time and space complexity of the BLNN and its variants. Figures can be found in the attached PDF. This discussion will be added to the updated version of the paper.

## Theoretical Discussion

Concerning the time complexity, a forward pass of BLNN with $T$ iterations for the LFT has a time complexity of $O(T)$, where we supposed that the computation of a neural network is $O(1)$. Based on Theorem 3.7, the backward pass will require $O(d_x^3+d_\theta)$ as we have to solve one linear system and compute one Hessian vector product consecutively. $d_x$ is the dimension of the input and $d_\theta$ is the number of parameters. Concerning the space complexity, using Theorem 3.7, the storage capacity is $O(d_x+d_\theta)$ independent of $T$. If we are in an over-parameterized regime with $d_\theta>>d_x^3$ then we can neglect the dependence on $d_x$. With the brute force method (i.e., without Theorem 3.7), the forward is at least of $\Omega(T(d_x+d_\theta))$ and the storage capacity is at least proportional to $T$. Therefore, if we are in an over-parameterized regime with large iteration steps $T$, then the approach of Theorem 3.7 becomes more scalable in both time and memory.

## Experiments

We also computed with experiments the time and space complexity for one iteration of BLNN and its variants: BLNN (with brute force backpropagation), BLNN with Theorem 3.7, PBLNN with only one variable constrained to be bi-Lipschitz, and a traditional feed forward network as a control. Respective parameter sizes are 1.42M, 1.42M, 1.11M and 1.38M. The input size was 3x32x32, simulating a CIFAR dataset, and the batch size was varied in the set {1, 5, 25, 75, 100}.  Results are shown in the attached PDF.

As we can observe in Figures 1 and 2 of the PDF, while both our BLNN and BLNN with Theorem 3.7 present higher time and storage complexity than a traditional feed-forward neural network, Theorem 3.7 greatly contributes to reducing both computational and space requirements (improvements of order of $10$ and $10^2$, respectively). Comparing our model with Theorem 3.7 to a traditional feed-forward neural network, we can conclude that their difference in complexities is only a constant factor of order 10. This explains the scalability of our model to large datasets. Finally, PBLNN provides evidence that we can considerably decrease the complexity of the model by limiting the number of variables we impose bi-Lipschitzness.

---

### Author Response · Authors · 2024-08-12

Thank you once again to all the reviewers for evaluating our work. We extend additional thanks to those reviewers who have already reviewed our rebuttal, as well as to the area chair for managing the review process of our paper. We remain available until the end of the discussion period to discuss and provide further clarifications if any reviewers have remaining or additional concerns regarding our paper.

---

### Decision · Program_Chairs · 2024-09-25

**Decision:**

Accept (poster)

**Comment:**

This paper presents a novel bi-Lipschitz network approach. It is heavily theoretical with proof of concept ML experiments. Reviewers agreed that it is original and insightful. Productive rebuttal period yielded meaningful improvements and added essential information (e.g. time complexity). If accepted, please make sure to incorporate them.